# MYB orchestrates T cell exhaustion and response to checkpoint inhibition

Carlson Tsui[1,12], Lorenz Kretschmer[2,12], Svenja Rapelius[2,12], Sarah S. Gabriel[1],
David Chisanga[3,4,5,6], Konrad Knöpper[7], Daniel T. Utzschneider[1], Simone Nüssing[8,9],
Yang Liao[3,4,5,6], Teisha Mason[1], Santiago Valle Torres[1], Stephen A. Wilcox[4], Krystian Kanev[10],
Sebastian Jarosch[2], Justin Leube[2], Stephen L. Nutt[4], Dietmar Zehn[10], Ian A. Parish[8,9],
Wolfgang Kastenmüller[7], Wei Shi[3,4,5,11], Veit R. Buchholz[2,13✉] & Axel Kallies[1,13✉]

CD8[+] T cells that respond to chronic viral infections or cancer are characterized by the expression of inhibitory receptors such as programmed cell death protein 1 (PD-1) and by the impaired production of cytokines. This state of restrained functionality—which is referred to as T cell exhaustion[1,2]—is maintained by precursors of exhausted T ($T_{PEX}$) cells that express the transcription factor T cell factor 1 (TCF1), self-renew and give rise to TCF1[−] exhausted effector T cells[3–6]. Here we show that the long-term proliferative potential, multipotency and repopulation capacity of exhausted T cells during chronic infection are selectively preserved in a small population of transcriptionally distinct CD62L[+] $T_{PEX}$ cells. The transcription factor MYB is not only essential for the development of CD62L[+] $T_{PEX}$ cells and maintenance of the antiviral CD8[+] T cell response, but also induces functional exhaustion and thereby prevents lethal immunopathology. Furthermore, the proliferative burst in response to PD-1 checkpoint inhibition originates exclusively from CD62L[+] $T_{PEX}$ cells and depends on MYB. Our findings identify CD62L[+] $T_{PEX}$ cells as a stem-like population that is central to the maintenance of long-term antiviral immunity and responsiveness to immunotherapy. Moreover, they show that MYB is a transcriptional orchestrator of two fundamental aspects of exhausted T cell responses: the downregulation of effector function and the long-term preservation of self-renewal capacity.

T cell exhaustion is an important physiological adaptation to continuous antigen stimulation in chronic infection and cancer, and although it protects against excessive immune-mediated tissue damage, it also contributes to viral or tumour persistence[1,2,4,7]. $T_{PEX}$ cells have the ability to continuously self-renew and give rise to functionally restrained effector cells, and therefore have an essential role in maintaining chronically antigen-stimulated T cells and their exhausted phenotype[3–5,8,9]. $T_{PEX}$ cells also mediate the response to therapeutic checkpoint inhibition[3,5,10,11], which can reinvigorate exhausted CD8[+] T cell responses and has revolutionized cancer therapy[12]. In mice, $T_{PEX}$ cells are defined by the co-expression of PD-1, the transcriptional regulators TCF1 and ID3 and the surface molecules CXCR5 and Ly108. By contrast, exhausted effector T ($T_{EX}$) cells co-express PD-1 and TIM-3 but lack the expression of TCF1, ID3, CXCR5 and Ly108 (refs. [3–6,8,9,13]). Thus, exhausted CD8[+] T cells constitute a dynamic network of phenotypically and functionally distinct populations that ultimately depend on the functionality of $T_{PEX}$ cells. We and others have shown that $T_{PEX}$ and $T_{EX}$ cells are controlled by specific transcriptional and metabolic networks that support their differentiation and maintenance[13–18]. It remains, however, unclear how precisely longevity, self-renewal and responsiveness to checkpoint inhibition are orchestrated within the $T_{PEX}$ cell compartment.

## CD62L[+] $T_{PEX}$ cells have stem-like potential

To identify factors that promote the self-renewal and multipotency of $T_{PEX}$ cells, we performed single-cell RNA sequencing (scRNA-seq) of $T_{PEX}$-cell-enriched (PD-1[+]TIM-3[−]) CD8[+] T cells sorted at 30 days post-infection (dpi) from mice chronically infected with lymphocytic choriomeningitis virus (LCMV) clone-13 (Cl13). Combined analysis of our data and publicly available scRNA-seq datasets[11,19] (Fig. 1a and Extended Data Fig. 1a,b) identified two distinct $T_{PEX}$ cell clusters, both marked by high expression of *Tcf7* and *Id3* (Fig. 1a–c). The smaller of these clusters was characterized by high expression of transcripts that are typically associated with naive or central memory T cells, including

[1]Department of Microbiology and Immunology, The Peter Doherty Institute for Infection and Immunity, University of Melbourne, Melbourne, Victoria, Australia. [2]Institute for Medical Microbiology, Immunology and Hygiene, School of Medicine, Technical University of Munich (TUM), Munich, Germany. [3]Olivia Newton-John Cancer Research Institute, Melbourne, Victoria, Australia. [4]The Walter and Eliza Hall Institute of Medical Research, Melbourne, Victoria, Australia. [5]Department of Medical Biology, University of Melbourne, Melbourne, Victoria, Australia. [6]School of Cancer Medicine, La Trobe University, Melbourne, Victoria, Australia. [7]Würzburg Institute of Systems Immunology, Max Planck Research Group, Julius-Maximilians-Universität Würzburg, Würzburg, Germany. [8]Peter MacCallum Cancer Centre, Melbourne, Victoria, Australia. [9]Sir Peter MacCallum Department of Oncology, University of Melbourne, Melbourne, Victoria, Australia. [10]Division of Animal Physiology and Immunology, School of Life Sciences Weihenstephan, Technical University of Munich (TUM), Freising, Germany. [11]School of Computing and Information Systems, University of Melbourne, Melbourne, Victoria, Australia. [12]These authors contributed equally: Carlson Tsui, Lorenz Kretschmer, Svenja Rapelius. [13]These authors jointly supervised this work: Veit R. Buchholz, Axel Kallies. ✉e-mail: veit.buchholz@tum.de; axel.kallies@unimelb.edu.au

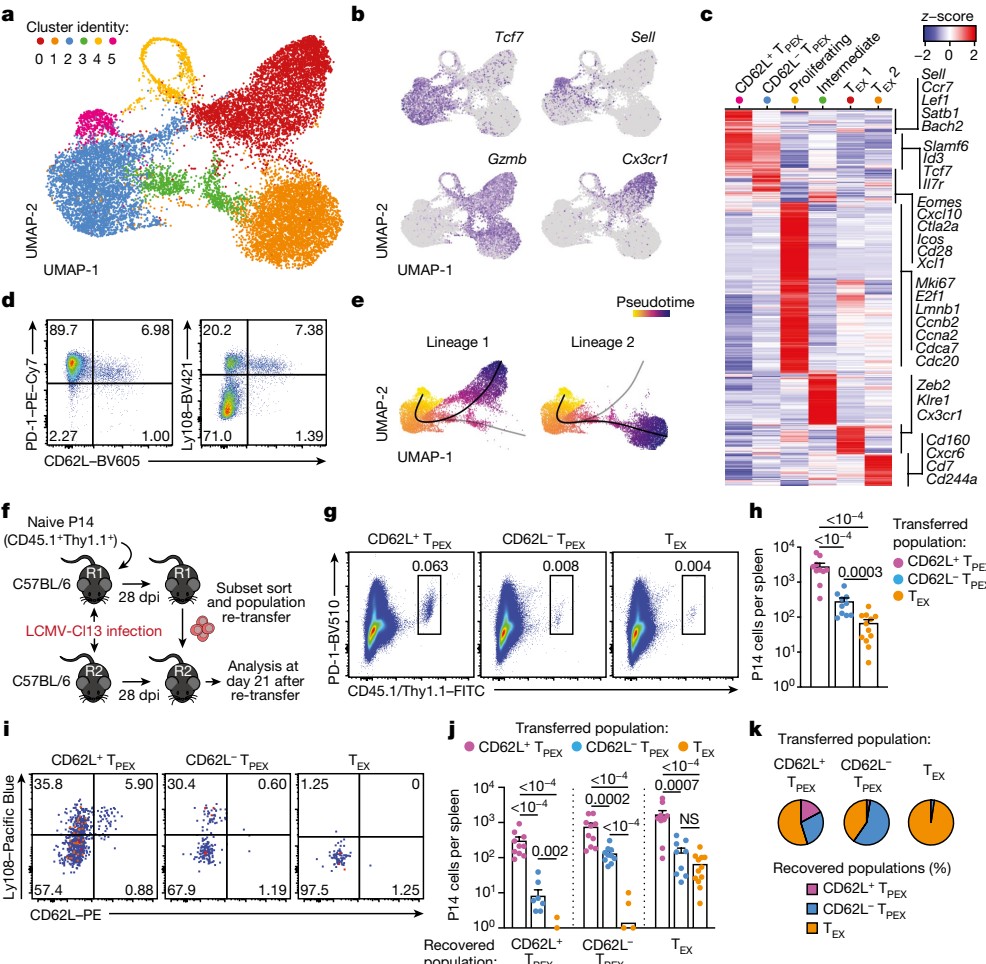

**Fig. 1 | CD62L marks transcriptionally distinct and functionally superior T_PEX cells during chronic infection. a–c**, Naive wild-type mice were infected with LCMV-Cl13 and T_PEX-cell-enriched (PD-1⁺TIM-3^lo) CD8⁺ T cells were sorted and subjected to scRNA-seq at 30 dpi. The resulting data were combined with publicly available scRNA-seq datasets from mouse exhausted CD8⁺ T cells[11,19] and analysed. **a**, Uniform manifold approximation and projection (UMAP) plot of 15,743 single exhausted T cells coloured according to cluster classification. **b**, Normalized gene expression of *Tcf7*, *Sell*, *Gzmb* and *Cx3cr1* projected onto the UMAP. **c**, Heat map showing the expression of all identified cluster signature transcripts. **d**, Congenically marked naive P14 cells were transferred into recipient mice, which were subsequently infected with LCMV-Docile and analysed at 21 dpi. Flow cytometry plots show the expression of PD-1, Ly108 and CD62L in splenic P14 T cells. **e**, UMAP plot showing two predicted developmental trajectories generated using Slingshot analysis. Cells are colour-coded on the basis of pseudotime prediction. **f–k**, Congenically marked naive P14 T cells were transferred into primary recipient (R1) mice, which were then infected with LCMV-Cl13. The indicated subsets of P14 T cells were sorted at 28 dpi and $3 \times 10^3$–$15 \times 10^3$ cells were re-transferred to infection-matched secondary recipient (R2) mice. Splenic P14 T cells of R2 mice were analysed at day 21 after re-transfer. **f**, Schematic of the experimental set-up. **g,h**, Flow cytometry plots (**g**) and cell numbers (**h**) of recovered progenies at day 21 after re-transfer (gated on CD4⁻CD19⁻ cells). **i–k**, Flow cytometry plots (**i**), numbers (**j**) and average percentages (**k**) of recovered CD62L⁺ T_PEX, CD62L⁻ T_PEX and T_EX cells per spleen in R2 mice. Cells were gated on P14 cells (day 21 after re-transfer). Dots in graphs represent individual mice (**h,j**); horizontal lines and error bars of bar graphs indicate mean and s.e.m., respectively. Data are representative of at least two independent experiments. *P* values are from Mann–Whitney tests (**h,j**); *P* > 0.05, not significant (NS).

*Sell* (encoding CD62L), *Ccr7*, *S1pr1*, *Lef1*, *Satb1* and *Bach2* (referred to as CD62L⁺ T_PEX cells; Fig. 1a–c and Supplementary Table 1). By contrast, the larger T_PEX cell cluster showed low expression of *Sell* but was enriched for other T_PEX-cell-associated transcripts, including *Icos*, *Xcl1*, *Cxcl10*, *Cd28* and *Eomes* (CD62L⁻ T_PEX cells; Fig. 1a–c and Supplementary Table 1). In line with previous findings[20,21], we identified two T_EX cell clusters, both marked by the expression of *Gzmb* and lack of *Tcf7*, but distinguished by the differential expression of *Cx3cr1* (Fig. 1a–c and Supplementary Table 1). The two remaining clusters expressed intermediate levels of both T_PEX and T_EX cell marker genes (cluster 3) or cell-cycle-related genes such as *Mki67*, *Ccnb2* and *E2f1* (cluster 4) (Fig. 1a–c and Supplementary Table 1). To examine the heterogeneity of T_PEX cells experimentally, we used CD8⁺ *Id3^GFP* P14 T cells, which express a transgenic T cell receptor (TCR) specific for the LCMV epitope gp33, and GFP under the control of *Id3*, specific to T_PEX cells[13]. *Id3^GFP* P14 cells

were adoptively transferred into naive mice, which were subsequently inoculated with LCMV-Docile, which causes chronic infection (Fig. 1d and Extended Data Fig. 1c–j). Both early T_PEX and T_EX cells were readily detectable during the acute phase (5–9 dpi) of the immune response (Extended Data Fig. 1d), and about 30% of T_PEX cells expressed CD62L, which gradually declined and stabilized at around 10% by three weeks after infection (Fig. 1d and Extended Data Fig. 1d,e). CD62L⁺ T_PEX cells were enriched in the spleen and lymph nodes, but largely absent from the blood, bone marrow and liver (Extended Data Fig. 1f). CD62L⁺ and CD62L⁻ T_PEX cells expressed high levels of PD-1, the activation marker CD44, the exhaustion-associated transcription factor TOX and the co-stimulatory molecule ICOS (Extended Data Fig. 1g,i), indicating that both populations were chronically stimulated, and both expressed low amounts of CD160, 2B4 and TIGIT (Extended Data Fig. 1h,j). Consistent with the notion that T_PEX cells are particularly dependent on strong

TCR signals[13,15], both CD62L[+] and CD62L[−] T$_{PEX}$ cells expressed higher levels of the TCR-induced transcriptional regulator NUR77 than T$_{EX}$ cells (Extended Data Fig. 1k–p). There were no major differences in cytokine production between the two T$_{PEX}$ subsets, but IFNγ[+] cells were enriched among CD62L[+] T$_{PEX}$ cells (Extended Data Fig. 2a,b). CD62L[+] T$_{PEX}$ cells were also found among endogenous gp33-specific and among polyclonal antigen-responsive PD-1[+]CD8[+] T cells in LCMV-Docile-infected mice (Extended Data Fig. 2c–e). Notably, CD62L[+] T$_{PEX}$ cells were transcriptionally distinct from both naive and memory T cells derived from acute LCMV infection (Extended Data Fig. 2f,g).

Slingshot analysis of our scRNA-seq data revealed a developmental trajectory that began with CD62L[+] T$_{PEX}$ cells and progressed into CD62L[−] T$_{PEX}$ cells, from which it bifurcated into either CX3CR1[+] or CX3CR1[−] T$_{EX}$ cells (Fig. 1e). Similar results were obtained when we sorted P14 T$_{PEX}$ cells based on a *Tcf7*$^{GFP}$ reporter from LCMV-Cl13-infected mice and performed scRNA-seq followed by RNA velocity analysis (Extended Data Fig. 2h–j). Overall, these data suggest a one-way developmental trajectory that originates from CD62L[+] T$_{PEX}$ cells. To test this model experimentally, we sorted CD62L[+] T$_{PEX}$, CD62L[−] T$_{PEX}$ and T$_{EX}$ P14 cells on day 28 after infection with LCMV-Cl13, separately re-transferred them into congenically marked infection-matched hosts and analysed three weeks later (Fig. 1f–k). Compared with CD62L[−] T$_{PEX}$ and T$_{EX}$ cells, CD62L[+] T$_{PEX}$ cells showed a superior repopulation capacity (Fig. 1g,h) and were able to efficiently self-renew and give rise to both CD62L[−] T$_{PEX}$ and T$_{EX}$ cells (Fig. 1i–k). These characteristics were maintained even after repetitive adoptive transfers (Extended Data Fig. 3a–f). By contrast, the few CD62L[+]T$_{EX}$ cells that were detected in the P14 compartment (around 1–2%) did not expand or generate progeny efficiently (Extended Data Fig. 3g–l). We confirmed the superior developmental properties of CD62L[+] T$_{PEX}$ cells using single T cell transfer and fate-mapping via retrogenic colour barcoding[22–25] (Extended Data Fig. 4). Notably, single CD62L[+] T$_{PEX}$ cells exhibited self-renewal and multipotent repopulation capacity, akin to single naive T cells (Extended Data Fig. 4a–h). In line with the epigenetic imprint of exhaustion[26–28], progeny derived from single CD62L[+] T$_{PEX}$ cells maintained high levels of PD-1 expression compared to their naive-derived counterparts (Extended Data Fig. 4d,g,i). The CD62L-linked developmental hierarchy uncovered here is unrelated to previously proposed T$_{PEX}$ cell subsets based on differential CD69 expression[29] (Extended Data Fig. 5a–j). Together, these results show that CD62L[+] T$_{PEX}$ cells represent a transcriptionally distinct population with stem-like developmental capacity that maintains the responses of exhausted CD8[+] T cells during chronic infection.

## MYB governs exhausted T cell function and longevity

Functional annotation of our scRNA-seq data identified *Myb*, encoding the transcription factor MYB (also called c-Myb), as specifically enriched among CD62L[+] T$_{PEX}$ cells (Fig. 2a,b and Supplementary Table 1). MYB has important roles in the self-renewal of haematopoietic stem cells and cancer cells[30], T cell leukaemia[31] and CD8[+] memory T cells[32,33]. To characterize the dynamics of *Myb* expression in chronic infection, we infected *Myb*$^{GFP}$ reporter mice[34] with LCMV-Docile (Fig. 2c and Extended Data Fig. 5k), and found that the expression of *Myb* was highest in CD62L[+] T$_{PEX}$ cells (Fig. 2c and Extended Data Fig. 5k). *Myb* expression in CD8[+] T cells responding to LCMV-Docile infection was significantly higher than in those responding to LCMV-Armstrong infection (Extended Data Fig. 5l), and was further enhanced by the inhibition of PD-1 signalling in vivo (Extended Data Fig. 5m–o). Moreover, in vitro TCR stimulation induced the expression of *Myb* in a dose-dependent manner (Extended Data Fig. 5p). Finally, the proportions of CD62L[+] antigen-specific CD8[+] T cells were 10-fold higher in LCMV-Docile versus LCMV-Armstrong infection (Extended Data Fig. 5q,r). Together, these data indicate that strong and persistent TCR stimulation favours the sustained expression of MYB and retention of CD62L[+] T$_{PEX}$ cells during chronic infection.

To study the role of MYB in CD8[+] T cells during viral infection, we infected *Myb*$^{fl/fl}$*Cd4*$^{Cre}$ mice[35] (which lack MYB specifically in T cells) and *Myb*$^{fl/fl}$ (control) littermates with LCMV-Docile or LCMV-Armstrong (Fig. 2d). Before infection, *Myb*$^{fl/fl}$*Cd4*$^{Cre}$ mice showed no major abnormalities in the thymic and mature CD8[+] T cell compartments (Extended Data Fig. 6). LCMV-Armstrong-infected *Myb*$^{fl/fl}$*Cd4*$^{Cre}$ mice mounted CD8[+] T cell responses that were similar to those of controls, and showed no overt signs of disease (Fig. 2e,f and Extended Data Fig. 7a–d). By contrast, LCMV-Docile-infected *Myb*$^{fl/fl}$*Cd4*$^{Cre}$ but not control mice exhibited signs of severe immunopathology and most became moribund within 10 dpi (Fig. 2g and Extended Data Fig. 7e–i). Depletion of CD8[+] T cells averted these symptoms and protected LCMV-Docile-infected *Myb*$^{fl/fl}$*Cd4*$^{Cre}$ mice (Extended Data Fig. 7j,k), indicating that MYB-deficient CD8[+] T cells mediated the fatal immunopathology in chronic LCMV infection. In line with these findings, splenic gp33[+]CD8[+] T cells accumulated at increased frequencies in *Myb*$^{fl/fl}$*Cd4*$^{Cre}$ mice at 8 dpi (Fig. 2h). MYB-deficient T$_{PEX}$ and T$_{EX}$ cells expressed significantly higher levels of IFNγ and TNF, whereas T$_{EX}$ cells also expressed more granzyme B and underwent increased proliferation (measured by the expression of Ki67) compared to controls (Fig. 2i and Extended Data Fig. 7l–o), despite the viral titres being similar in both groups of mice (Extended Data Fig. 7p). MYB-deficient gp33[+]CD8[+] T cells showed increased expression of the inhibitory receptors PD-1 and TIM-3 compared to control cells (Extended Data Fig. 7q), which suggests that the increased function and proliferation of effector cells was not due to impaired expression of inhibitory receptors. Notably, CD62L[+] T$_{PEX}$ cells were specifically lost in the absence of MYB (Fig. 2j and Extended Data Fig. 7r,s).

To longitudinally examine the cell-intrinsic role of MYB in the absence of potentially confounding immune pathology, we generated mixed bone marrow chimeric mice that contained small numbers of *Myb*$^{fl/fl}$*Cd4*$^{Cre}$ (10–20%) and congenically marked *Cd4*$^{Cre}$ control CD8[+] T cells and infected them with LCMV-Docile (Fig. 2k–n and Extended Data Fig. 8). Consistent with our observations in non-chimeric mice, MYB-deficient antigen-specific CD8[+] T cells proliferated more and exhibited increased expression of effector molecules compared to controls (Extended Data Fig. 8a–e). Similarly, the MYB-deficient CD8[+] T cell compartment was devoid of CD62L[+] T$_{PEX}$ cells (Extended Data Fig. 8f,g). Although MYB-deficient TCF1[+] T$_{PEX}$ cells initially developed, they were poorly maintained (Fig. 2l and Extended Data Fig. 8h–j). This observation concurred with a premature termination of cell-cycle activity in MYB-deficient T$_{PEX}$ and T$_{EX}$ cells and a marked contraction of the entire antigen-specific compartment (Fig. 2m,n and Extended Data Fig. 8k–n). Similar results were obtained from adoptively transferred MYB-deficient and control P14 T cells (Extended Data Fig. 9a–g). Thus, MYB mediates the development of CD62L[+] T$_{PEX}$ cells and functional exhaustion of CD8[+] T cells during the acute phase, sustains long-term proliferative capacity and prevents the attrition of antigen-specific T cells during the chronic phase of infection.

## MYB orchestrates exhausted T cell transcription

We next sorted MYB-deficient and control P14 T$_{PEX}$ cells from LCMV-Docile-infected mice and performed transcriptional profiling by RNA-seq (Extended Data Fig. 10a). The analysis showed that there was a loss of the CD62L[+] T$_{PEX}$ cell signature in MYB-deficient compared with control T$_{PEX}$ cells (Extended Data Fig. 10b), confirming that MYB deficiency resulted in the loss of CD62L[+] T$_{PEX}$ cells and not merely CD62L expression. We next performed RNA-seq of control and MYB-deficient P14 T$_{EX}$ cells and T$_{PEX}$ cells sorted for differential expression of CD62L (Fig. 3a). The analysis revealed transcriptional divergence between all subsets and identified 584 differentially expressed genes ($P < 0.05$) between control CD62L[+] and CD62L[−] T$_{PEX}$ cells (Fig. 3b, Extended Data Fig. 10c and Supplementary Table 2). CD62L[+] T$_{PEX}$ cells expressed higher levels of transcripts that encode molecules related to lymph node homing (for example, *Sell*, *Ccr7* and *S1pr1*), and higher levels of the cell-cycle

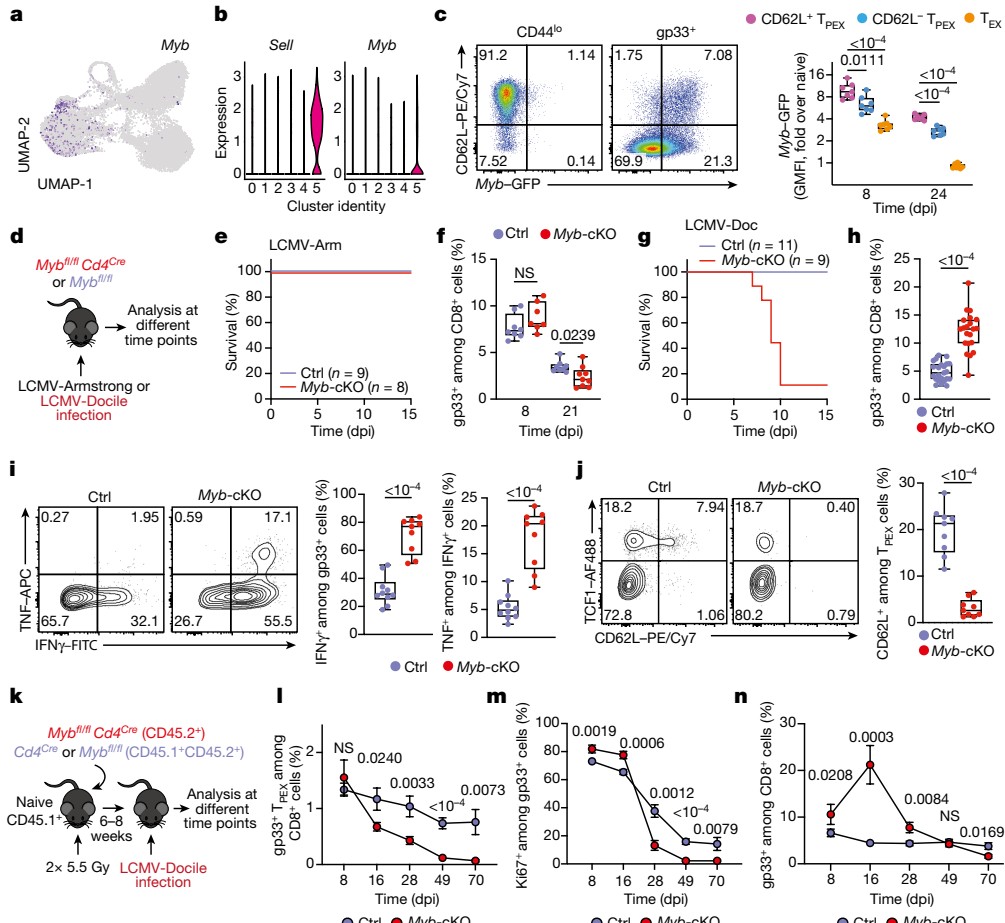

**Fig. 2 | The transcription factor MYB is required for the generation of CD62L⁺ T_PEX cells and the functional exhaustion of T cells during chronic infection. a**, Normalized gene expression of *Myb* projected onto the UMAP plot. **b**, Violin plots showing normalized expression of *Sell* and *Myb*. **c**, *Myb^GFP* reporter mice were infected with LCMV-Docile and splenic CD8⁺ T cells were analysed at the indicated time points after infection. Left, representative flow cytometry plots showing the expression of CD62L and *Myb*–GFP among naive (CD44^lo) and gp33⁺ CD8⁺ T cells. Right, quantification showing the geometric mean fluorescence intensity (GMFI) of *Myb*–GFP among CD62L⁺ T_PEX, CD62L⁻ T_PEX and T_EX cells as fold change over naive CD8⁺ T cells. **d–j**, *Myb^fl/fl Cd4^Cre* (*Myb*-cKO) and littermate *Myb^fl/fl* control (Ctrl) mice were infected with either LCMV-Armstrong (LCMV-Arm) or LCMV-Docile (LCMV-Doc). **d**, Schematic of the experimental set-up. **e–h**, Survival curves of *Myb*-cKO and control mice and box plots showing the frequencies of gp33⁺CD8⁺ T cells at the indicated time

points after infection with LCMV-Armstrong (**e,f**) or LCMV-Docile (**g,h**). **i,j**, Flow cytometry plots and quantification showing the expression of IFNγ and TNF after gp33 peptide restimulation (**i**) and the frequencies of CD62L⁺ T_PEX cells (**j**). Cells were gated on gp33⁺ cells; 8 dpi. **k–n**, Mixed bone marrow chimeric mice containing *Myb*-cKO and *Cd4^Cre* control T cells were infected with LCMV-Docile and analysed at the indicated time points. **k**, Schematic of the experimental set-up. **l–n**, Quantifications show the frequencies of gp33⁺ T_PEX cells (**l**), Ki67⁺ cells (**m**) and gp33⁺ cells (**n**). Dots represent individual mice; symbols and error bars represent mean and s.e.m., respectively; box plots indicate minimum and maximum values (whiskers), interquartile range (box limits) and median (centre line). Data are representative of all analysed mice (**e,g**), two (**c,f,i,j,l–n**) or three independent experiments (**h**). *P* values are from two-tailed unpaired *t*-tests (**c,f,h–j**) and Mann–Whitney tests (**l–n**).

inhibitors *Cdkn1b* and *Cdkn2d* and the quiescence factors *Klf2* and *Klf3*, compared with CD62L⁻ T_PEX cells. Genes that were upregulated in CD62L⁻ T_PEX cells included those that encode positive cell-cycle regulators (*E2f1*, *Cdc6*, *Skp2*, *Cdc25a* and *Kif14*), metabolic enzymes (*P2rx7*, *Hk2*, *Pfkm*, *Pkm* and *Gpd2*) and nutrient transporters (*Slc7a5*, *Slc19a2* and *Slc25a10*) (Extended Data Fig. 10c and Supplementary Table 2). A comparison of MYB-deficient and control CD62L⁻ T_PEX cells identified 580 differentially expressed genes (Supplementary Table 2), including genes that encode molecules related to T cell exhaustion and T_PEX cell identity (*Lef1*, *Eomes*, *Ctla2a*, *Irf4*, *Ikzf2*, *Nt5e* and *Cd160*), cell-cycle regulation and stem cell renewal (*E2f1*, *Rbl2*, *Kif14*, *Cdc25b*, *Bmp7* and *Wnt3*) (Fig. 3c). Consistent with the impaired expression of transcripts related to cell migration and lymph node homing (*Ccr7*, *Cxcr5*, *S1pr1*, *Itgb1* and *Itgb3*), MYB-deficient antigen-specific CD8⁺ T cells were largely excluded from the lymph nodes (Extended Data Fig. 10d,e). We also observed increased expression of *Kit*–encoding KIT, which is involved in haematopoiesis and T cell activation[36,37]–in

MYB-deficient versus wild-type T_PEX cells (Fig. 3c). Indeed, KIT was exclusively expressed in CD62L⁻ T_PEX cells and was highly upregulated in MYB-deficient T_PEX cells (Fig. 3d and Extended Data Fig. 10f,g). A comparison of MYB-deficient and control T_EX cells revealed further significant transcriptional changes (1,532 differentially expressed genes), including the upregulation of transcripts that encode cytotoxic molecules (*Gzma*, *Gzmc* and *Gzme*) or that are related to terminally exhausted T_EX cells (*Cd7*, *Cd244a*, *Cd160*, *Entpd1*, *Id2* and *Cd101*), and the downregulation of transcripts related to CX3CR1⁺ T_EX cells, which have been shown to be more effective in controlling viral burden compared to their CX3CR1⁻ counterparts[20,21] (*Cx3cr1*, *Zeb2*, *Klf2* and *S1pr1*) (Extended Data Fig. 10h–j and Supplementary Table 2). Flow cytometric analysis revealed a lack of CX3CR1⁺ cells and an increase in terminally exhausted CD101⁺ cells among MYB-deficient T_EX cells compared to controls (Extended Data Fig. 10k–l). Consistent with accelerated differentiation into terminally differentiated cells, T_PEX-cell-related transcripts, including *Tcf7*, *Slamf6*, *Lef1* and *Xcl1*, were more strongly

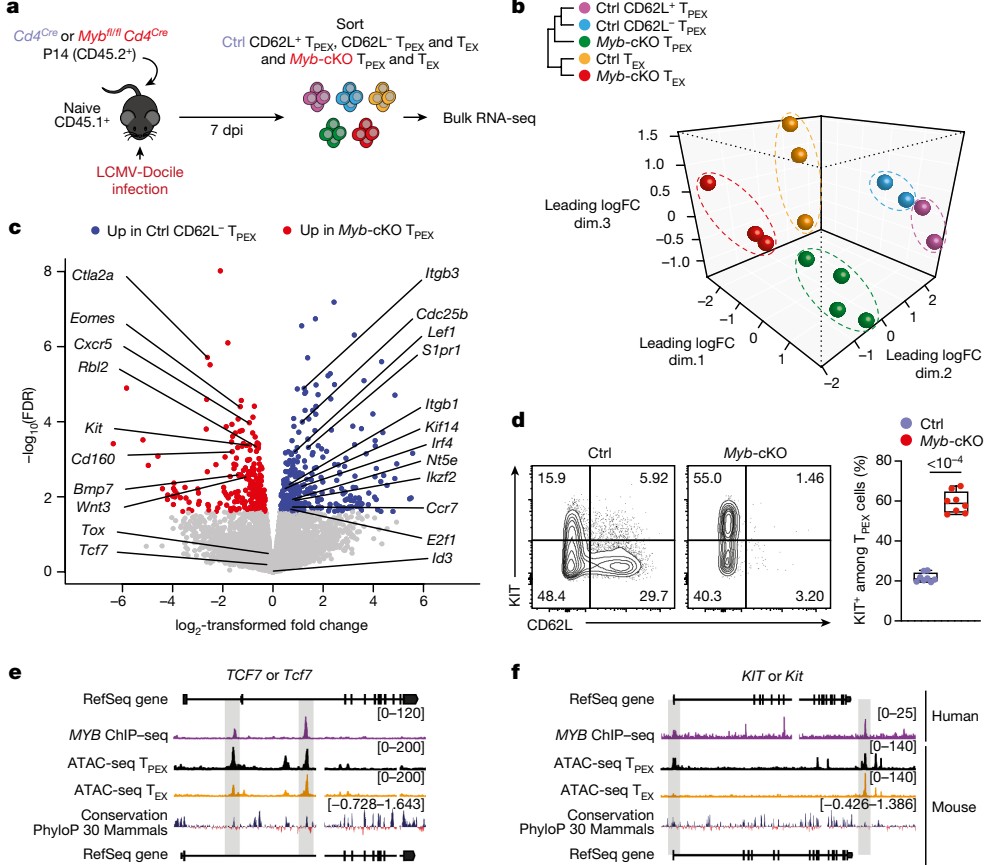

**Fig. 3 | MYB regulates the expression of genes that are critical for the function and maintenance of exhausted T cells. a–c,** Congenically marked *Myb^{fl/fl}Cd4^{Cre}* (*Myb*-cKO) and *Cd4^{Cre}* (control) P14 T cells were adoptively transferred into naive recipient mice, which were then infected with LCMV-Docile. Splenic P14 subsets were sorted at 7 dpi and processed for bulk RNA-seq. **a,** Schematic of the experimental set-up. **b,** Sample dendrogram and three-dimensional scaling plot of all the samples. logFC, log-transformed fold change. **c,** Volcano plot highlighting genes that are differentially expressed (false discovery rate (FDR) < 0.15) between *Myb*-cKO T_{PEX} and control CD62L^- T_{PEX} cells, with genes of interest annotated. **d,** Flow cytometry plots and

quantification show the frequencies of KIT^+ cells among control and *Myb*-cKO T_{PEX} P14 T cells at day 8 after infection with LCMV-Docile (gated on T_{PEX} cells). **e,f,** Tracks show MYB chromatin immunoprecipitation followed by sequencing (ChIP–seq) peaks in the *TCF7* (**e**) and *KIT* (**f**) gene loci of human Jurkat T cells and assay for transposase-accessible chromatin using sequencing (ATAC-seq) peaks of T_{PEX} and T_{EX} cells in the corresponding mouse gene loci aligned according to sequence conservation. Dots in graph represent individual mice; box plots indicate minimum and maximum values (whiskers), interquartile range (box limits) and median (centre line). Data are representative of two independent experiments (**d**). *P* values are from two-tailed unpaired *t*-tests (**d**).

downregulated in MYB-deficient than in control T_{EX} cells (Extended Data Fig. 10h,j). Many of the genes that were dysregulated in the absence of MYB, including *Tcf7*, *Kit*, *Slamf6*, *Lef1*, *Klf2*, *S1pr1*, *Icos*, *E2f1*, *Gzma*, *Gzmc* and *Myb* itself, contained MYB-binding regions in human T cells[38] (Supplementary Table 3), which were conserved and aligned with open chromatin regions in mouse exhausted T cells[13] (Fig. 3e,f, Extended Data Fig. 11a–d and Supplementary Table 3). Together, our results show that MYB is a central transcriptional orchestrator of T cell exhaustion that mediates the development of CD62L^+ T_{PEX} cells and restrains the terminal differentiation of exhausted T cells.

## CD62L^+ T_{PEX} cells fuel therapeutic reinvigoration

To test the functional potential of MYB-dependent CD62L^+ T_{PEX} cells, we separately transferred CD62L^+ and CD62L^- T_{PEX} cells as well as T_{EX} cells into either wild-type (Extended Data Fig. 11e–g) or T-cell-deficient *Tcra^{-/-}* mice (Fig. 4a,b), which were then infected with LCMV-Armstrong. Although all subsets maintained high expression of PD-1 (Extended Data Fig. 11g), progeny that were derived from CD62L^+ T_{PEX} cells expanded more efficiently (Fig. 4b and Extended Data Fig. 11f,g), contained more KLRG1^+ effector cells and provided significantly enhanced viral control compared to the other exhausted T cell subsets (Fig. 4b). CD62L^+ T_{PEX}

cells also gave rise to more CX3CR1^+ T_{EX} cells (Extended Data Fig. 11h,i), altogether indicating that they have a superior potential to generate functional effector cells as compared to their CD62L^- counterparts. We next tested the role of PD-1 and therapeutic PD-1 checkpoint blockade in the generation and function of CD62L^+ T_{PEX} cells. To this end, we generated P14 T cells that lack functional *Pdcd1* (encoding PD-1) using CRISPR–Cas9 (Extended Data Fig. 12a–f). Similar to previous studies[39–41], PD-1-deficient P14 T cells exhibited increased clonal expansion in response to LCMV-Docile, as compared with control cells (Extended Data Fig. 12b). Although the frequencies of T_{PEX} and T_{EX} cells were unaffected by the loss of PD-1 (Extended Data Fig. 12c), the frequencies—but not the numbers—of CD62L^+ T_{PEX} cells were markedly decreased compared to control P14 T cells (Extended Data Fig. 12d). This was due to a concurrent increase in the proportions and the absolute numbers of KIT^+ T_{PEX} cells and T_{EX} cells (Extended Data Fig. 12e,f). These results indicate that PD-1 signalling does not affect the development or maintenance of CD62L^+ T_{PEX} cells but limits their differentiation into CD62L^- T_{PEX} and T_{EX} cells. In line with this conclusion, the numbers of CD62L^+ T_{PEX} cells remained stable during PD-1 checkpoint inhibition, whereas the overall population of antigen-responsive PD-1^+CD8^+ T cells expanded robustly (Extended Data Fig. 12g–p). To directly test the role of CD62L^+ T_{PEX} cells in checkpoint blockade, we performed adoptive transfer

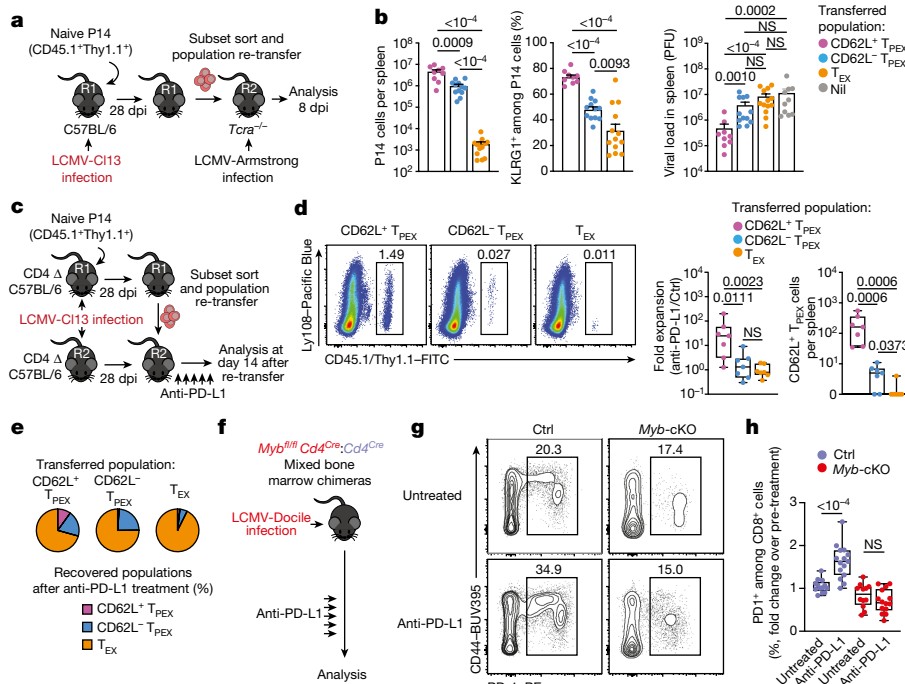

**Fig. 4 | CD62L⁺ T_PEX cells show enhanced potential for effector cell generation and selectively mediate responsiveness to PD-1 checkpoint blocking therapy. a,b**, Congenically marked naive P14 T cells were transferred into primary recipient (R1) mice, which were subsequently infected with LCMV-Cl13. Exhausted T cell subsets were sorted at 28 dpi and $1.0 \times 10^4$–$2.5 \times 10^4$ cells or no cells (Nil) were re-transferred into secondary $Tcra^{-/-}$ recipient (R2) mice. Splenic P14 T cells of R2 mice were analysed 8 days after infection with LCMV-Armstrong. **a**, Schematic of the experimental set-up. **b**, Numbers of recovered P14 T cells (left), percentages of KLRG1⁺ (middle) and splenic viral loads (right). PFU, plaque-forming units. **c–e**, Congenically marked naive P14 T cells were transferred into CD4-depleted R1 mice, which were subsequently infected with LCMV-Cl13. Exhausted T cell subsets were sorted at 28 dpi and re-transferred to infection-matched CD4-depleted (CD4 Δ) R2 mice, treated with anti-PD-L1 antibodies or phosphate-buffered saline (PBS) on days 1, 4, 7, 10 and 13 and analysed at day 14 after re-transfer. **c**, Schematic of the experimental set-up. **d**, Representative flow cytometry plots of splenic progeny derived from transferred T cell subsets after

treatment with anti-PD-L1, at day 14 after re-transfer (cells were gated on CD4⁻CD19⁻PD-1⁺ cells). Box plots show the relative progeny expansion in anti-PD-L1-treated versus PBS-treated mice (left) and the numbers of CD62L⁺ T_PEX cells among progeny after anti-PD-L1 treatment (right). **e**, Average subset distribution. **f–h**, Mixed bone marrow chimeric mice containing congenically marked $Myb$-cKO and $Cd4^{Cre}$ (control) T cells, infected with LCMV-Docile, were treated with anti-PD-L1 on days 33, 36, 39, 42 and 45 and analysed at 49 dpi. **f**, Schematic of the experimental set-up. **g,h**, Representative flow cytometry plots (**g**) and box plot (**h**) showing the fold change of frequencies of splenic polyclonal PD1⁺CD8⁺ T cells in anti-PD-L1-treated versus PBS-treated mice. Cells were gated on CD8⁺ cells; 49 dpi. Dots in graphs represent individual mice; box plots indicate minimum and maximum values (whiskers), interquartile range (box limits) and median (centre line); horizontal lines and error bars of bar graphs indicate mean and s.e.m., respectively. Data are representative of at least two independent experiments (**b,d–e,g–h**). $P$ values are from two-tailed unpaired $t$-tests (**b** (middle), **h**) and Mann–Whitney tests (**b** (left, right, **d**).

experiments in the context of therapeutic PD-1 inhibition (Fig. 4c–e). Re-transferred CD62L⁺ T_PEX cells proliferated strongly in response to PD-1 checkpoint inhibition and generated larger progenies compared with untreated controls, while undergoing concurrent self-renewal (Fig. 4d,e). In stark contrast, both CD62L⁻ T_PEX and T_EX cells showed no apparent proliferative response (Fig. 4d,e), which indicates that CD62L⁺ but not CD62L⁻ T_PEX cells fuel the generation of effector cells in response to checkpoint blockade. Consistent with this conclusion, MYB-deficient antigen-responsive PD-1⁺CD8⁺ T cells, which lack CD62L⁺ T_PEX cells, did not expand in response to PD-1 checkpoint inhibition (Fig. 4f–h). Together, our results reveal that MYB-dependent CD62L⁺ T_PEX cells exclusively fuel the proliferative burst in response to PD-1 checkpoint inhibition and therefore dictate the success of therapeutic checkpoint blockade.

Overall, our data show that the CD8⁺ T cell response in chronic infection is maintained by a small population of distinct T_PEX cells that co-express TCF1, CD62L and the transcription factor MYB. These cells, which we term stem-like exhausted T (T_SLEX) cells here, possess superior self-renewal, multipotency and long-term proliferative capacity compared to their TCF1⁺ but CD62L⁻ descendants. Loss of MYB abrogated the differentiation of T_SLEX cells and severely impaired the the persistence of the entire TCF1⁺ T_PEX cell compartment, ultimately resulting in

the collapse of the complete CD8⁺ T cell response. MYB also mediates functional exhaustion during chronic infection by restricting the initial expansion and effector function of antigen-responsive CD8⁺ effector T cells. As a result, mice that lacked MYB in their T cells succumbed to chronic but not acute viral infection, highlighting that T cell exhaustion is an essential adaptation to chronic infection. Thus, MYB represents a transcriptional checkpoint that instructs the differentiation and function of CD8⁺ T cells in response to severe or chronic infection. Our data also show that T_SLEX cells are exclusively required to mediate the response to PD-1 checkpoint inhibition. These findings not only advance our understanding of the mechanisms of T cell re-invigoration in the context of checkpoint inhibition, but also emphasize the need for new therapeutic strategies that target T_SLEX cells to harness the full potential of T cell-mediated immunotherapy. Furthermore, the superior proliferative and developmental potential of T_SLEX cells makes them prime targets of adoptive T cell transfer and chimeric antigen receptor (CAR) T cell therapies. Finally, our results show that two central but seemingly unrelated properties of exhausted T cells—limited function and longevity—are intimately linked by a single transcription factor MYB; this is a notable example of evolutionary parsimony, which ensures ongoing T cell immunity during chronic infection while preventing collateral damage to the host.

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

# Methods

## Mice and generation of mixed bone marrow chimeric mice

All mice used in this study were on a C57BL/6J background. Age- and sex-matched mice were used for experiments and allocated to experimental groups without further randomization or blinding. CD45.1 or CD45.2 mice were obtained from the Australian Resources Centre or were purchased from Envigo at 6–8 weeks of age. $Id3^{GFP}$ mice[42] expressing the P14 TCR transgene (JAX: Tg(TcrLCMV)327Sdz) were used in some experiments as described[13]. $Myb^{GFP}$ mice and $Myb^{fl/fl}Cd4^{Cre}$ mice were described previously[34]. $Myb^{fl/fl}Cd4^{Cre}$ mice were crossed to include the P14 TCR transgene for some experiments. Littermate $Myb^{fl/fl}$ mice were used as controls. Mixed bone marrow chimeric mice were generated by irradiating CD45.1 host mice (2× 5.5 Gy), before reconstitution with a mix of CD45.1/CD45.2 $Cd4^{Cre}$ bone marrow and CD45.2 $Myb^{fl/fl}Cd4^{Cre}$ bone marrow. Mice were left to recover for six to eight weeks before further experiments. P14 transgenic $Tcf7^{GFP}$ mice on a CD45.1 background were generated in the laboratory of D.Z. by inserting a GFP expression construct into the $Tcf7$ gene locus and will be described in detail elsewhere. P14 mice expressing diverse combinations of the congenic markers CD45.1/.2 and Thy1.1/1.2, as well as $Tcra^{-/-}$ mice were bred under specific-pathogen-free conditions at the mouse facility of the Institute for Medical Microbiology, Immunology and Hygiene at the Technical University of Munich. All mice were maintained and used in accordance with the guidelines of the University of Melbourne Animal Ethics Committee or the district government of upper Bavaria (Department 5–Environment, Health and Consumer Protection).

## Generation of colour-barcoded P14 cells

Retrogenic colour-barcoding was used to heritably label individual P14 cells and their progeny for in vivo single-cell fate-mapping experiments, as previously described[24,25]. In brief, bone marrow was collected from congenically marked (CD45.1+ or CD90.1+) P14 donor mice and stained with Ly6A/E (Sca-1), anti-mouse CD3 and CD19 antibodies, together with propidium iodide for live or dead discrimination. Haematopoietic stem cells (HSCs) were then sorted as live CD3−CD19−Sca-1+ cells and cultured at 37 °C in cDMEM (DMEM (Life Technologies), supplemented with 10% FCS, 0.025% L-glutamine, 0.1% HEPES, 0.1% gentamycin and 1% penicillin/streptomycin), supplemented with 20 ng ml−1 mouse IL-3, 50 ng ml−1 mouse IL-6 and 50 ng ml−1 mouse SCF, for three to four days in tissue-culture-treated 48-well-plates. Expanded stem cells were then retrovirally transduced with constructs encoding the fluorescent proteins GFP, YFP, BFP, CFP and T-Sapphire by spinoculation. After two days in culture, the transduced HSCs were suspended in fetal calf serum (FCS) and injected intravenously into irradiated C57BL/6 recipient mice (2 × 4.5 Gy, with a resting period of 4 h). After several weeks, colour-barcoded naive (CD8+CD44low) P14 cells were sorted from the peripheral blood of retrogenic mice and transferred into C57BL/6 recipients.

## Organ preparation and adoptive T cell transfer

Single-cell suspensions were obtained by mashing total spleens, lymph nodes or bone marrow through a 70-μm nylon cell strainer (BD). For liver samples, lymphocytes were obtained by density gradient centrifugation. Red blood cells were lysed with a hypotonic ammonium chloride-potassium bicarbonate (ACK) or ammonium chloride-Tris (ACT) buffer. For isolating naive CD8+ or transgenic P14 T cells, the mouse CD8+ T cell enrichment kit (Miltenyi Biotech) was used, or cells were sorted as live CD8+CD44low cells.

For primary population transfer experiments, 2,000–10,000 naive P14 T cells were injected into naive congenically marked primary recipients. For adoptive re-transfer experiments, P14 cells were first enriched from the spleens and lymph nodes of primary or secondary recipients by sorting CD45.1+Thy1.1+ cells, followed by staining with anti-mouse

CD62L, anti-mouse Ly108 and the eBioscience Fixable Viability Dye eFluor 780 or propidium iodide for live or dead discrimination. The indicated subsets were then sorted according to their expression profile of CD62L and Ly108 (note: the anti-mouse CD62L antibody was titrated to a dilution that precludes functional blocking of the molecule). Unless specified otherwise, equal numbers of cells of each subset were injected, ranging between 3,000 and 40,000 for secondary transfers and between 1,000 and 3,000 for tertiary transfers. In cases in which the numbers of transferred cells differed between experimental groups (Extended Data Fig. 3), a fold expansion factor was calculated by dividing the number of recovered P14 cells by the number of transferred cells. A 10% take rate was assumed for these calculations, based on our measurements in Extended Data Fig. 4f.

For primary single-cell transfer experiments, naive P14 cells were isolated from the peripheral blood or spleens of naive retrogenic P14 donor mice by staining with anti-mouse CD8, anti-mouse CD44, anti-mouse CD45.1 and anti-mouse Thy1.1. For secondary single-cell re-transfers, anti-mouse CD45.1, anti-mouse CD62L and anti-mouse Ly108 were used, together with the eBioscience Fixable Viability Dye eFluor 780 or propidium iodide for live or dead discrimination. Single P14 cells were then isolated by successively sorting individual cells according to their unique congenic or retrogenic colour barcode and their CD62L/Ly108 phenotype into a 96-well V-bottom plate containing a pellet of $4 \times 10^5$ C57BL/6 splenocytes. The unique congenic and retrogenic colour barcodes of sorted cells enabled the simultaneous transfers of multiple individual cells for fate-mapping. After sorting, the whole content of each well was injected into separate C57BL/6 recipients.

## Gene deletion by CRISPR–Cas9–sgRNA complex electroporation

$Pdcd1$ gene deletion was conducted as reported previously[41]. In brief, P14 cells were purified using an EasySep mouse CD8+ T cell isolation kit (STEMCELL Technologies) according to the manufacturer's instructions, after which cells were electroporated (Pulse DN100) with a complex of Alt-R S.p. Cas9 Nuclease (Integrated DNA Technologies) and a previously described $Pdcd1$-targeting sgRNA (Synthego)[41] using the P3 primary cell 4D-Nucleofector X kit S electroporation kit (Lonza) and Lonza 4D-Nucleofector Core Unit (Lonza). Cells were rested in fully supplemented RPMI medium (see above) at 37 °C for 10 min, after which P14 cells were counted, and 5,000 P14 cells were injected intravenously into recipient mice before infection with LCMV.

## LCMV infections and checkpoint blockade

LCMV-Docile, LCMV-Cl13 and LCMV-Armstrong were propagated and quantified as previously described[26]. For LCMV-Docile and LCMV-Cl13 infection, frozen stocks were diluted in PBS and $2 \times 10^6$ PFU were injected intravenously. For LCMV-Armstrong infection, frozen stocks were diluted in PBS and $2 \times 10^5$ PFU were injected intraperitoneally. For infection of $Tcra^{-/-}$ mice, a dosage of $2 \times 10^3$ PFU was used. For CD4+ T cell depletion, mice were injected twice intraperitoneally with 200 μg per mouse of anti-CD4 monoclonal antibody (GK1.5, BioXCell) one day before and one day after infection with LCMV-Cl13. For CD8+ T cell depletion, mice were injected intraperitoneally with 100 μg per mouse of anti-CD8 monoclonal antibody (YTS-169, BioXCell) on days 1, 3 and 5 of infection. For PD-1 blockade, monoclonal anti-PD-L1 antibodies (B7-H1, BioXCell) were injected intraperitoneally at 200 μg per mouse at the specified days after infection.

## In vitro culture of naive CD8+ T cells

Cell-culture 48-well or 96-well plates were prepared by coating with anti-CD3 at various concentrations for at least 2 h at 4 °C. Control wells were coated with PBS for the same duration. The wells were washed twice using PBS. Enriched naive CD8+ T cells were seeded in the wells and were cultured in RPMI medium supplemented with 10% FCS,

55 µM β-mercaptoethanol, 2 mM Glutamax, 25 mM HEPES buffer and 100 U ml$^{-1}$ penicillin and 10 µg ml$^{-1}$ streptomycin for three days in a humidified incubator at 37 °C with 5% $CO_2$.

## Surface and intracellular antibody staining of mouse cells

Surface staining was performed for 30 min at 4 °C in PBS supplemented with 2% FCS (FACS buffer) with the following antibodies: CD8a (53-6.7, BD), CD44 (IM7, BD), CD45.1 (A20, BD or Biolegend), CD45.2 (104, BD), CD90.1 (HIS52, Thermo Fisher Scientific) CX3CR1 (SA011F11, Biolegend), PD-1 (RMP1-30 or 29F.1A12, Biolegend), CD62L (MEL-14, Biolegend), TIM-3 (RMT3-23, Biolegend), CD101 (Moushi101, Thermo Fisher Scientific), Ly108 (eBio13G3-18D, BD), CD117 (KIT) (ACK2, Thermo Fisher Scientific), CD244 (2B4) (eBio244F4, Thermo Fisher Scientific), CD160 (eBioCNX46-3, eBioscience), TIGIT (GIGD7, Thermo Fisher Scientific) and KLRG1 (2F1, Biolegend). LCMV-derived D$^b$/gp33-41 tetramers were obtained from the NIH Tetramer Facility; tetramer staining was performed for 30–60 min at 4 °C in FACS buffer. Each cell staining reaction was preceded by a 10-min incubation with purified anti-mouse CD16/32 Ab (FcgRII/III block; 2.4G2) and (fixable) viability dye (Thermo Fisher Scientific). For intracellular cytokine staining, splenocytes were ex vivo restimulated with gp33-41 (gp33) peptide (5 mM) for 5 h in the presence of brefeldin A (Sigma) for the last 4.5 h, fixed and permeabilized using the Cytofix/Cytoperm (BD) or transcription factor staining kit (eBioscience) and stained with anti-IFNγ (XMG1.2, Thermo Fisher Scientific), TNF (MP6-XT22, Thermo Fisher Scientific). Other intracellular staining was performed with the Foxp3 transcription factor staining kit (eBioscience) and the following antibodies: TCF1 (C63D9, Cell Signaling), GZMB (MHGB04, Thermo Fisher Scientific) and Ki67 (FM264G, BD).

## In vitro activation of T cells

CD8$^+$ T cells were isolated using the CD8$^+$ T cell enrichment kit (Miltenyi Biotech) and, in some instances, CTV labelled. Wild-type cells were stimulated with plate-bound anti-CD3 at the indicated concentration and in fully supplemented tissue-culture medium (RPMI plus 10% FCS, 2 mM Glutamax, 1 mM pyruvate, 55 µM mercaptoethanol, 100 U ml$^{-1}$ penicillin, 10 µg ml$^{-1}$ streptomycin) and 100 U ml$^{-1}$ IL-2.

## Histology

For immunofluorescence, spleens were embedded and frozen in OCT, sectioned at 15 µm and mounted on SuperFrostPlus Adhesion glass (Thermo Fisher Scientific). Sections were dehydrated using silica beads, fixed with 4% paraformaldehyde for 10 min and washed with PBS. Samples were blocked using 5% normal goat serum for 2 h before staining. Samples were incubated with antibodies against B220 (RA3-6B2, eBioscience), CD3 (17A2, eBioscience) and F4/80 (BM8, Biolegend) diluted in 5% NGS for 2 h at room temperature in the dark. After staining, samples were washed with PBS at least three times. Samples were then mounted using ProLong Gold Antifade Mountant (Invitrogen) and imaged using an inverted LSM780 microscope (Carl Zeiss) and a plan apochromat 63× NA 1.40 oil-immersion objective (Carl Zeiss). For haematoxylin and eosin (H&E) staining, organs were collected and fixed in 10% formalin. Fixed samples were embedded in paraffin and sectioned at 10 µm, mounted on SuperFrostPlus Adhesion glass and stained using H&E. Mounted samples were imaged using a Nikon SMZ1270 Stereo Microscope. Imaging data were analysed using Fiji (ImageJ) software (NIH).

## scRNA-seq and analysis

Relating to the dataset introduced in Fig. 1: T$_{PEX}$-cell-enriched CD8$^+$ T cells were sorted as CD8$^+$PD-1$^+$TIM-3$^{low}$ from the spleens of chronically infected mice (LCMV-Cl13) using a FACSAria III (BD Biosciences). Afterwards, the single cells were encapsulated into droplets with the Chromium™ Controller (10X Genomics) and processed following the manufacturer's specifications. Bead captured transcripts in all encapsulated cells were uniquely barcoded using a combination of a 16-bp 10X barcode and a 10-bp unique molecular identifier (UMI). The Chromium Single Cell 3' Library & Gel Bead Kit v2 for the wild-type untreated sample or v3 for wild type treated with 200 µg of anti-PD-L1 antibody (10F.9G2, BioXCell) for 24 h were used to generate cDNA libraries (10X Genomics) following the protocol provided by the manufacturer. Libraries were quantified by Qubit™ 3.0 Fluometer (Thermo Fisher Scientific) and quality was checked using 2100 Bioanalyzer with High Sensitivity DNA kit (Agilent). For library sequencing the NovaSeq 6000 platform (S1 Cartridge, Illumina) in 50-bp paired-end mode was used. The sequencing data were demultiplexed using Cell-Ranger software (v.2.0.2) and the reads were aligned to the mouse mm10 reference genome using STAR aligner. Aligned reads were used to quantify the expression level of mouse genes and generate the gene-barcode matrix. Subsequent data analysis was performed using Seurat R package (v.3.2)[43]. The sequencing data are available at the National Center for Biotechnology Information (NCBI) Gene Expression Omnibus (GEO) (http://www.ncbi.nlm.nih.gov/geo) under the accession number GSE168282 (ref. [19]). For further analysis in this study, datasets GSM5135522 and GSM5135523 (ref. [19]) were combined with another publicly available scRNA-seq dataset of mouse exhausted CD8$^+$ T cells, accessed from GSE122712 (ref. [11]) and analysed using Seurat R package (v.3.2)[43]. The 2,000 most variable genes were included for the anchoring process and used for downstream analysis to calculate principal components of log-normalized and scaled expression data. On the basis of the principal component analysis (PCA), a UMAP of the identified clusters was visualized. Cluster-specific genes were identified with the FindAllMarker function in Seurat with parameters min. pct = 0.25, logfc.threshold = 0.25. Trajectories were predicted using the Slingshot 1.4.0 package[44], using the function slingshot with default settings and starting with the CD62L$^+$ T$_{PEX}$ cell cluster. The functional annotation tool DAVID (LHRI) was used to interrogate gene sets to identify transcription factors of interest. Selected lists of genes were then further explored using enrichment analyses against existing RNA-seq datasets[13,20].

For the dataset used in Extended Data Fig. 2h–j, T$_{PEX}$ cells were sorted as live CD8$^+$PD1$^+$CD45.1$^+$*Tcf7*$^-$-GFP$^+$ cells from the spleens of chronically infected mice (LCMV-Cl13, 28 dpi) using a MoFlo Astrios cell sorter (Beckman Coulter) and processed using the 10X Genomics technology, according to the manufacturer's protocol (Chromium Single Cell 3' GEM v3 kit). Quality control was performed with a High Sensitivity DNA Kit (Agilent 5067-4626) on a Bioanalyzer 2100, as recommended in the protocol. Libraries were quantified with the Qubit dsDNA HS Assay Kit (Life Technologies Q32851). All steps were performed using RPT filter tips (Starlab) and LoBind tubes (Sigma). The library was sequenced with 20,000 reads per cell. Illumina paired-end sequencing was performed with 150 cycles on a Novaseq 6000. Annotation of the sequencing data was performed using CellRanger software (v.5.0.0) against the mouse reference genome GRCm38 (mm10-2020-A). All subsequent analysis was performed using SCANPY (v.1.6)[45]. After general pre-processing (less than 15% mitochondrial genes, regressing out cell cycle, filtering mitochondrial genes and total counts), the data were count-normalized per cell and logarithmized. RNA velocities were calculated using Velocyto[46] and analysed with scVelo[47]. The sequencing data are available at the NCBI GEO (http://www.ncbi.nlm.nih.gov/geo) under the accession number GSE205608.

## Bulk RNA extraction, sequencing and analysis

Relating to the dataset in Extended Data Fig. 2f,g, the indicated subsets of exhausted P14 cells (CD62L$^+$ T$_{PEX}$, CD62L$^-$ T$_{PEX}$, T$_{EX}$) were sorted from the spleen of chronically infected mice (LCMV-Cl13) at 28 dpi. As a comparison, memory P14 cells were sorted from the spleen of LCMV-Armstrong-infected mice at 28 dpi according to the following phenotypes: CD62L$^+$Ly108$^+$ (CD62L$^+$ memory), CD62L$^-$Ly108$^+$ (CD62L$^-$ memory) and CD62L$^-$Ly108$^-$ (effector). In addition, naive P14 cells were

included for the analysis. RNA extraction from sorted P14 T cells was performed using the RNeasy Plus Mini Kit (Qiagen) according to the manufacturer's instructions. Each sample group consisted of two to three biological replicates. Sequencing was performed on an Illumina Novaseq by Novogene, generating 150-bp paired-end reads. RNA-seq reads were aligned to the mouse reference genome GRCm38/mm10 using STAR (v.2.5.4)[48]. Read counts per gene locus were obtained with htseq-count (v.0.11.4)[49]. Statistical analysis was performed in R (v.3.6.3). Genes with total reads lower than 200 across all samples were excluded. Normalization and differential gene expression analysis was performed using DESeq2 (v.1.26.0). Batch effects were identified using sva (v.3.34.0) and subsequently modelled in the DESeq2 design formula. Genes were considered differentially expressed when they achieved an FDR of less than 0.05 and a $\log_2$-transformed fold change of greater than 1. The sequencing data are available at the NCBI GEO (http://www.ncbi.nlm.nih.gov/geo) under the accession number GSE205608.

Relating to the data in Fig. 3 and Extended Data Fig. 10, RNA extraction from sorted P14 T cells was performed using the RNeasy Plus Mini Kit (Qiagen) according to the manufacturer's instructions. Each sample group consisted of two experimental replicates. All samples were sequenced on an Illumina NextSeq500 generating 80-bp paired-end reads. RNA-seq reads were aligned to the mouse reference genome GRCm38/mm10 using the Subread aligner (Rsubread v.2.2.6)[50]. Gene-level read counts were obtained by running featureCounts[51], a read count summarization program within the Rsubread package[52] and the inbuilt Rsubread annotation that is a modified version of the NCBI RefSeq mouse (mm10) genome annotation build 38.1. Pseudogenes, or genes that did not meet a counts per million reads (CPM) cut-off of 0.5 in at least two libraries were excluded from further analysis. Read counts were converted to $\log_2$-CPM, quantile normalized and precision weighted with the voom function of the limma package[53,54] after accounting for batch effects. A linear model was fitted to each gene, and the empirical Bayes moderated $t$-statistic was used to assess differences in expression[55,56]. Raw $P$ values were adjusted to control the global FDR across all comparisons using the 'global' option in the decideTests function in the limma package. Genes were called differentially expressed if they achieved an FDR of 15% or less. Enrichment analysis of Gene Ontology (GO) terms on the differentially expressed genes was performed using the goana function within the limma package[57]. Pathway enrichment against the Kyoto Encyclopedia of Genes and Genomes (KEGG) pathways on the differentially expressed genes was performed using the kegga function also implemented in the limma package. Gene set enrichment analysis was performed using Gene Set Enrichment Analysis (GSEA) software (v.4.0.3)[58].

## ChIP–seq analysis
Previously published raw ChIP–seq data for the MYB transcription factor[38] were downloaded from the NCBI GEO with accession number GSE59657. Reads were mapped to the human genome (GRCh38) using the align function in Rsubread (refs. [50,52]). Peak calling was performed using Homer (v.4.11)[59] with an FDR set to $1 \times 10^{-8}$. In brief, tags for the aligned libraries were first created using the makeTagsDirectory function within Homer then followed by peak calling using the 'style' factor parameter with called peaks annotated to the nearest genes. Overlap between differentially expressed genes from the RNA-seq data (mouse) and ChIP–seq data (human) was performed by first transforming the human genes associated with each annotated peak to their corresponding mouse homologues using information available in the Ensembl database through the biomaRt Bioconductor package[60]. The two sets of genes were then compared for common genes.

## Analysis of evolutionary conservation
Genomic conservation data for the human and mouse genomes were obtained from UCSC Genome Browser (https://genome.ucsc.edu). Annotated tracks of human ChIP data were manually aligned to annotated tracks of mouse ATAC data using conserved loci of 100 vertebrates against the human genome and conserved loci of 30 mammals against the mouse genome as reference points.

## Flow cytometry
Flow cytometry was performed using a Fortessa or Cytoflex LX (Beckman Coulter) and sort purification was performed on a BD FACSAria Fusion or MoFlo Astrios (Beckman Coulter). All data were analysed using FlowJo 10 (Tree Star). Graphs and statistical analyses were done with Prism 7 (GraphPad Software).

## Statistics
A paired or unpaired Student's $t$-test (two-tailed), Welch's $t$-test, Mann–Whitney $U$ test or one-way ANOVA was used to assess significance. Statistical methods to predetermine sample size were not used.

## Reporting summary
Further information on research design is available in the Nature Research Reporting Summary linked to this article.

## Data availability
Most of the sequencing data generated for this study have been deposited in the NCBI GEO database with accession number GSE188526. The sequencing data shown in Extended Data Fig. 2f–j have been deposited with accession number GSE205608. Source data are provided with this paper.

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

**Acknowledgements** We acknowledge I. Andrä and M. Schiemann for cell sorting, and R. Gloury and I. Hensel for technical help. This work was supported by the National Health and Medical Research Council of Australia (NHMRC) Research Fellowship (to A.K.) and Ideas Grants (APP2004333 to A.K. and C.T.; APP2001719 to I.A.P.); the European Research Council (starting grant 949719 SCIMAP to V.R.B.); the Else Kröner-Fresenius-Stiftung (EKFS 2019_A91 to V.R.B.); the Deutsche Forschungsgemeinschaft (DFG, German Research Foundation; SFB-TRR 338/1 2021–452881907, SFB 1054–210592381 to V.R.B.); and the Deutsche Krebshilfe (DKH 70113918 to V.R.B). A.K. is a Senior Research Fellow of the NHMRC; D.T.U. is a Special Fellow of The

Leukemia & Lymphoma Society and is supported by an NHMRC fellowship (1194779). We acknowledge the Melbourne Cytometry Platform for provision of flow cytometry services and the NIH Tetramer Facility for providing tetramers.

**Author contributions** C.T., L.K., V.R.B. and A.K. designed the study and wrote the manuscript. C.T., L.K. and S.R. performed core experimental work and data analysis. S.S.G. performed initial key experiments. D.C., K. Knöpper, Y.L., W.S., L.K. and S.J. performed computational analyses. K. Kanev, W.K., D.T.U., S.N., T.M., S.V.T., S.A.W., I.A.P., S.J. and J.L. performed supporting experimental work and data analysis. S.L.N., K. Kanev and D.Z. contributed new reagents or analytic tools.

**Competing interests** The authors declare no competing interests.

**Additional information**
**Correspondence and requests for materials** should be addressed to Veit R. Buchholz or Axel Kallies.

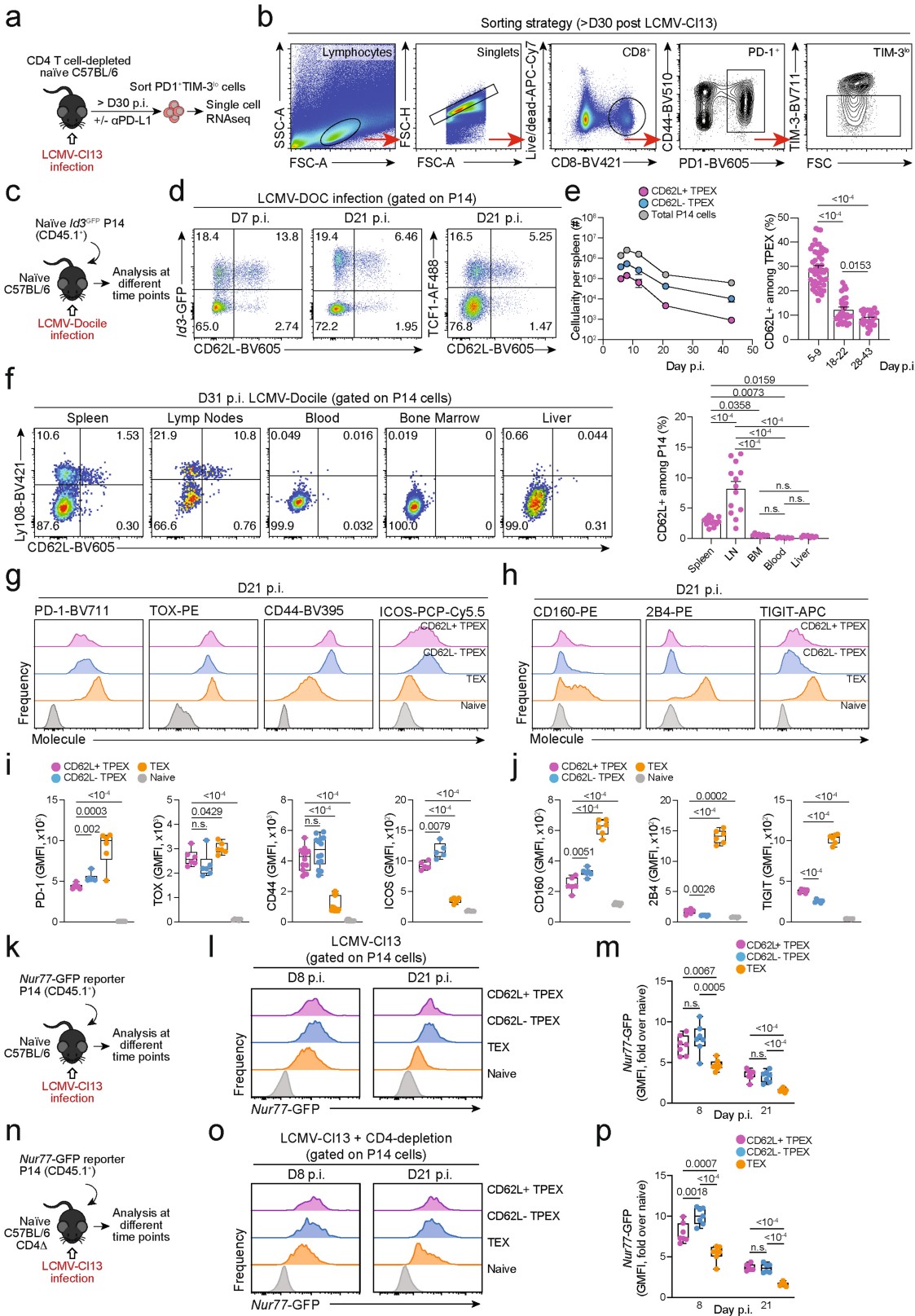

**Extended Data Fig. 1** | See next page for caption.

**Extended Data Fig. 1 | Isolation of polyclonal exhausted T cells for scRNA-seq and phenotypic characterization of CD62L⁺ T_PEX cells in chronic infection.** (**a**, **b**) CD4⁺ T cell-depleted naive mice were infected with LCMV-Cl13, treated with or without anti-PD-L1, and exhausted PD-1⁺TIM-3^lo T cells were sorted at >day 30 post-infection as described[19]. (**a**) Schematic of the experimental set-up. (**b**) Flow cytometry plots showing the sorting strategy. (**c**–**j**) Naive congenically marked (CD45.1⁺) *Id3*-GFP P14 cells were transferred to naive recipients (Ly.5.2), which were then infected with LCMV-Docile. Splenic P14 T cells were analysed at the indicated time points after infection. (**c**) Schematic of the experimental set-up. (**d**) Flow cytometry plots showing the expression of *Id3*-GFP, TCF1 and CD62L among splenic P14 T cells at 7 and 21 dpi. (**e**) Quantification showing absolute numbers of splenic CD62L⁺ T_PEX, CD62L⁻ T_PEX and total P14 cells (left) and frequencies of CD62L⁺ cells among T_PEX cells (right) at the indicated time points after infection (**f**) Flow cytometry plots showing the expression of Ly108 and CD62L and quantification of CD62L⁺ T_PEX cells among P14 T cells in the spleen, lymph nodes, blood, bone marrow and liver at day 31 post LCMV-Docile infection. (**g**–**j**) Histograms (**g**, **h**) and quantification (**i**, **j**) of expression of molecules as indicated in P14 T cell subsets and naive CD8⁺ T cells. (**k**–**p**) Congenically marked naive *Nur77*-GFP reporter P14 T cells were transferred into naive (**k**–**m**) or CD4-T-cell-depleted (**n**–**p**) recipient mice, which were subsequently infected with LCMV-Cl13. *Nur77*-GFP expression was analysed at indicated time points post-infection. (**k**, **n**) Schematics of the experimental set-up. Histograms (**l**, **o**) and quantifications (**m**, **p**) showing *Nur77*-GFP expression in the indicated P14 T cell subsets at 8 and 21 dpi. GMFI, geometric mean fluorescence intensity. Dots in graphs represent individual mice; box plots indicate range, interquartile and median; horizontal lines and error bars of bar graphs indicate mean and s.e.m. Data are representative of two independent experiments (**e**, **f**, **i**, **j**) and all analysed mice (**m**, **p**). *P* values are from two-tailed unpaired t-tests (**e**, **i**, **j**), two-way ANOVA (**f**), and one-way ANOVA (**m**, **p**); *P* > 0.05, not significant (n.s.).

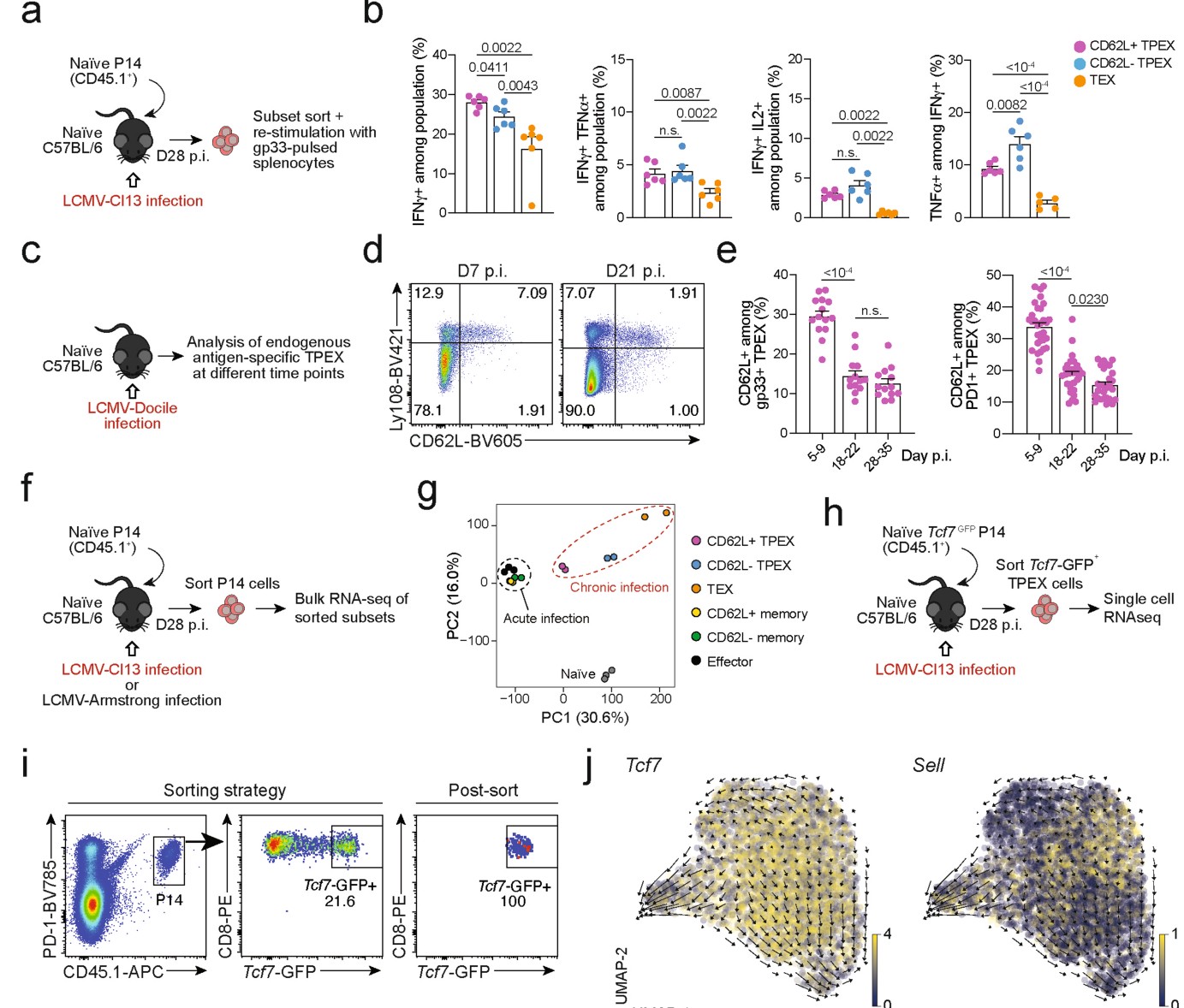

**Extended Data Fig. 2 | Functional and transcriptional profiling of exhausted T cell subsets and RNA velocity analysis showing that differentiation streams originate from CD62L⁺ T_PEX cells. (a, b)** Congenically marked naïve P14 T cells were adoptively transferred into naive recipient mice, which were then infected with LCMV-Cl13. Splenic P14 T cells from each group were sorted at day 28 post-infection and restimulated independently using gp33-pulsed splenocytes *in vitro*. (**a**) Schematic of the experimental set-up. (**b**) Quantifications showing cytokine production of each subset after restimulation. (**c**–**e**) Wild-type mice were infected with LCMV-Docile and splenic CD8⁺ T cells were analysed at the indicated time points after infection. (**c**) Schematic of the experimental set-up. (**d**) Flow cytometry plots showing the expression of CD62L in T_PEX (Ly108^hi) and T_EX (Ly108^lo) cells among endogenous gp33-specific CD8⁺ T cells. (**e**) Quantification showing the proportions of CD62L-expressing cells among gp33⁺ T_PEX cells (left) and polyclonal PD-1⁺ T_PEX cells (right) at the indicated time points after infection (**f**, **g**) Congenically

marked naive P14 T cells were adoptively transferred into naive recipient mice, which were then infected with LCMV-Cl13 or LCMV-Armstrong. Splenic P14 compartments from each group were sorted at 28 dpi and processed for bulk RNA-seq. (**f**) Schematic of the experimental set-up. (**g**) Principal component plot showing the transcriptional landscapes of sorted populations as indicated. (**h**–**j**) Congenically marked naive *Tcf7*-GFP P14 T cells were adoptively transferred into naive mice, which were then infected with LCMV-Cl13. P14 T_PEX cells were sorted at day 28 post-infection based on the expression of *Tcf7*-GFP. (**h**) Schematic of the experimental set-up. (**i**) Flow cytometry plots showing the sorting strategy and post-sort purity. (**j**) RNA velocity analysis showing developmental trajectories of T_PEX cells, together with the expression of *Tcf7* (left) and *Sell* (right). Horizontal lines and error bars of bar graphs indicate mean and s.e.m., respectively. Data are representative of two independent experiments (**b**) and all analysed mice (**e**). *P* values are from Mann–Whitney tests (**b**) and two-tailed unpaired *t*-tests (**e**); *P* > 0.05, not significant (n.s.).

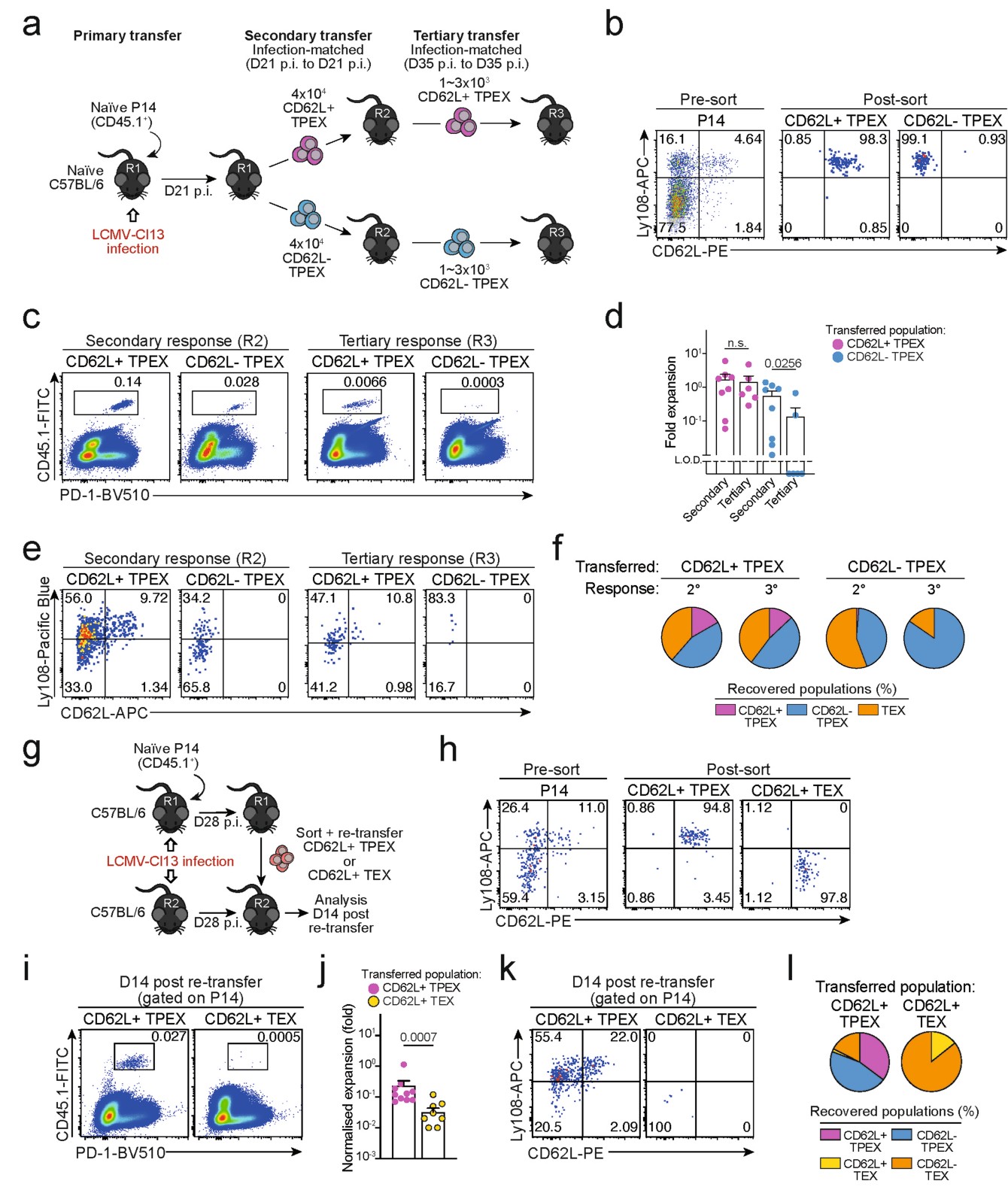

**Extended Data Fig. 3** | See next page for caption.

**Extended Data Fig. 3 | Capacity for self-renewal and multipotent differentiation is restricted to the CD62L⁺ T_PEX cell compartment.**
(**a**–**f**) Congenically marked naive P14 T cells were transferred into primary recipient mice (R1), which were then infected with LCMV-Cl13. The indicated subsets of P14 T cells were sorted at 21 dpi and $4×10^4$ cells were re-transferred to infection-matched secondary recipient mice (R2). The indicated subsets of P14 T cells were sorted from R2 mice at 35 dpi and $1-3×10^3$ cells were re-transferred to infection-matched tertiary recipient mice (R3). Splenic P14 T cells of R3 mice were analysed at day 14 post re-transfer. (**a**) Schematic of the experimental set-up. (**b**) Representative flow cytometry plots showing the sorting strategy and post-sort purities. Flow cytometry plots (**c**) and calculated fold expansion (**d**) of recovered P14 progenies at day 14 after secondary and tertiary re-transfers. Flow cytometry plots and quantifications showing expression of Ly108 and CD62L of splenic P14 cells in R2 and R3 mice (**e**) and average percentages of recovered CD62L⁺ T_PEX, CD62L⁻ T_PEX and T_EX cells per spleen in R2 and R3 mice (**f**) at day 14 post re-transfer, respectively. (**g**–**l**) Congenically marked naive P14 T cells were transferred into primary recipient mice (R1), which were then infected with LCMV-Cl13. The indicated subsets of P14 T cells were sorted at 28 dpi and $3-30 × 10^3$ cells were re-transferred to infection-matched secondary recipient mice (R2). Splenic P14 T cells of R2 mice were analysed at day 14 post re-transfer. Of note, maximum cell numbers attainable for each subset were transferred to allow for reliable evaluation of phenotypic diversification in expanded progenies. Fold expansion of recovered progenies was then normalized to distinct input numbers. (**g**) Schematic of the experimental set-up. (**h**) Representative flow cytometry plots showing the sorting strategy and post-sort purities. Flow cytometry plots (**i**) and fold expansion (**j**) of recovered progenies at day 14 post re-transfer. Flow cytometry plots and quantifications showing expression of Ly108 and CD62L of splenic P14 cells of R2 mice (**k**) and average percentages of recovered CD62L⁺ T_PEX, CD62L⁻ T_PEX, CD62L⁺ T_EX and CD62L⁻ T_EX cells per spleen (**l**) in R2 mice at day 14 post re-transfer. Dots in graphs represent individual mice; horizontal lines and error bars of bar graphs indicate mean and s.e.m., respectively. Data are representative of two independent experiments (**b**, **c**, **e**, **h**, **i**, **k**) and all analysed mice (**d**, **f**, **j**, **l**). P values are from Mann–Whitney tests (**d**, **j**); $P > 0.05$, not significant (n.s.).

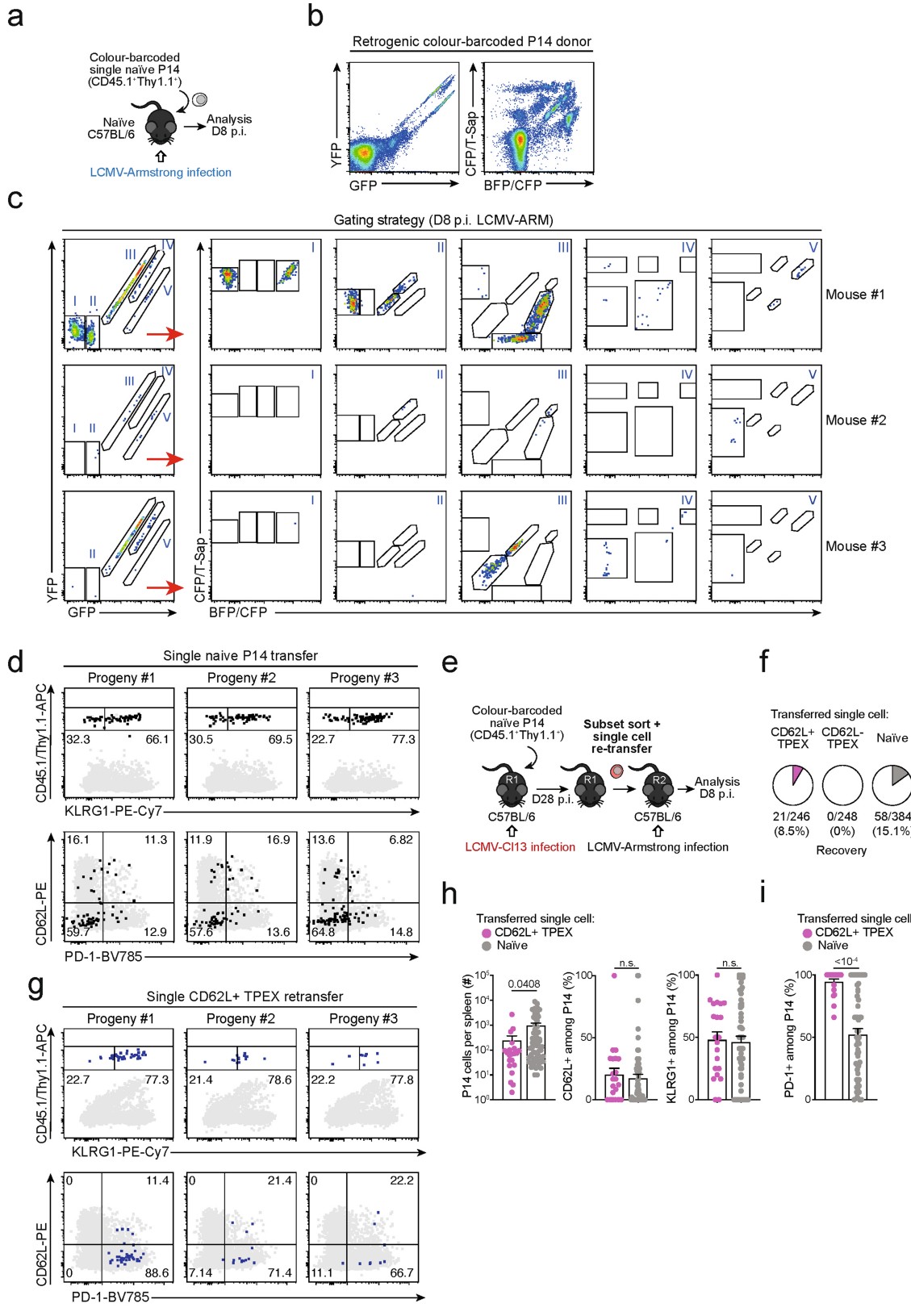

**Extended Data Fig. 4** | See next page for caption.

**Extended Data Fig. 4 | Single CD62L⁺ T_PEX cells show a stem-like capacity for self-renewal and multipotent differentiation.** (**a**–**d**) Single naive colour-barcoded P14 T cells were transferred to primary recipient mice, which were then infected with LCMV-Armstrong. Splenic P14 T cells were analysed at day 8 post LCMV-Armstrong infection. (**a**) Schematic of the experimental set-up for the naive P14 single-cell transfer. (**b**) Flow cytometry plots showing expression of GFP and YFP (left) or BFP/CFP and CFP/T-Sap (right) in peripheral blood of retrogenic P14 donor mice (pre-gated on CD8⁺CD44^loCD45.1⁺). (**c**) Tracking of colour-barcoded single-cell-derived progenies at 8 dpi in the spleens of three representative recipient mice. Recovered progenies were distinguished according to their combinatorial expression of GFP and YFP into populations I, II, III, IV and V, which were further subdivided by their expression of T-Sapphire, CFP, and BFP into progenies characterized by their unique combinatorial colour barcode. Note: in the display used, CFP emission appears on the diagonal between the BFP (x-axis) and T-Sapphire signal (y-axis) and is therefore indicated on both axes. (**d**) Flow cytometry plots depicting combined staining of CD45.1 and Thy1.1 with KLRG1 (upper row), or CD62L with PD-1 (lower row) for three progenies derived from adoptively transferred single naive P14 cells (grey: endogenous CD4⁻CD19⁻ cells). (**e**–**i**) Colour-barcoded naive P14 T cells were transferred into primary recipient mice (R1), which were subsequently infected with LCMV-Cl13. P14 T cells were sorted at 28 dpi and single CD62L⁺ or CD62L⁻ T_PEX cells were re-transferred into naive secondary recipient mice (R2), which were subsequently infected with LCMV-Armstrong. Splenic P14 T cells were analysed at day 8 post LCMV-Armstrong infection. (**e**) Schematic of the experimental set-up. (**f**) Percentages of transferred single cells of CD62L⁺ T_PEX, CD62L⁻ T_PEX cell or naive phenotype from which progenies were recovered at 8 dpi. (**g**) As in (**d**), but for adoptively re-transferred single CD62L⁺ T_PEX cells. (**h**) Size of single-T-cell-derived progenies and frequencies of CD62L⁺, KLRG1⁺ and (**i**) PD-1⁺ cells therein. Dots in graphs represent individual clones derived from a single transferred cell. Horizontal lines and error bars of bar graphs indicate mean and s.e.m., respectively. Data show all analysed mice (**h**, **i**). *P* values are from Mann–Whitney tests (**h**–**i**); *P* > 0.05, not significant (n.s.).

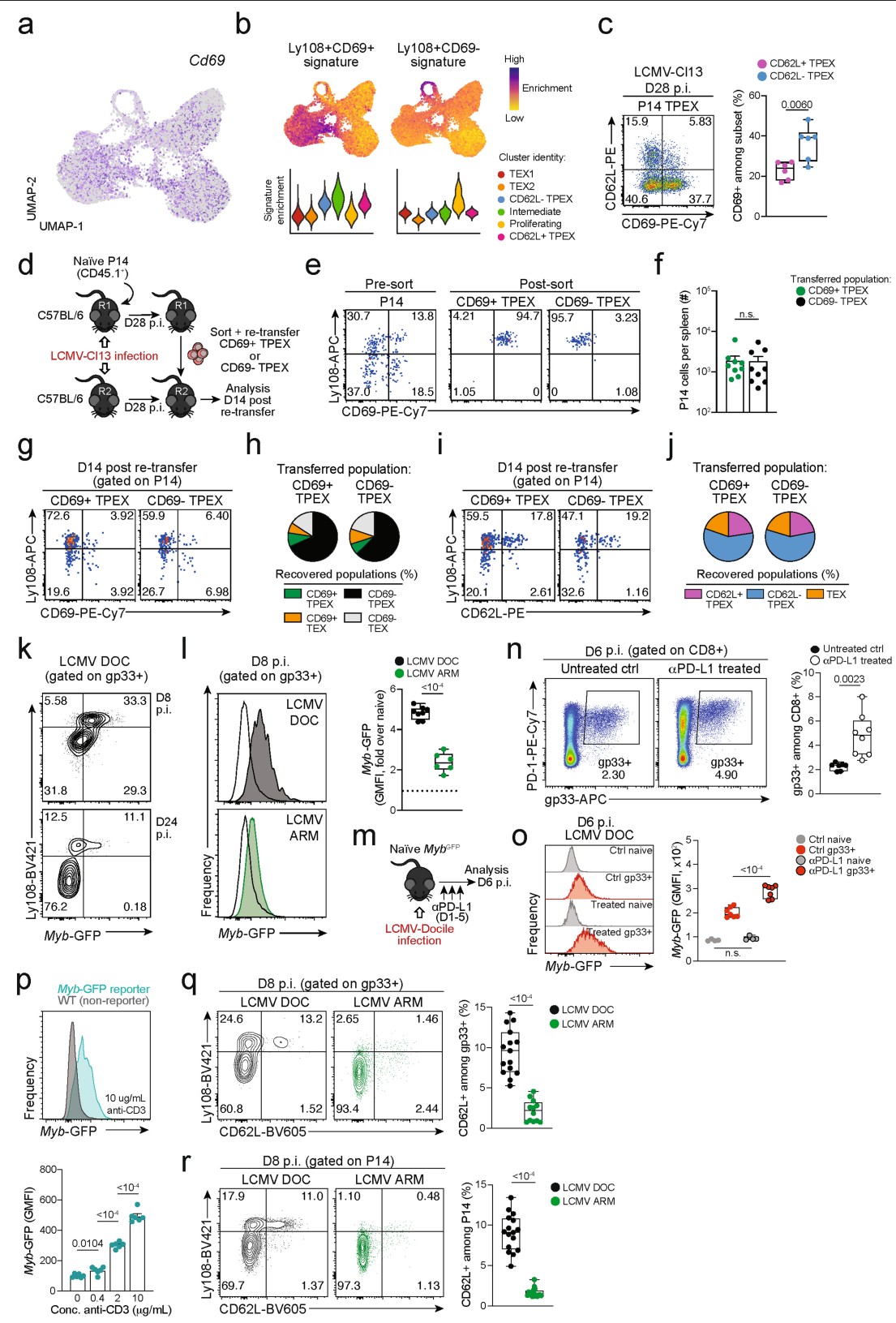

**Extended Data Fig. 5** | See next page for caption.

**Extended Data Fig. 5 | CD69 expression in T_PEX cells does not correlate with CD62L expression and does not predict developmental and repopulation potential; chronic LCMV infection and strong TCR stimuli favour MYB expression and the formation of stem-like CD62L⁺ T_PEX cells.** (**a**) Normalized expression of *Cd69* projected on the UMAP of scRNA-seq data as in Fig. 1. (**b**) Enrichment of Ly108⁺CD69⁺ ("T_EX prog1") and Ly108⁺CD69⁻ ("T_EX prog2") signatures[29] at single-cell and cluster levels. (**c**) Flow cytometry plots and quantification showing CD69 expression in CD62L⁺ and CD62L⁻ P14 T_PEX cells on day 28 post LCMV-Cl13 infection. (**d–j**) Congenically marked naive P14 T cells were transferred into primary recipient mice (R1), which were then infected with LCMV-Cl13. The indicated subsets of P14 T cells were sorted at 28 dpi and re-transferred to infection-matched secondary recipient mice (R2). Splenic P14 T cells of R2 mice were analysed at day 14 post re-transfer. (**d**) Schematic of the experimental set-up. (**e**) Representative flow cytometry plots showing the sorting strategy and post-sort purities. (**f**) Quantification of recovered P14 cells at day 14 post re-transfer. (**g**) Flow cytometry plots and quantifications showing expression of Ly108 and CD69 of splenic P14 cells of R2 mice and (**h**) average percentages of recovered CD69⁺ T_PEX, CD69⁻ T_PEX, CD69⁺ T_EX and CD69⁻ T_EX cells per spleen in R2 mice at day 14 post re-transfer. (**i**) Flow cytometry plots and quantifications showing expression of Ly108 and CD62L of splenic P14 cells of R2 mice and (**j**) average percentages of recovered CD62L⁺ T_PEX, CD62L⁻ T_PEX and T_EX cells per spleen in R2 mice at day 14 post re-transfer. (**k-l**) Naive *Myb^GFP* reporter mice were infected with either LCMV-Docile or LCMV-Armstrong and CD8⁺ T cells were analysed at the indicated time points after infection. (**k**) Representative flow cytometry plots depict Ly108 and *Myb*-GFP expression among antigen-specific (gp33⁺) CD8⁺ T cells. (**l**) Histograms (filled) show *Myb*-GFP expression of gp33⁺ CD8⁺ T cells in mice infected with LCMV-Docile (top)

and LCMV-Armstrong (bottom). Empty histograms depict *Myb*-GFP expression in naive CD8⁺ T cells in the same samples. Corresponding quantification show the fold change of geometric mean fluorescence intensity (GMFI) of *Myb*-GFP in the indicated populations. (**m–o**) LCMV-Docile-infected *Myb^GFP* reporter mice were treated with or without anti-PD-L1. Splenic CD8⁺ T cells were analysed at 6 dpi. (**m**) Schematic of the experimental set-up. (**n**) Flow cytometry plots and quantification showing frequencies of splenic gp33⁺ CD8⁺ T cells in anti-PD-L1-treated and untreated control mice at 6 dpi. (**o**) Histograms showing *Myb*-GFP expression of gp33⁺ and naive (gated on CD62L⁺CD44⁻) CD8⁺ T cells in the same mice. (**p**) Naive *Myb^GFP* and wild-type (non-reporter, control) CD8⁺ T cells were stimulated and cultured *in vitro* using plate-bound anti-CD3. Representative histogram and normalized quantification show GMFI of *Myb*-GFP expression in CD8⁺ T cells stimulated with plate-bound anti-CD3 at the indicated concentrations. (**q**) Flow cytometry plots and quantification show the frequencies of Ly108⁺ and CD62L⁺ cells among splenic antigen-specific (gp33⁺) T cells in wild-type mice at day 8 post LCMV-Armstrong infection. (**r**) Congenically marked P14 T cells were adoptively transferred into naive recipient mice, which were then infected with either LCMV-Docile or LCMV-Armstrong. Splenic P14 T cells were analysed at 8 dpi. Flow cytometry plots and quantification show the frequencies of Ly108⁺ and CD62L⁺ cells among splenic P14 T cells. Dots in graphs represent individual mice (**c**, **f**, **l**, **o**, **q**, **r**) and individual wells (**p**); box plots indicate range, interquartile and median; horizontal lines and error bars in indicate mean and s.e.m., respectively. Data are representative of two independent experiments (**c**, **l**, **o**, **p**) and all analysed mice (**f**, **q**, **r**). *P* values are from two-tailed unpaired *t*-tests (**c**, **l**, **o–r**) and Mann–Whitney tests (**f**); *P* > 0.05, not significant (n.s.).

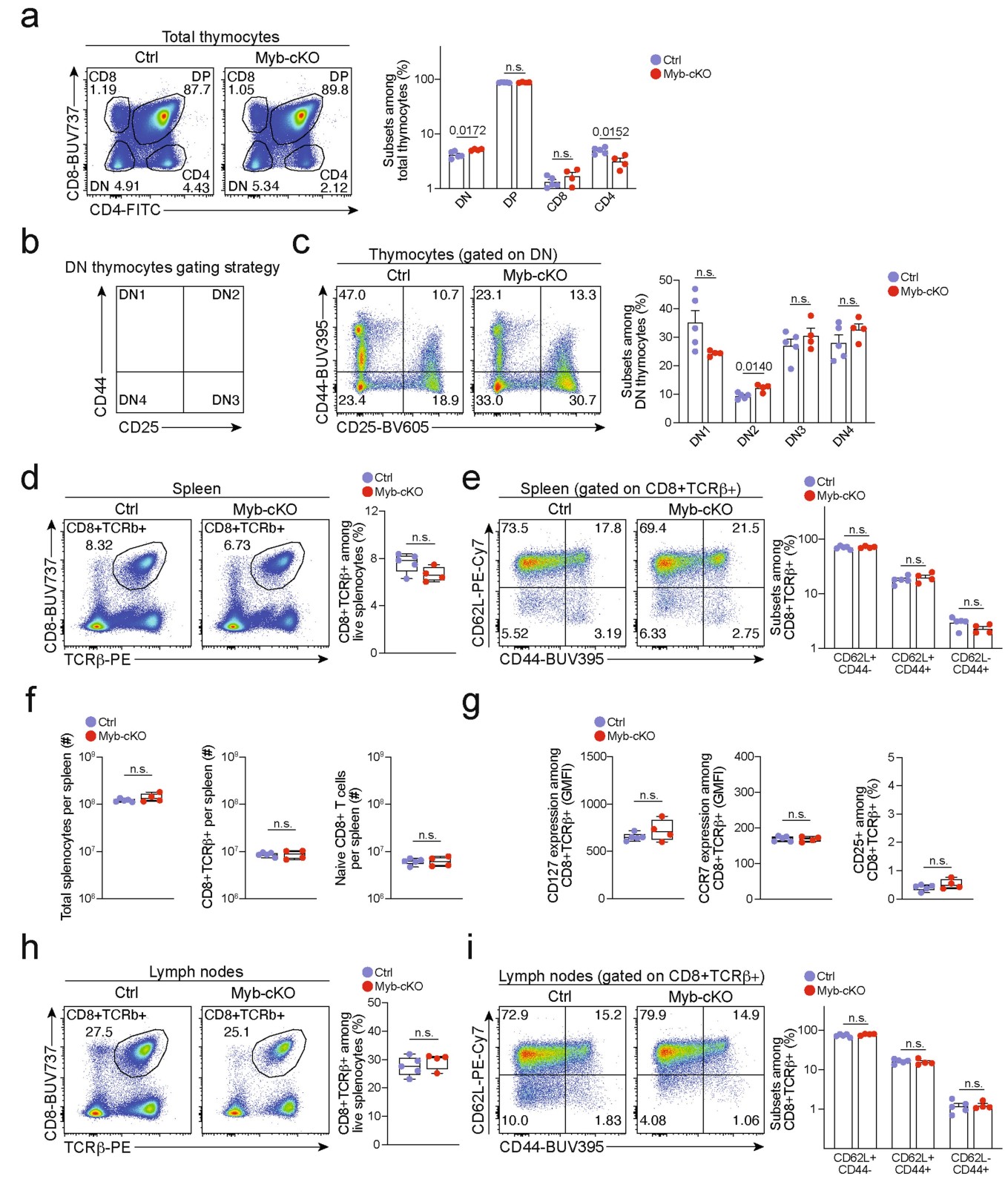

**Extended Data Fig. 6 | Development of mature CD8⁺ T cells is largely normal in *Myb^fl/fl^Cd4^Cre^* mice.** Adult 8-12 weeks *Myb^fl/fl^Cd4^Cre^* (*Myb*-cKO) and littermate *Myb^fl/fl^* control (Ctrl) mice were euthanized, and T cell populations were analysed in the thymus, spleen and lymph nodes. Flow cytometry and quantifications showing (**a–c**) frequencies of thymocyte subsets, (**d–f**) frequencies and abundance of mature splenic CD8⁺ T cells, (**g**) surface expression of CD127 (IL-7R), CCR7 and CD25 (IL-2R) of splenic CD8⁺ T cells and (**h, i**) frequencies of mature CD8⁺ T cells residing in lymph nodes. (**h**). Dots in graphs represent individual mice; box plots indicate range, interquartile and median. All data are representative of two independent experiments. *P* values are from two-tailed unpaired *t*-tests (**d, f–h**) and Mann–Whitney tests (**a, c, e, i**); *P* > 0.05, not significant (n.s.).

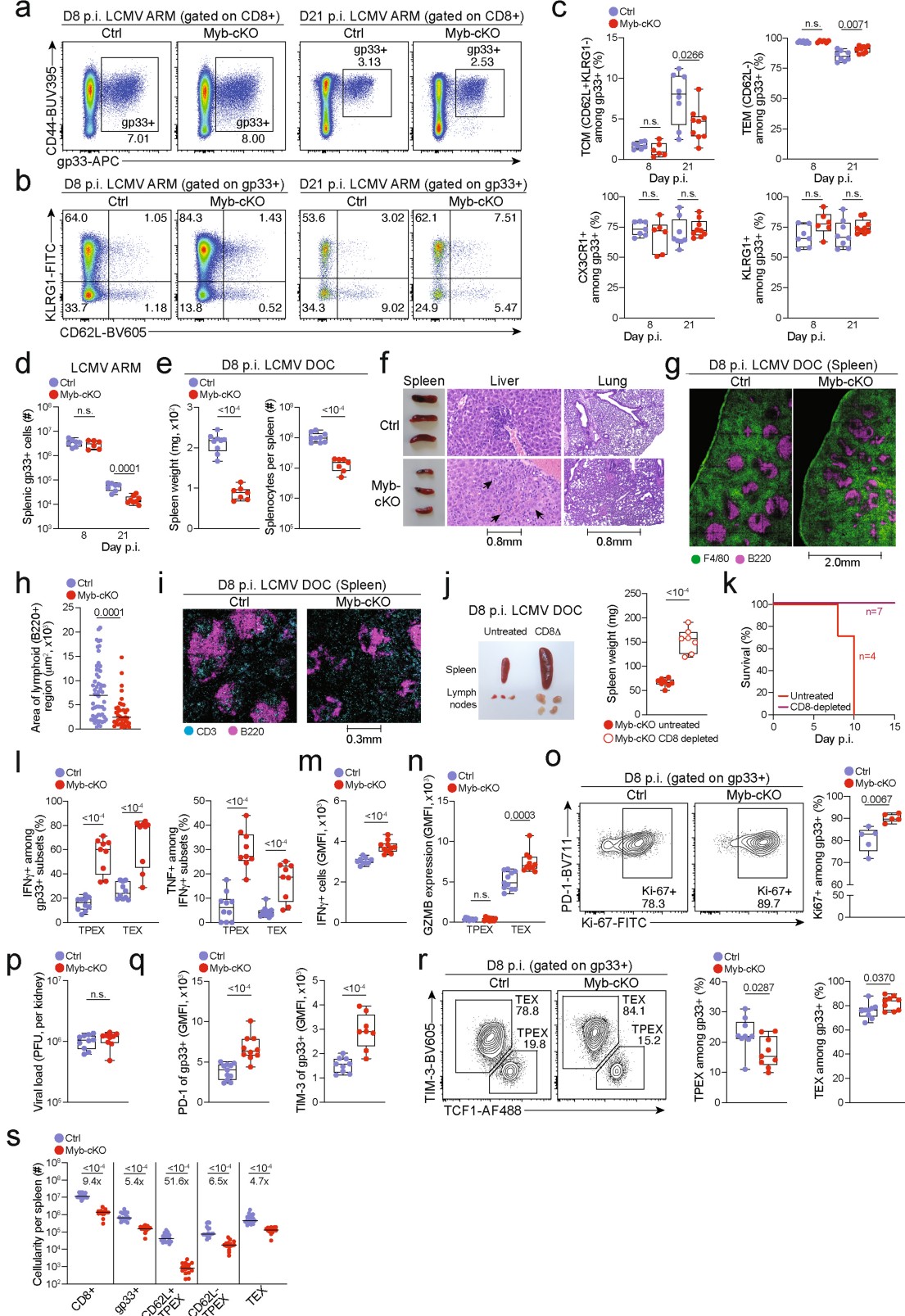

**Extended Data Fig. 7 |** See next page for caption.

**Extended Data Fig. 7 | MYB is required to limit CD8⁺ T cell expansion and cytotoxicity in response to chronic infection.** (a–s) $Myb^{fl/fl}Cd4^{Cre}$ ($Myb$-cKO) mice and littermate $Myb^{fl/fl}$ control mice (Ctrl) were infected with either LCMV-Armstrong (**a–d**) or LCMV-Docile (**e–s**). (**a–b**) Flow cytometry plots showing (**a**) splenic antigen-specific (gp33⁺) CD8⁺ cells and (**b**) expression of CD62L and KLRG1 among antigen-specific cells in $Myb$-cKO and control mice at indicated time points post LCMV-Armstrong infection. (**c**) Quantification of central memory ($T_{CM}$), effector memory ($T_{EM}$), CX3CR1⁺ and KLRG1⁺ cells among gp33⁺ CD8⁺ cells in $Myb$-cKO and control mice at indicated time points post LCMV-Armstrong infection. (**d**) Numbers of splenic gp33⁺CD8⁺ T cells in $Myb$-cKO and control mice at indicated time points post LCMV-Armstrong infection. (**e**) Box plots showing the weights of spleens (left) and the total numbers of splenocytes (right) in $Myb$-cKO and control mice at day 8 post LCMV-Docile infection. (**f**) Spleen size (left) and haematoxylin and eosin staining of sections showing infiltration of immune cells (arrows) in livers (middle) and lungs (right) in $Myb$-cKO and control mice at 8 dpi. (**g**) Confocal images of F4/80 and B220 expression in frozen spleen sections and (**h**) quantification showing the cellular organization and area of lymphoid regions in $Myb$-cKO and control mice at day 8 post LCMV-Docile infection. (**i**) Confocal images of CD3 and B220 expression in frozen spleen sections showing the distribution of B and T cells in the spleens of $Myb$-cKO and control mice at day 8 post LCMV-Docile infection. (**j**) Image and box plot showing the size and weights of spleens in untreated and CD8⁺ T-cell-depleted $Myb$-cKO mice at day 8 post LCMV-Docile infection. (**k**) Survival curve of CD8-depleted $Myb$-cKO mice post LCMV-Docile infection. (**l**) Proportion of cytokine-producing antigen-specific $T_{PEX}$ and $T_{EX}$ cell subsets after gp33 peptide restimulation of $Myb$-cKO and control mice at day 8 post LCMV-Docile infection. (**m, n**) Quantification of IFNγ expression (**m**), and granzyme B (GZMB) expression in $T_{PEX}$ and $T_{EX}$ cells (**n**) in $Myb$-cKO and control mice at day 8 post LCMV-Docile infection. (**o**) Flow cytometry plots and quantification showing the proportions of Ki67⁺ within the gp33⁺ compartment in $Myb$-cKO and control mice at day 8 post LCMV-Docile infection. (**p**) Box plots showing viral titres in the kidneys of $Myb$-cKO and control mice at day 8 post LCMV-Docile infection. (**q**) Box plots showing the expression of PD-1 (left) and TIM-3 (right) among gp33⁺ CD8⁺ T cells of control and $Myb$-cKO mice at day 8 post LCMV-Docile infection. (**r**) Flow cytometry plots and quantification show the frequencies of $T_{PEX}$ cells (TCF1⁺TIM-3⁻) and $T_{EX}$ cells (TCF1TIM-3⁺) among splenic gp33⁺ CD8⁺ T cells of $Myb$-cKO and control mice. (**s**) Quantification showing the absolute numbers of splenic CD8⁺, gp33⁺, CD62L⁺ $T_{PEX}$, CD62L⁻ $T_{PEX}$ and $T_{EX}$ cells in $Myb$-cKO and control mice at day 8 post LCMV-Docile infection. GMFI, geometric mean fluorescence intensity. Dots in graphs represent individual mice; box plots indicate range, interquartile and median; horizontal lines in (**h**) indicate median. Data are representative of two independent experiments (**c–e**, **k–r**) or all mice (**j, s**) and images (**h**) analysed; $P > 0.05$, not significant (n.s.).

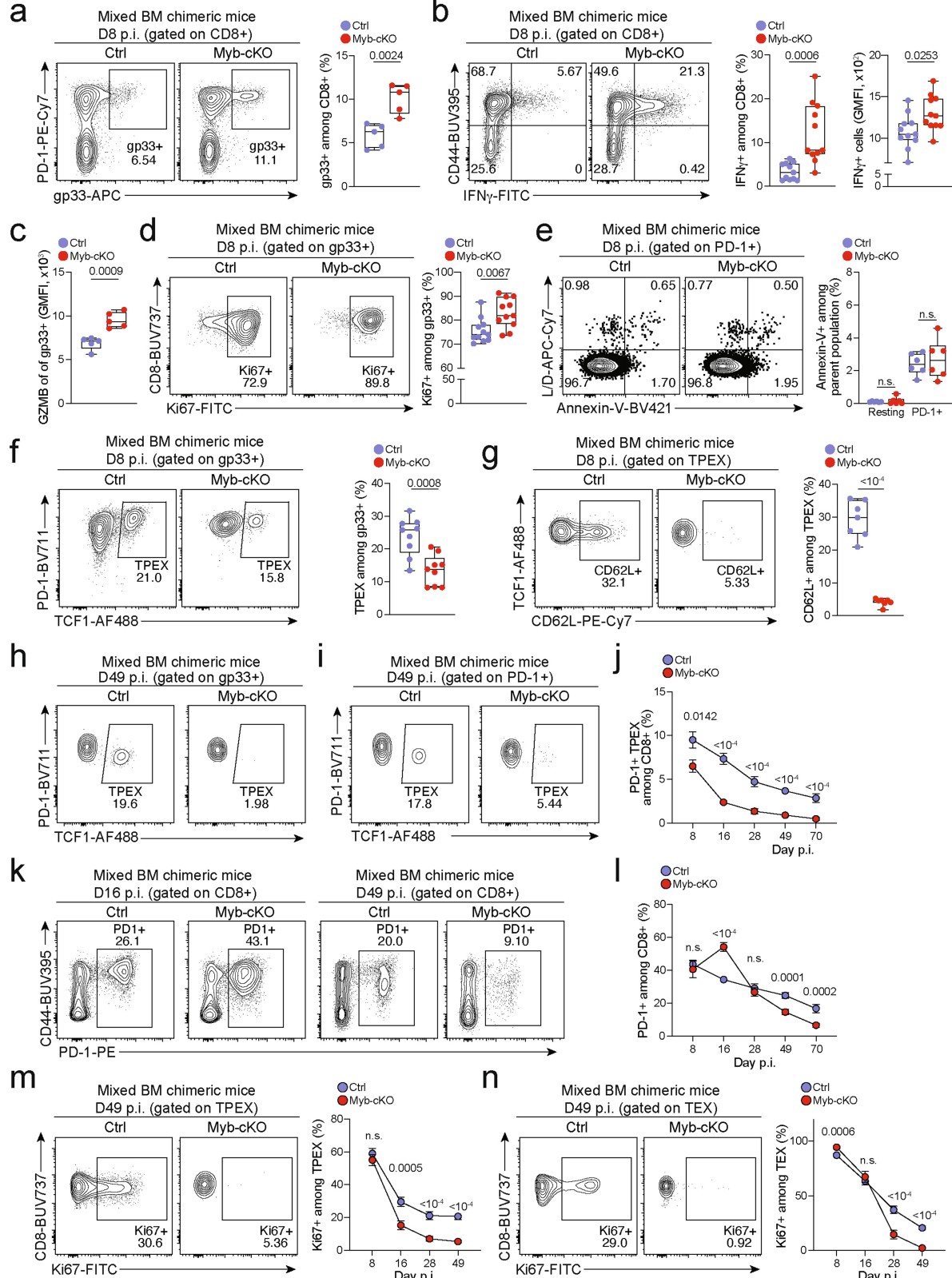

**Extended Data Fig. 8 |** See next page for caption.

**Extended Data Fig. 8 | MYB limits CD8⁺ T cell expansion and cytotoxicity during chronic infection in a cell-intrinsic manner.** (**a–n**) Naive CD45.1 mice were lethally irradiated and reconstituted using a mixture of $Myb^{fl/fl}Cd4^{Cre}$ ($Myb$-cKO) and $Cd4^{Cre}$ or littermate $Myb^{fl/fl}$ control (Ctrl) bone marrow. Chimeric mice were subsequently infected with LCMV-Docile and analysed at the indicated time points after infection. Quantification showing the frequencies of (**a**) polyclonal antigen-specific gp33⁺ cells among $Myb$-cKO and control CD8⁺ T cells at 8 dpi. (**b**) Flow cytometry plots and quantification of IFNγ⁺ cells among $Myb$-cKO and control CD8⁺ T cells after peptide restimulation *in vitro* at 8 dpi. (**c**) Quantification of GZMB expression among gp33⁺ cells of the indicated genotypes. (**d, e**) Flow cytometry plots and quantification showing the frequencies of (**d**) Ki67⁺ cells and (**e**) annexin-V⁺ cells among $Myb$-cKO and control antigen-responsive CD8⁺ T cells. (**f, g**) Flow cytometry plots and quantification showing the frequencies of TCF1⁺ T$_{PEX}$ cells among antigen-specific T cells (**f**) and CD62L⁺ cells among T$_{PEX}$ cells (**g**) in $Myb$-cKO and control compartments at 8 dpi. (**h**) Flow cytometry plots showing the frequencies of T$_{PEX}$ cells among gp33⁺ cells at 49 dpi. (**i–j**) Flow cytometry plots (**i**) and quantification (**j**) showing kinetics of splenic polyclonal PD-1⁺ T$_{PEX}$ cells among $Myb$-cKO and control CD8⁺ T cells after infection. (**k–l**) Flow cytometry plots (**k**) and quantification (**l**) showing the frequencies of the entire antigen-responsive PD-1⁺ cell compartment among $Myb$-cKO and control CD8⁺ T cells at indicated time points after infection. (**m, n**) Flow cytometry plots and quantifications showing the frequencies of Ki67⁺ cells among $Myb$-cKO and control polyclonal T$_{PEX}$ (**m**) and T$_{EX}$ (**n**) cells at indicated time points after infection. GMFI, geometric mean fluorescence intensity. Dots in graphs represent individual mice; box plots indicate range, interquartile and median. Symbols and error bars represent mean and s.e.m., respectively. All data are representative of two independent experiments. $P$ values are from two-tailed unpaired $t$-tests (**a–g**) and Mann–Whitney tests (**j–n**); $P > 0.05$, not significant (n.s.).

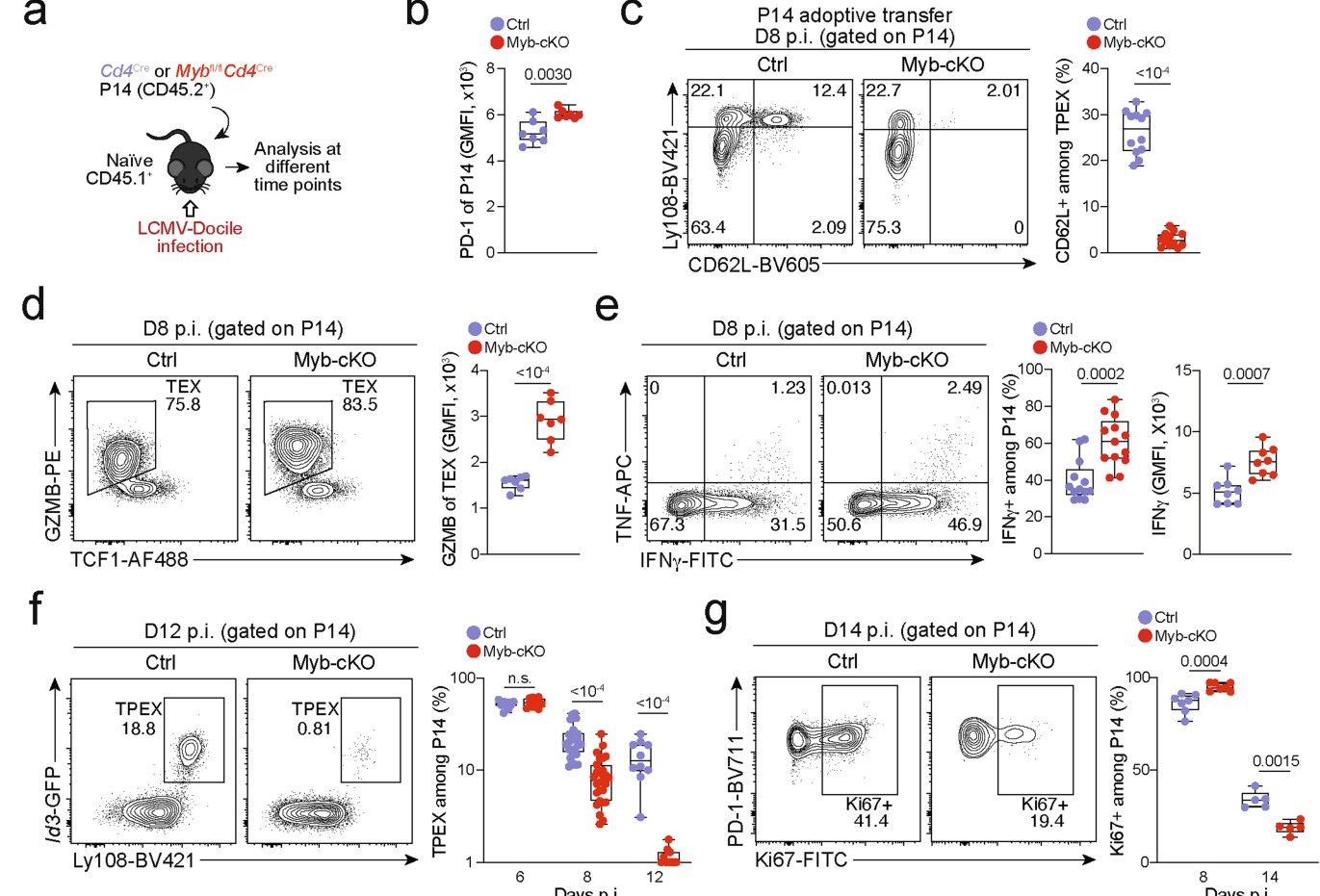

**Extended Data Fig. 9 | MYB limits proliferation and cytotoxicity and sustains the long-term self-renewal of exhausted CD8⁺ T cells.**
(**a–g**) Congenically marked naive control (*Cd4^Cre*) and *Myb^fl/fl^Cd4^Cre^* (*Myb*-cKO) P14 T cells were adoptively transferred into naive recipient mice, which were subsequently infected with LCMV-Docile. Splenic P14 T cells were analysed at indicated time points post-infection (p.i.). (**a**) Schematic of the experimental set-up. (**b**) Box plot showing PD-1 expression of transferred P14 cells at 8 dpi. (**c**) Flow cytometry plots and quantification showing frequencies of CD62L⁺ cells among *Myb*-cKO and control P14 T$_{PEX}$ cells. (**d**) Flow cytometry plots and quantification showing the expression of granzyme B (GZMB) in *Myb*-cKO and control T$_{EX}$ P14 cells at 8 dpi. (**e**) Flow cytometry plots and quantifications showing the production of cytokines as indicated from *Myb*-cKO and control P14 T cells after gp33 peptide restimulation at 8 dpi. (**f**) Flow cytometry plots and quantification showing the frequencies of T$_{PEX}$ cells among *Myb*-cKO and control P14 T cells at the indicated time points after infection. (**g**) Flow cytometry plots and quantification showing the frequencies of Ki67⁺ cells among *Myb*-cKO and control P14 T cells at indicated time points after infection. GMFI, geometric mean fluorescence intensity. Dots in graphs represent individual mice; box plots indicate range, interquartile and median; Data are representative of two independent experiments (**b–g**). *P* values are from two-tailed unpaired *t*-tests; *P* > 0.05, not significant (n.s.).

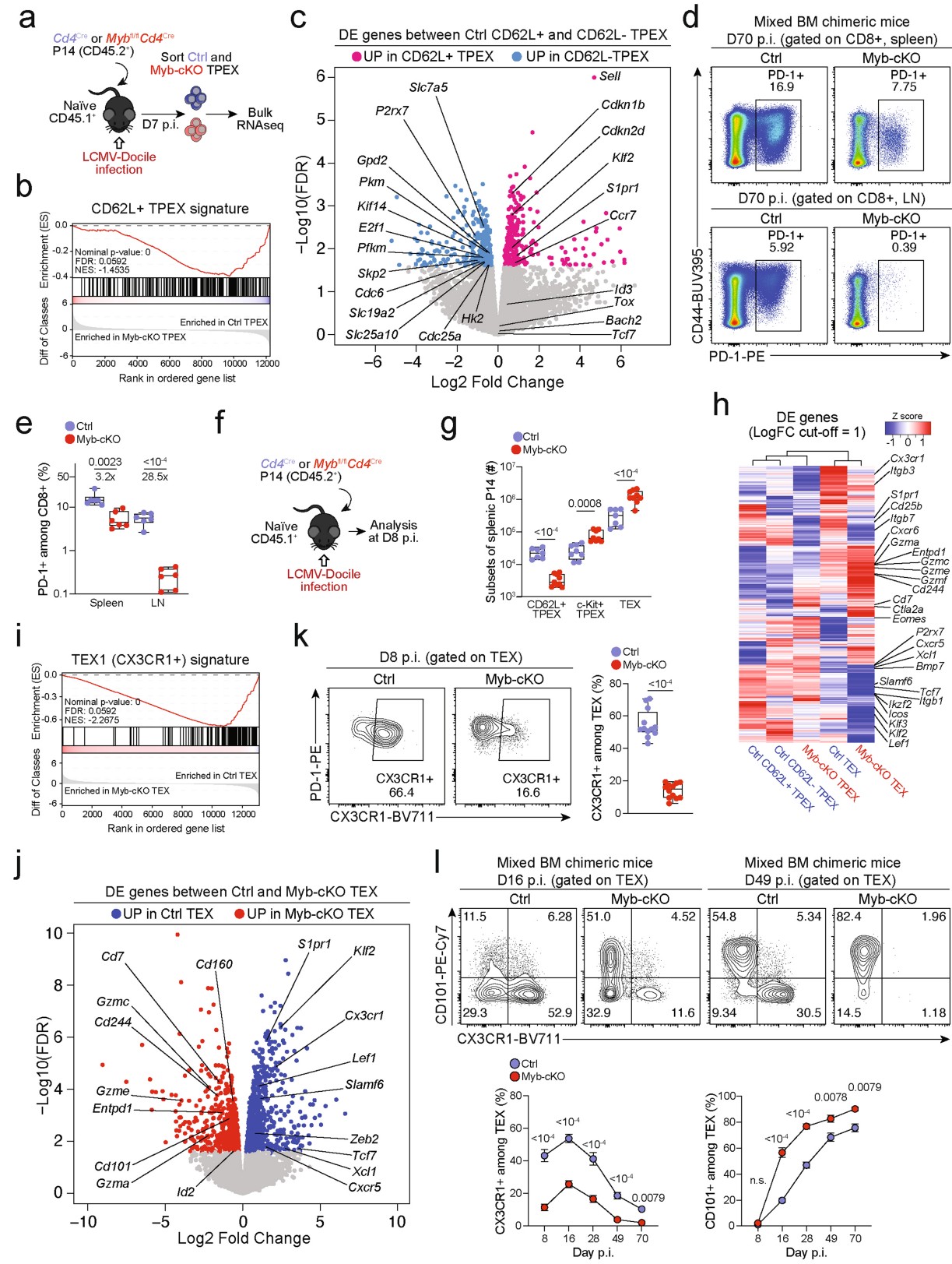

**Extended Data Fig. 10** | See next page for caption.

**Extended Data Fig. 10 | MYB regulates the expression of multiple genes that are critical to exhausted T cell function and maintenance.** (**a**, **b**) Congenically marked *Myb*$^{fl/fl}$*Cd4*$^{Cre}$ (*Myb*-cKO, CD45.2$^+$) and *Cd4*$^{Cre}$ (Ctrl, CD45.2$^+$ or CD45.2$^+$ CD45.1$^+$) P14 T cells were adoptively transferred into separate naive recipient (CD45.1) mice, which were then infected with LCMV-Docile. Splenic P14 T$_{PEX}$ cells were sorted at day 7 post-infection and processed for bulk RNA-seq. (**a**) Schematic of the experimental set-up. (**b**) Gene set enrichment analysis showing loss of CD62L$^+$ T$_{PEX}$ transcriptional signature in *Myb*-cKO T$_{PEX}$ cells compared to control T$_{PEX}$ cells. (**c**) Volcano plots highlighting genes differentially expressed (FDR < 0.15) between control CD62L$^+$ T$_{PEX}$ and CD62L$^-$ T$_{PEX}$ cells. (**d**–**e**) Mixed bone marrow chimeric mice containing congenically marked *Myb*-cKO and control CD8$^+$ T cells were infected with LCMV-Docile. Flow cytometry plots (**d**) and quantification (**e**) showing the frequencies of the entire antigen-responsive PD-1$^+$ cell compartment among *Myb*-cKO and control CD8$^+$ T cells in the spleen and lymph nodes at day 70 post-infection. (**f**, **g**) Congenically marked *Myb*-cKO and Ctrl P14 T cells were adoptively transferred into separate naive recipient mice, which were then infected with LCMV-Docile. Splenic P14 T$_{PEX}$ cells were analysed at day 8 post-infection. (**f**) Schematic of the experimental set-up. (**g**) Quantification showing the abundances of the indicated P14 subsets per spleen. (**h**) Heat map depicting genes differentially expressed (FDR < 0.15, FC > 1) between control CD62L$^+$ T$_{PEX}$ and CD62L$^-$ T$_{PEX}$ cell or *Myb*-cKO and control T$_{PEX}$ and T$_{EX}$ cells, with genes of interest annotated. (**i**) Gene set enrichment analysis showing loss of CX3CR1$^+$ T$_{EX}$ transcriptional signature in P14 *Myb*-cKO T$_{EX}$ cells compared to control T$_{EX}$ cells. (**j**) Volcano plot highlighting genes differentially expressed (FDR < 0.15) between control and *Myb*-cKO T$_{EX}$ cells with genes of interested annotated. (**k**) Flow cytometry plots and quantification show the frequencies of CX3CR1$^+$ cells among control and *Myb*-cKO T$_{EX}$ P14 T cells at day 8 post LCMV-Docile infection. (**l**) Flow cytometry plots and quantifications showing CX3CR1 and CD101 expression in *Myb*-cKO and control T$_{EX}$ cells at the indicated time points after infection. Dots in graph represent individual mice; box plots indicate range, interquartile and median. Symbols and error bars in (**l**) represent mean and s.e.m., respectively Data are representative of two independent experiments (**e**, **g**, **k**, **l**). *P* values are from two-tailed unpaired *t*-tests; *P* > 0.05, not significant (n.s.).

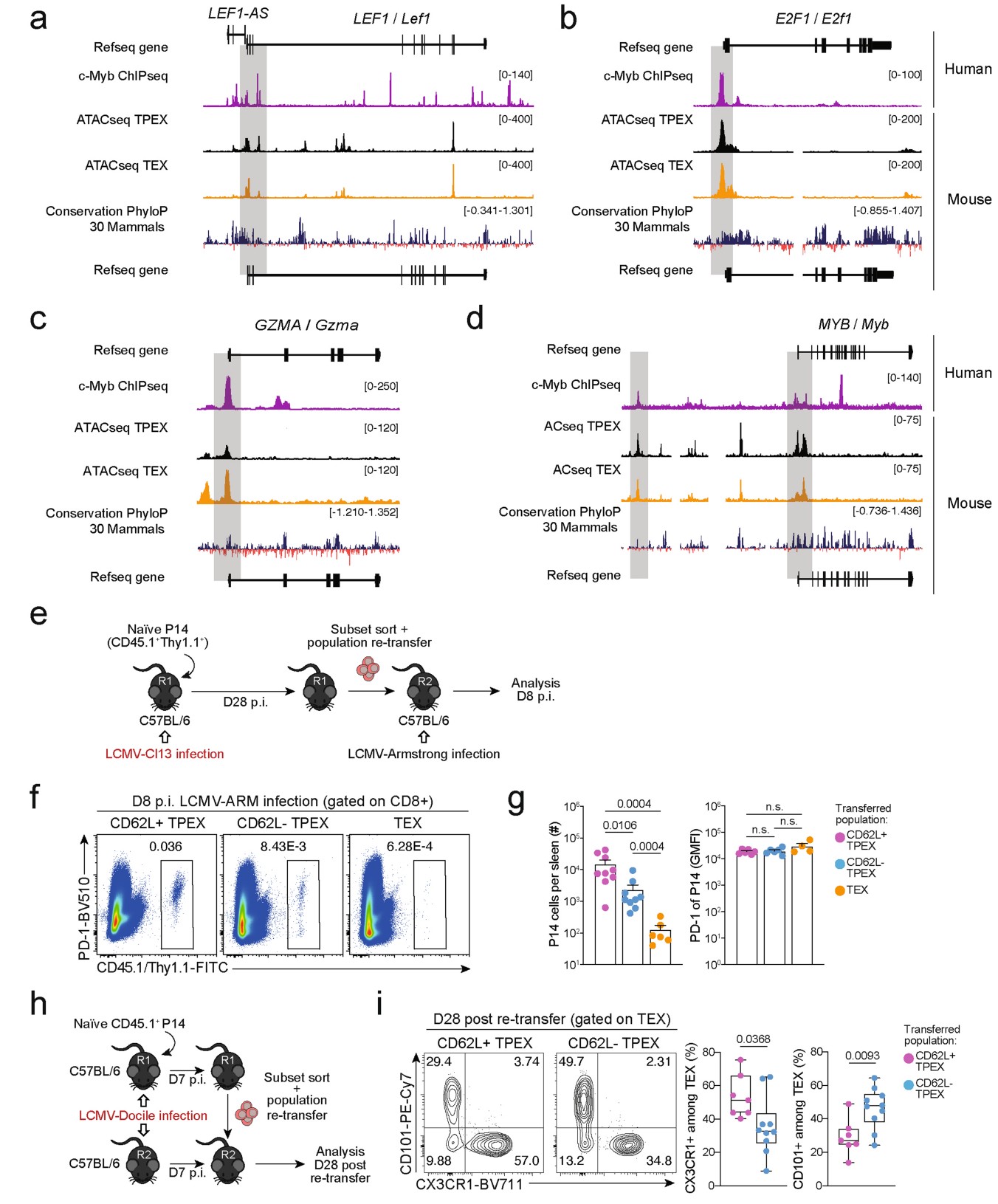

**Extended Data Fig. 11** | See next page for caption.

**Extended Data Fig. 11 | MYB directly regulates target gene expression, and CD62L⁺ T_PEX cells have a superior potential to give rise to CX3CR1⁺ T_EX cells.** (**a**–**d**) Representative tracks showing MYB ChIP–seq peaks in the *LEF1* (**a**), *E2F1* (**b**), *GZMA* (**c**), and *MYB* (**d**) gene loci of human Jurkat T cells and ATAC-seq peaks of $T_{PEX}$ and $T_{EX}$ cells in the corresponding mouse gene loci aligned according to the sequence conservation. (**e**–**g**) Congenically marked naive P14 cells were transferred to primary recipient mice (R1), which were subsequently infected with LCMV-Cl13. The indicated subsets of P14 cells were sorted at 28 dpi and re-transferred to naive secondary recipient mice (R2), which were then infected with LCMV-Armstrong. Splenic P14 T cells in R2 mice were analysed at 8 dpi. (**e**) Schematic of the experimental set-up. (**f**) Flow cytometry plots of progenies recovered at 8 dpi. (**g**) Cell numbers (left) and quantification of PD-1 expression (right) in P14 T cell populations derived from the indicated transferred subsets at 8 dpi. (**h**, **i**) Congenically marked naive P14 T cells were transferred into primary recipient mice (R1), which were then infected with LCMV-Docile. The indicated subsets of P14 T cells were sorted at 7 dpi and $7.5 \times 10^4$ cells were re-transferred to infection-matched (LCMV-Docile) secondary recipient mice (R2). Splenic P14 T cells of R2 mice were analysed at day 28 post re-transfer. (**h**) Schematic of the experimental set-up. (**i**) Flow cytometry plots and box plots showing the frequencies of CX3CR1⁺ and CD101⁺ cells among recovered $T_{EX}$ cells derived from the indicated re-transferred $T_{PEX}$ subsets at day 28 post re-transfer. Data are representative of two independent experiments. *P* values are from Mann–Whitney tests (**g**) and two-tailed unpaired *t*-test (**i**); *P* > 0.05, not significant (n.s.).

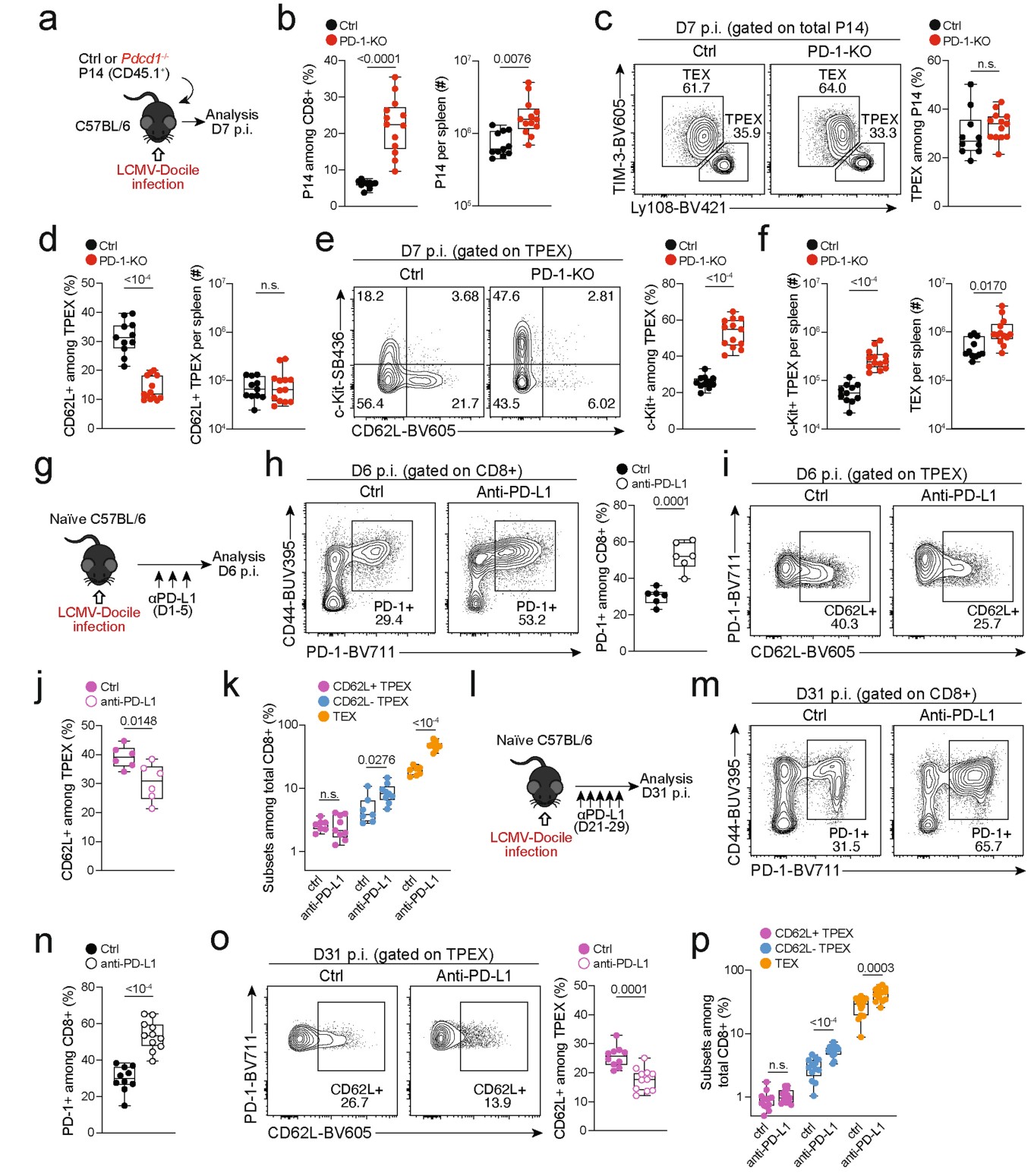

**Extended Data Fig. 12** | See next page for caption.

**Extended Data Fig. 12 | Effect of PD-1 signalling on CD62L$^+$ T$_{PEX}$ cells.**
(**a**–**f**) Congenically marked PD-1-deficient (*Pdcd1$^{-/-}$*) and control P14 T cells were transferred to naive mice, which were subsequently infected with LCMV-Docile. Splenic P14 T cells were analysed at 7 dpi. (**a**) Schematic of the experimental set-up. (**b**) P14 T cell frequencies and numbers of indicated genotypes. (**c**) Flow cytometry plots and frequencies of T$_{PEX}$ (Ly108$^{hi}$TIM-3$^{lo}$) and T$_{EX}$ (Ly108$^{lo}$TIM-3$^{hi}$) cells. (**d**) Box plots show frequencies and numbers of CD62L$^+$ T$_{PEX}$ cells among control and PD-1-deficient P14 cells. Flow cytometry plots and box plots show (**e**) frequencies of KIT$^+$ T$_{PEX}$ cells and (**f**) numbers of KIT$^+$ T$_{PEX}$ and T$_{EX}$ cells per spleen. (**g**–**k**) Wild-type mice were infected with LCMV-Docile and treated with anti-PD-L1 at 200 μg/mouse at 1, 3 and 5 dpi. Splenic CD8$^+$ T cells were analysed at 6 dpi. (**g**) Schematic of the experimental set-up. (**h**) Flow cytometry plots and quantification showing the frequencies of PD-1$^+$ cells among splenic CD8$^+$ T cells. (**i**–**j**) Flow cytometry plots (**i**) and quantification (**j**) showing expression of CD62L among polyclonal T$_{PEX}$ cells (Ly108$^{hi}$TIM-3$^{lo}$). (**k**) Quantification showing the population sizes of CD62L$^+$ T$_{PEX}$, CD62L$^-$ T$_{PEX}$ and T$_{EX}$ cells among total CD8$^+$ T cells in untreated and anti-PD-L1-treated mice. (**l**–**p**) Wild-type mice were infected with LCMV-Docile and treated with anti-PD-L1 at 200 μg/mouse at 21, 23, 25, 27 and 29 dpi. Splenic CD8$^+$ T cells were analysed at 31 dpi. (**l**) Schematic of the experimental set-up. (**m**–**n**) Flow cytometry plots (**m**) and quantification (**n**) showing the frequencies of the PD-1$^+$ cells among splenic CD8$^+$ T cells. (**o**) Flow cytometry plots and quantification showing the expression of CD62L among polyclonal T$_{PEX}$ cells (Ly108$^{hi}$TIM-3$^{lo}$). (**p**) Quantification showing the population sizes of CD62L$^+$ T$_{PEX}$, CD62L$^-$ T$_{PEX}$ and T$_{EX}$ cells among total CD8$^+$ T cells in untreated and anti-PD-L1-treated mice. Dots in graphs represent individual mice; box plots indicate range, interquartile and median. Data are representative of at least two independent experiments. *P* values are from two-tailed unpaired *t*-tests; *P* > 0.05, not significant (n.s.).

Veit R. Buchholz

# Reporting Summary

## Statistics

For all statistical analyses, confirm that the following items are present in the figure legend, table legend, main text, or Methods section.

| n/a | Confirmed | |
|-----|-----------|---|
| ☐ | ☒ | The exact sample size ($n$) for each experimental group/condition, given as a discrete number and unit of measurement |
| ☐ | ☒ | A statement on whether measurements were taken from distinct samples or whether the same sample was measured repeatedly |
| ☐ | ☒ | The statistical test(s) used AND whether they are one- or two-sided<br>*Only common tests should be described solely by name; describe more complex techniques in the Methods section.* |
| ☐ | ☒ | A description of all covariates tested |
| ☐ | ☒ | A description of any assumptions or corrections, such as tests of normality and adjustment for multiple comparisons |
| ☐ | ☒ | A full description of the statistical parameters including central tendency (e.g. means) or other basic estimates (e.g. regression coefficient) AND variation (e.g. standard deviation) or associated estimates of uncertainty (e.g. confidence intervals) |
| ☒ | ☐ | For null hypothesis testing, the test statistic (e.g. $F$, $t$, $r$) with confidence intervals, effect sizes, degrees of freedom and $P$ value noted<br>*Give P values as exact values whenever suitable.* |
| ☒ | ☐ | For Bayesian analysis, information on the choice of priors and Markov chain Monte Carlo settings |
| ☒ | ☐ | For hierarchical and complex designs, identification of the appropriate level for tests and full reporting of outcomes |
| ☒ | ☐ | Estimates of effect sizes (e.g. Cohen's $d$, Pearson's $r$), indicating how they were calculated |

*Our web collection on statistics for biologists contains articles on many of the points above.*

## Software and code

Policy information about availability of computer code

| | |
|---|---|
| Data collection | Flow Cytometry:<br>CytExpert (Beckman Coulter) v2.3.1.22 |
| Data analysis | Flow Cytometry:<br>FlowJo v10.6.1<br><br>Data analysis and visualization:<br>Prism v7<br><br>Visualization:<br>Adobe Illustrator v24.1.2<br><br>Annotation of scRNA-seq data:<br>CellRanger v5.0.0<br><br>scRNA-seq analysis:<br>SCANPY v1.6<br>Velocyto 0.17.17<br>scVelo v0.2.1 |

For manuscripts utilizing custom algorithms or software that are central to the research but not yet described in published literature, software must be made available to editors and reviewers. We strongly encourage code deposition in a community repository (e.g. GitHub). See the Nature Portfolio guidelines for submitting code & software for further information.

# Data

Policy information about availability of data

All manuscripts must include a data availability statement. This statement should provide the following information, where applicable:
- Accession codes, unique identifiers, or web links for publicly available datasets
- A description of any restrictions on data availability
- For clinical datasets or third party data, please ensure that the statement adheres to our policy

Mathematical codes are available upon request.

# Field-specific reporting

Please select the one below that is the best fit for your research. If you are not sure, read the appropriate sections before making your selection.

☒ Life sciences  ☐ Behavioural & social sciences  ☐ Ecological, evolutionary & environmental sciences

For a reference copy of the document with all sections, see nature.com/documents/nr-reporting-summary-flat.pdf

# Life sciences study design

All studies must disclose on these points even when the disclosure is negative.

| | |
|---|---|
| Sample size | No sample-size calculation was performed, the data presented in this study was collected in repeated independent experiments with at least 2-3 mice per group, as indicated in the Figure Legends. |
| Data exclusions | No data were excluded from the analysis. |
| Replication | The presented data was successfully replicated, in some cases experiments were replicated by different authors. |
| Randomization | Age- and sex-matched mice were allocated to groups based on the experimental treatment (no randomization). |
| Blinding | Blinding was not performed, as data analysis was strictly quantitative and not subjective. Computational analysis was not blinded. |

# Reporting for specific materials, systems and methods

We require information from authors about some types of materials, experimental systems and methods used in many studies. Here, indicate whether each material, system or method listed is relevant to your study. If you are not sure if a list item applies to your research, read the appropriate section before selecting a response.

## Materials & experimental systems

| n/a | Involved in the study |
|---|---|
| ☐ | ☒ Antibodies |
| ☒ | ☐ Eukaryotic cell lines |
| ☒ | ☐ Palaeontology and archaeology |
| ☐ | ☒ Animals and other organisms |
| ☒ | ☐ Human research participants |
| ☒ | ☐ Clinical data |
| ☒ | ☐ Dual use research of concern |

## Methods

| n/a | Involved in the study |
|---|---|
| ☒ | ☐ ChIP-seq |
| ☐ | ☒ Flow cytometry |
| ☒ | ☐ MRI-based neuroimaging |

# Antibodies

| | |
|---|---|
| Antibodies used | Name, Clone, Supplier:<br>CD45.1, A20, Biolegend or BD Bioscience<br>CD45.2, 104, BD Bioscience<br>CD90.1, HIS51, Thermo Fisher Scientific<br>CD4, RM4-5, Biolegend<br>CD4, GK1.5, Biolegend<br>CD44, IM7, BD Bioscience<br>CD19, 6D5, Biolegend<br>CD117, ACK2, Thermo Fisher Scientific<br>CD101, Moushi101, Thermo Fisher Scientific<br>CD160, eBioCNX46-3, eBioscience<br>CD244, eBio244F4, Thermo Fisher Scientific |

CD90.2, 30-H12, Biolegend
CD69, H1.2F3, Biolegend
CD62L, MEL-14, Biolegend
CD8, 53-6.7, Biolegend or BD Bioscience
CX3CR1, SA011F11, Biolegend
Ki-67, 16A8, Biolegend
KLRG1, 2F1, Biolegend
Ly108, 330-AJ, Biolegend
PD1, 29F1.A12, Biolegend
PD1, RMPI-30, Biolegend
Tim3, RMT3-23, Biolegend
Tigit, GIGD7, Thermo Fisher Scientific
Tox, TXRX10, Thermo Fisher Scientific

Validation | All antibodies were obtained commercially and validation was based on the descriptions provided on the manufacturer's homepage.

# Animals and other organisms

Policy information about studies involving animals; ARRIVE guidelines recommended for reporting animal research

Laboratory animals
- 6-8 weeks C57BL/6 mice were obtained from the Australian Resources Centre or Envigo
- P14 TCR transgenic mice expressing diverse combinations of the congenic markers CD45.1/.2 and CD90.1/.2, as well as TCRa knockout mice were bred at the mouse facility of Technische Universität München, München, Germany.
Age- and sex-matched mice (all on C57BL/6 background) were used in this study and entered experiments at 6-8 weeks of age.
- MybGFP mice and Mybfl/flCd4Cre mice were bred and housed at the mouse facility of Peter Doherty Institute of Infection and Immunity

Wild animals | The study did not involve wild animals.

Field-collected samples | The study did not involve samples collected in the field.

Ethics oversight | Experimental protocols were approved by the committee for experimentation with laboratory animals of the district government of Upper Bavaria (Germany) or University of Melbourne Animal Ethics Commitee.

Note that full information on the approval of the study protocol must also be provided in the manuscript.

# Flow Cytometry

## Plots

Confirm that:

☒ The axis labels state the marker and fluorochrome used (e.g. CD4-FITC).

☒ The axis scales are clearly visible. Include numbers along axes only for bottom left plot of group (a 'group' is an analysis of identical markers).

☒ All plots are contour plots with outliers or pseudocolor plots.

☒ A numerical value for number of cells or percentage (with statistics) is provided.

## Methodology

Sample preparation | Single cell suspensions from different organs were obtained as described in the methods section.

Instrument | Cell sorting was performed on a MoFlo Astrios (Beckman Coulter). For data collection, a Cytoflex Lx cytometer (Beckman Coulter) was used.

Software | CytExpert (Beckman Coulter) v2.3.1.22 software was used for data collection. The acquired samples were analyzed using FlowJo software v10.6.1.

Cell population abundance | Sort purities were routinely confirmed, as assessed by post-sort measurements of the respective target cell populations.

Gating strategy
1. FSC/SSC gates were used to select lymphocyte populations.
2. FSC/FSC width gates were used to identify singlets.
3. Live/dead exclusion was performed using propidium iodide (PI) or eBioscience Fixable Viability Dye eF780
4. Further gating of target-cell populations relied on the respective marker combinations, for cell surface or intracellular markers, used in the specific experiments.

☒ Tick this box to confirm that a figure exemplifying the gating strategy is provided in the Supplementary Information.