## [Peer Review File · Nature]

Manuscript Title: c-Myb orchestrates T cell exhaustion and response to checkpoint inhibition

Reviewer Comments & Author Rebuttals

Reviewer Reports on the Initial Version:

Referees' comments:

Referee #1 (Remarks to the Author):

In this manuscript by Tsui et al., the authors dissect the heterogeneity of progenitor exhausted T cells to discover a subset of CD62Lhi progenitors with superior self-renewal and multipotency that is responsible for the long-term maintenance of exhausted T cell responses during chronic infections and their proliferative expansion following PDL-1 blockade. The authors further identify c-Myb as an essential transcription factor regulating CD62Lhi TPEX cell development/maintenance and T cell expansion in response to checkpoint inhibition. Paralleling CD8+ T cell biology in acute infections, c-Myb acted by limiting T cell effector functions and differentiation into terminally exhausted T cells.

The findings are novel and of interest to a wide audience given the major implications on immune checkpoint blockade therapy. However, several points need to be addressed to strengthen the authors' conclusions.

Major comments

1. The mechanistic part of the study is particularly underdeveloped. The authors conclude that "c-Myb regulates key genes required for self-renewal and function of exhausted T cells" based on the results of their RNA-seq profiling of WT and c-Myb-deficient TPEX cells. However, these results are profoundly biased by a skewing in TPEX subsets in the c-Myb cKO group. This can be clearly appreciated in the FACS plots depicted in Figure 3C, where CD62Lhi progenitors are virtually absent among c-Myb cKO cells. Thus, it is unclear if the transcriptional differences detected are directly dependent on the absence of c-Myb or secondary to the lack of CD62Lhi progenitors. The authors should perform new RNA-seq analyses on phenotypically normalized TPEX cells. For example, the frequency of c-KITnegCD62Lneg cells appears to be sufficient to perform bulk RNA-seq.
2. The authors propose that c-Myb expression is regulated by TCR signaling based on the experiments shown in Extended Data Figure 5. How do they reconcile this model with the observed downregulation of c-Myb in terminally exhausted cells, which likely are the cells receiving most antigenic stimulation? Also, if c-Myb is induced by TCR signaling, why do PD1-KO cells, which experience heightened TCR signaling, fail to form the c-Myb-dependent CD62Lhi TPEX subset? The mechanistic part exploring how c-Myb expression is regulated in exhausted T cells should be investigated in more detail.
3. The authors conclude that the fatal immunopathology occurring in Mybfl/fICd4cre animals is due to unrestricted effector CD8+ T cell responses since mice are protected by depletion of CD8+ T cells. While CD8+ T cells are clearly the final mediator of immunopathology in Mybfl/fICd4cre animals, it is unlikely that CD8+ T cell unrestricted effector differentiation alone is the primary cause. The lack of immunopathology in mixed Mybfl/fICd4Cre and Cd4Cre BM chimeras indicates that unrestricted

effector CD8+ T cell responses in c-Myb cKO cells are not sufficient to cause immunopathology. Given that c-Myb also plays a critical role in Treg differentiation and immunosuppressive function (Dias et al. *Immunity* 2017), it is plausible that competent Treg cells in chimeric animals are sufficient to prevent immunopathology. This should be experimentally addressed.

4. While the authors convincingly showed that c-Myb is critical for the early development of CD62Lhi TPEX, its long-term effect on exhausted CD8 T cells requires further validation. Since Mybfl/fl mice were originally maintained on a C57BL/6 × 129Sv background (Emambokus et al. *EMBO J*, 2003), the attrition of c-Myb cKO cells could be caused by immune rejection. To exclude this possibility, the authors should validate their results using Mybfl/flCd4Cre BM chimeras with Mybfl/fl instead of Cd4cre controls. Also, the authors present virtually all their c-Myb cKO data in terms of frequencies and not numbers. It is critical to provide absolute numbers since they can dramatically affect the overall conclusions (see PD1KO data, Figure 4F, where the authors show significant differences in frequency but not in cell numbers).

5. A well-established target of c-Myb is BCL-2. Do the authors see any differences in apoptosis/cell death in c-Myb cKO cells during chronic infection?

6. CD62Lhi TPEX display higher levels of Bach2, which has been recently shown to orchestrate the transcriptional and epigenetic programs of TPEX cells, including c-Myb (Yao et al. *Nat. Immunol* 2021). It would be important to investigate in a little more depth how these factors influence each other, and which one has a more dominant effect on the development of CD62Lhi TPEX cells.

7. A more semantic observation: given that CD62Lneg TPEX barely provide viral control at the given dose (Figure 4B), fail to expand in response to PD-1 antibody (Fig. 4J) and are biased towards forming CX3CR1neg (Tex2) progeny, should they be still called TPEX cells?

Minor points

1. The authors minimize the effect of c-Myb deletion in the formation of TCM cells: “These results were in stark contrast to the minor role of c-Myb in the differentiation of central memory T cells in response to acute LCMV-Armstrong infection”. It should be pointed out that Extended data Figure 6A-C examines memory T cell formation only at very early time points (day 21), and in animal lacking c-Myb in both CD8 and CD4 T cell compartments. Nevertheless, consistent with the literature, the differences in TCM formation were statistically significant. The authors should tone down this statement.

2. Fig. 2N, in the legend mistakenly referred to as Fig. 2L: the range of values the axis should be linear.

Referee #2 (Remarks to the Author):

In this study, the authors investigate CD8 T cell precursors of exhausted cells (Tpex) in the mouse LCMV model. They identify CD62L as a marker for a subset of Pex cells with enhanced capacity to self-renew in the context of chronic infection. After identifying myb as a gene that is preferentially expressed by CD62L+ Tpex, a T cell conditional knockout was used to assess the roles of myb in Tpex. This gave a dramatic and striking phenotype and resulted in fatal immunopathology in the context of chronic infection and not acute infection. This result was nicely investigated to determine that Myb is required for the function and self-renewal of exhausted cells, and that Myb expressing cells also mediate the effects of PD1 blockade. Thus the authors show that CD62L+ PD-1+Myb-

expressing Tpex cells can undergo a proliferative burst and differentiate in response to checkpoint blockade. While this study focuses only on the LCMV model of T cell exhaustion, this work provides a better understanding of the mechanisms of anti-PD1 based checkpoint blockade and could facilitate the development of strategies for identifying novel targets of intervention aimed at improving responses to checkpoint blockade immunotherapy. My specific comments are minor.

Lines 63-66 should mention relevance of Ly108 only to mice.

How were batch effects accounted when combining public datasets? How equal were the cluster compositions (and subsets of Tpex cells) seen across each of the samples/datasets analyzed? Was it necessary to include these additional datasets to obtain the results shown?

How was myb selected from the list of differentially expressed genes shown for Fig. 2ab? While some rationale is provided, this aspect of the narrative could be improved. Maybe a volcano plot could be used to further describe how this gene was selected?

Did the authors assess viral loads for mice showing in Fig. 2h,i? Was immunopathology associated with any increase in antigen load? I presume not if CD8 depletion was protective but I wonder if the authors could comment on this.

Referee #3 (Remarks to the Author):

The CD8+ T cell response to chronic viral infection is sustained by a specialized subset of CD8+ T cells, referred to as progenitor exhausted cells (Tpex). Tpex have been defined by the expression of various markers including Cxcr5, Ly108, Tcf1 or more recently Id3. Only Tcf1 has been shown to be essential for the generation/maintenance of Tpex.

Tsui et al have investigated whether Tpex (mostly defined here by Ly108 or Id3 expression) are heterogenous. They suggest that that long-term self-renewal capacity, multipotency and repopulation potential during chronic infection is associated with a subset of Tpex that expresses CD62L and that represents around 30% of Tpex. The authors further suggest that the transcription factor c-Myb is not needed for the generation of a Tpex compartment but is essential for the development of CD62L+ Tpex cells and for the long-term maintenance of the TCF1+ Tpex cell compartment. Finally, they provide evidence that the proliferative expansion in response to PD-1 checkpoint inhibition originates from CD62L+ Tpex cells in a c-Myb-dependent fashion.

The experiments are overall well performed and controlled. However, the overall conclusions drawn from this work are not fundamentally novel, as they are essentially the same as those drawn from the original discovery of Tpex based on Tcf1 expression (Im 2016, Utzschneider 2016, Wu 2016). The authors provide refinement of the prior analyses and suggest that the previously defined key functions of Tpex are assigned to a subset of the originally defined population. To what extent this refinement is novel, remains to be verified (see point 1 below).

Main points:

1. CD62L expression by a subset of T_{pex} has been reported before (see e.g. Im 2016, Utzschneider 2016). Further, functional heterogeneity of T_{pex} has also been reported before based on CD69 expression (Beltra 2020). Indeed, multiple aspects of CD69⁺ T_{pex} appear similar to those of CD62L⁺ T_{pex}, including gene expression profiles, performance in transfer experiments, etc. Thus, the relationship between CD62L⁺ and CD69⁺ T_{pex} needs to be addressed. For example, the Beltra paper suggested that CD69⁺ T_{pex} are resident in secondary lymphoid organs and that these cells yield recirculating CD69⁻ T_{pex}.

2. CD62L can be shed from the surface of cells and may thus not be a reliable marker. Key aspects of the cluster of Sell⁺ cells identified by scRNAseq data should be validated at the protein level. The question is whether the key gene expression differences between Sell⁺ versus Sell⁻ T_{pex} can be confirmed at the protein level in CD62L⁺ cells. Further, around 20% of CD62L⁺ cells lack T_{pex} markers (Ex Fig 2b), these CD62L⁺ T_{ex} cells should thus also be analyzed.

3. The role of Myb in the T cell response to chronic infection is interesting, but not completely dissected and understood. Myb is required for T cell development. Thus, it has to be determined whether the naïve Myb-deficient mice used here have a normal CD8⁺ T cell compartment and whether Myb-deficient and wild type bone marrow contributes equally to the naive CD8⁺ T cell compartment in the mixed bone marrow chimera. Further, as Myb expression during chronic infection is not restricted to CD62L⁺ T_{pex}, the effects of Myb-deficiency are difficult to assign. Provided the authors include additional controls (see just above and the remark regarding Fig2l below) the loss of CD62L⁺ T_{pex} is easy to assign. In contrast, the reasons for the transiently increased abundance of T_{ex}, the increased effector functions and/or the increased immunopathology can currently not be pinpointed. An elegant experiment would be the deletion of Myb in T_{ex}. At the least, functions of Myb-deficient cells should be addressed separately for T_{pex} and T_{ex} (Fig 2) (see also below).

4. The abstract states that CD62L⁺ T_{pex} cells “selectively preserve long-term self-renewal capacity, multipotency and repopulation potential during chronic infection.” To formally claim self-renewal capacity, the authors would need to perform tertiary transfers. To suggest multipotency CD62L⁺ T_{pex} would need to yield at least two types of progeny. However, based on adoptive transfers the authors suggest a single and linear developmental trajectory downstream of CD62L⁺ T_{pex}.

Additional points:

Fig. 1e: The abundance of CD62L⁺ and CD62L⁻ T_{pex} over time should be included.

Fig. 1g: The repopulation capacity of cells is influenced (among other things) by the initial take. Is that the same for the different populations tested herein? This issue is also crucial for the data shown in Fig.4.

Line 145 states that the “role of Myb in the exhausted T cell network is unknown”. However, the effects of Myb overexpression have been addressed (Chen 2019). This paper provides evidence that enforced Myb expression promotes the Ly108⁺ CD39⁻ (T_{pex}) subset and that this occurs at the expense of the Tim-3⁺ CD39⁺ (T_{ex}) subset. While this paper is cited, the statement that “Myb was shown to promote the survival of T_{pex} cells” seems rather vague.

Fig. 2g: What is the abundance of gp33⁺ cells?

Fig. 2j: Does the abundance of gp33⁺ cells differ?

Fig. 2k Ex Fig 6j: The authors should address where these changes occur both in T_{pex} and T_{ex}.

Incidentally it has also not been addressed whether wild-type CD62L⁺ and CD62L⁻ T_{pex} differ with regard to cytokine production.

Fig. 2l: This should be shown as Tcf1 vs CD62L. Ex Fig 6n: The abundance of CD62L⁺ T_{pex} appears

decreased while that of CD62L- T_{pex} is increased. Can the authors rule out the possibility that the cells are there but do simply not express CD62L? Can they define the population in an independent way?

Fig 2n: Do the mixed chimeras suffer from fatal immunopathology? It is stated that the entire Myb response contracts, but day 28 T_{ex} cells are still as abundant as the corresponding wild type cells, and later time points have not been analyzed. Ex Fig 7e,f: This should be shown as Tcf1 vs CD62L.

Fig 3c: What is the abundance of the different T_{pex} subpopulations?

Author Rebuttals to Initial Comments:

Manuscript 2021-11-18130C-Z

Tsui et al. ‘*c-Myb orchestrates T cell exhaustion and response to checkpoint inhibitor therapy during chronic infection*’

We would like to thank all reviewers for their interest and critical assessment of our manuscript and for giving us the opportunity to further clarify our findings. We have carefully considered all comments raised by the reviewers and addressed most of them with additional experiments. Overall, these data have allowed us to strengthen our manuscript and provide further support for our original conclusions. In summary, we have:

- 1. identified a new population of precursor exhausted T cells with true stem-like potential that display**
 - i) superior self-renewal, multipotency and expansion potential,
 - ii) an increased potential to differentiate into effector cells, and
 - iii) are the main mediators of the response to checkpoint inhibition.

- 2. identified c-Myb as a central transcriptional regulator of the exhausted T cell network that is**
 - i) specifically required for the differentiation and function of our newly identified stem-like precursor cells
 - ii) a central regulator of functional T cell exhaustion, and
 - iii) required for population expansion in response to checkpoint inhibition.

The new data presented in the revised version of our manuscript based on the reviewer’s comments are summarised here. We provide:

- i) new RNAseq data and an expanded analysis to further clarify the role of c-Myb in CD8+ T cell exhaustion in the context of chronic infection (**Fig. 3, Extended Data Fig. 15**), including new phenotypic and cellular data that confirm results of our RNAseq results (**Extended Data Fig. 15f, h**).
- ii) further evidence in support of our original conclusion that CD62L+ TPEX cells uniquely mediate stem-like self-renewal and expansion potential (**Extended Data Figs. 4, 5 and 7**).
- iii) a more detailed analysis of c-Myb-deficient TPEX and TEX cells, including functional and kinetic analyses (**Fig. 2m-p; Extended Data Figs 11k-r and 13**)
- iv) new evidence to support the notion that strong TCR signalling is a feature of CD62L+ TPEX cells (**Extended Data Fig. 8**) and a major driver of Myb expression in CD8+ T cells in LCMV infection (**Extended Data Fig. 9f**).

Overall, we believe that our findings represent a significant advancement in our understanding of the processes that control chronic CD8+ T cell responses and success of checkpoint inhibition. Due to its wide implication for immunotherapy, we are convinced that our work is of interest to a broad audience, including basic researchers and clinicians. Given the new data, we hope that the reviewers will find this work to be novel, solid and suitable for publication.

Point-by-point reply to reviewers' comments

Referee #1 (Remarks to the Author):

In this manuscript by Tsui et al., the authors dissect the heterogeneity of progenitor exhausted T cells to discover a subset of CD62Lhi progenitors with superior self-renewal and multipotency that is responsible for the long-term maintenance of exhausted T cell responses during chronic infections and their proliferative expansion following PDL-1 blockade. The authors further identify c-Myb as an essential transcription factor regulating CD62Lhi TPEX cell development/maintenance and T cell expansion in response to checkpoint inhibition. Paralleling CD8+ T cell biology in acute infections, c-Myb acted by limiting T cell effector functions and differentiation into terminally exhausted T cells.

The findings are novel and of interest to a wide audience given the major implications on immune checkpoint blockade therapy. However, several points need to be addressed to strengthen the authors' conclusions.

We would like to thank the reviewer for their positive and encouraging assessment of our study and the recognition of the novelty of our work. We have taken careful steps to address all the comments raised by the reviewer with new experimental data. The results have further strengthened our original conclusions and our manuscript as a whole. The specific points raised are addressed below:

Major comments

1. The mechanistic part of the study is particularly underdeveloped. The authors conclude that "c-Myb regulates key genes required for self-renewal and function of exhausted T cells" based on the results of their RNA-seq profiling of WT and c-Myb-deficient TPEX cells. However, these results are profoundly biased by a skewing in TPEX subsets in the c-Myb cKO group. This can be clearly appreciated in the FACS plots depicted in Figure 3C, where CD62Lhi progenitors are virtually absent among c-Myb cKO cells. Thus, it is unclear if the transcriptional differences detected are directly dependent on the absence of c-Myb or secondary to the lack of CD62Lhi progenitors. The authors should perform new RNA-seq analyses on phenotypically normalized TPEX cells. For example, the frequency of c-KITnegCD62Lneg cells appears to be sufficient to perform bulk RNA-seq.

Response: We agree with the reviewer and thank them for their suggestion. In line with the reviewer's suggestion, we have repeated our RNA-sequencing analysis. We have, however, opted to not sort for differential c-Kit expression as *Kit* is expressed by TPEX cells within the CD62L-negative compartment and a direct downstream target of c-Myb. Instead, we have sorted equivalent CD62L-negative Myb-cKO and control TPEX cells as well as control CD62L+ TPEX cells and compared them by RNAseq (**Fig. 3, Extended Data Fig. 15c, g**). Our extended results clearly reveal major transcriptional differences between all populations and show that c-Myb not only regulates the differentiation and transcriptional profile of CD62L+ TPEX cells but also impacts gene expression in CD62L- TPEX cells and TEX cells. Furthermore, we have successfully confirmed differential gene expression data at the protein level by demonstrating differential expression of c-Kit (as in original submission, **Fig. 3d**), differential localisation of PD-1+ c-Myb-KO and control CD8 T cells (**Extended Data Fig. 15f**), and differential representation of the CX3CR1+ and CD101+ TEX cell populations (new **Fig. 3g** and **Extended Data Fig. 15h**). These results further strengthen our originally proposed model where c-Myb plays crucial transcriptional roles in i) the development/maintenance of CD62L+ TPEX and ii) maintenance of long-term TPEX activity and iii) limiting effector/cytotoxic activities in TEX cells in the context of chronic infection.

2. *The authors propose that c-Myb expression is regulated by TCR signaling based on the experiments shown in Extended Data Figure 5. How do they reconcile this model with the observed downregulation of c-Myb in terminally exhausted cells, which likely are the cells receiving most antigenic stimulation? Also, if c-Myb is induced by TCR signaling, why do PD1-KO cells, which experience heightened TCR signaling, fail to form the c-Myb-dependent CD62Lhi TPEX subset? The mechanistic part exploring how c-Myb expression is regulated in exhausted T cells should be investigated in more detail.*

Response: We thank the reviewer for their comment and for the opportunity to clarify our data. Firstly, it is important to highlight that we and others have shown previously that TPEX cells experience higher TCR signalling than TEX cells and indeed form in response to strong TCR signals (Utzschneider *et al.* Nat. Immunol. 2020). To further test this model, we have now performed new adoptive transfer experiments using P14 cells expressing a Nur77-GFP reporter, which accurately indicates the strength of TCR signalling. Our data show that both CD62L+ and CD62L- TPEX cells receive heightened TCR signalling compared to TEX cells (**Extended Data Fig. 8**).

Secondly, we would like to emphasise that PD1-KO P14 cells used in our original experiments, which experience heightened TCR signalling, generate the same numbers of CD62L+ TPEX cells as control cells as shown in **Fig. 4f** (right panel). To further explore the roles of positive vs. negative signalling on *Myb* expression during chronic infection, we have now performed new experiments, in which we blocked inhibitory PD-1 signalling *in vivo* in chronically infected *Myb*-GFP reporter mice. In support of our original conclusion, we found that *Myb*-GFP expression was significantly enhanced by checkpoint inhibition in antigen-responsive cells (**Extended Data Fig. 9f**). In conjunction with our original data, which showed i) higher

Myb-GFP expression in antigen-specific cells in chronic compared to acute infection, and ii) *in vitro* dose-dependent response of *Myb*-GFP expression to TCR stimulation, we believe we can conclude that *Myb* is induced by TCR signalling.

3. The authors conclude that the fatal immunopathology occurring in *Myb^{fl/fl}Cd4^{cre}* animals is due to unrestricted effector CD8⁺ T cell responses since mice are protected by depletion of CD8⁺ T cells. While CD8⁺ T cells are clearly the final mediator of immunopathology in *Myb^{fl/fl}Cd4^{cre}* animals, it is unlikely that CD8⁺ T cell unrestricted effector differentiation alone is the primary cause. The lack of immunopathology in mixed *Myb^{fl/fl}Cd4^{Cre}* and *Cd4^{Cre}* BM chimeras indicates that unrestricted effector CD8⁺ T cell responses in *c-Myb* cKO cells are not sufficient to cause immunopathology. Given that *c-Myb* also plays a critical role in Treg differentiation and immunosuppressive function (Dias *et al.* *Immunity* 2017), it is plausible that competent Treg cells in chimeric animals are sufficient to prevent immunopathology. This should be experimentally addressed.

Response: We thank the reviewer for their comment and giving us the opportunity to clarify our results and provide new experimental data. Mixed bone marrow chimeric mice utilized in our original manuscript were reconstituted in a manner that only 10-20% of the cells were derived from the KO while the majority of the immune cells was wildtype derived. These mice stayed healthy for the duration of the experiments (up to 8 weeks). In contrast, mice harbouring 50% *c-Myb*-deficient CD8⁺ T cells were not protected from immunopathology, strongly suggesting that wildtype Tregs were not sufficient to protect the mice and CD8⁺ T cells were a major determining factor of disease outcome (**Reviewer Fig. 1**). We now provide a more detailed explanation of our experimental setup in the manuscript (lines 202-204).

To directly explore the role of Tregs in this context we have now performed additional experiments. To this end, we adoptively transferred splenic Tregs enriched from wildtype mice to either *Myb^{fl/fl}Cd4^{Cre}* (*Myb*-cKO) or *Myb^{fl/fl}* (littermate control) mice before infection with LCMV-Docile. This was done in a 1:1 donor-to-recipient ratio, ie Treg numbers corresponding to one wildtype spleen were transferred into each one recipient mouse. As seen in **Reviewer Fig. 2**, adoptive transfer of WT Tregs was not sufficient to rescue LCMV Docile-infected *Myb*-cKO mice. While we cannot completely rule out a role of impaired Treg function in exacerbating the immunopathology in *Myb*-cKO mice in chronic LCMV infection, our findings suggest a dominant role of CD8⁺ T cells.

4. While the authors convincingly showed that *c-Myb* is critical for the early development of CD62L^{hi} TPEX, its long-term effect on exhausted CD8 T cells requires further validation. Since *Myb^{fl/fl}* mice were originally maintained on a C57BL/6 × 129Sv background (Emambokus *et al.* *EMBO J*, 2003), the attrition of *c-Myb* cKO cells could be caused by immune rejection. To exclude this possibility, the authors should validate their results using *Myb^{fl/fl}Cd4^{Cre}* BM chimeras with *Myb^{fl/fl}* instead of *Cd4^{cre}* controls. Also, the authors present virtually all their *c-Myb* cKO data in terms of frequencies and not numbers. It is critical to provide absolute numbers since they can dramatically affect the overall conclusions (see

PDIKO data, Figure 4F, where the authors show significant differences in frequency but not in cell numbers).

Response: We thank the reviewer for raising this important point, and we agree that it is important to explore further the role of *Myb* in the long-term response of CD8⁺ T cells to chronic infection. We have therefore performed a series of additional experiments. Firstly, it is important to note that *Myb*^{fl/fl} mice, although made on C57BL/6 × 129Sv background, were backcrossed to C57B6 mice for >10 generations. Nevertheless, to rule out any rejection issues, we performed new experiments by generating two different cohorts of mixed bone marrow chimeras, using either *Cd4*Cre⁺ (**Reviewer Fig. 3a-d**) or littermate *Myb*^{fl/fl} mice (**Reviewer Fig. 3e-k**) as donors for the control compartment. After infection with LCMV-Docile, we found that both control CD8⁺ compartments (*Cd4*Cre⁺ and littermate *Myb*^{fl/fl}) behaved similarly. Specifically, in the context of chronic infection, in both cohorts we observed accelerated contraction of *Myb*-cKO antigen-responsive CD8⁺ T cells as seen in our original experiments, thus ruling out the role of immune rejection in the observed attrition (**Reviewer Fig. 3**). Importantly, we have now also extended our experimental observation period significantly and provide comprehensive analyses for later time points post infection (**Fig. 2m-p, Extended Data Fig. 13**), overall showing that *Myb* is required for i) the differentiation of CD62L⁺ TPEX, ii) the maintenance of TPEX and iii) the long-term maintenance of the CD8⁺ T cell response in general. As requested, we now also provide absolute numbers of CD8⁺ T cells, antigen-specific CD8⁺ T cells, CD62L⁺ TPEX, CD62L⁻ TPEX and TEX cells in the spleens of control and *Myb*-cKO animals after LCMV-Docile infection (**Extended Data Fig. 11r**), which show a specific depletion of CD62L⁺ TPEX cells in the *Myb*-cKO mice.

5. *A well-established target of c-Myb is BCL-2. Do the authors see any differences in apoptosis/cell death in c-Myb cKO cells during chronic infection?*

Response: We thank the reviewer for raising this important point, which we have addressed in new experiments. To this end, we measured BCL2 expression and apoptosis in *Myb*-cKO and control antigen-responsive CD8 T cells after infection with LCMV-Docile. We observed no significant difference in either BCL2 or apoptosis (**Extended Data Fig. 12e and Reviewer Fig. 4**), supporting the notion that c-Myb exerts its function in the maintenance of chronic T cell responses mainly by regulating cell cycle activity.

6. *CD62L^{hi} TPEX display higher levels of Bach2, which has been recently shown to orchestrate the transcriptional and epigenetic programs of TPEX cells, including c-Myb (Yao et al. Nat. Immunol 2021). It would be important to investigate in a little more depth how these factors influence each other, and which one has a more dominant effect on the development of CD62L^{hi} TPEX cells.*

Response: This is an interesting point. Indeed, *Bach2* is an important regulator of TPEX cells during chronic infection as we and others have shown (Utzschneider *et al.* Nat. Immunol. 2020, Yao *et al.* Nat. Immunol 2021). To comprehensively address the role of *Bach2*, we infected

Bach2^{fl/fl}*Cd4*Cre (Bach2-cKO) and littermate *Bach2*^{fl/fl} control mice with LCMV Docile and analysed on day 12 post infection both the TPEX and TEX compartments. Our results indicated that, in stark contrast to Myb-cKO mice, Bach2-cKO mice showed no signs of immunopathology and mounted similar CD8⁺ T cell responses to chronic infection. Importantly, while frequencies of TPEX were decreased (as we and others published), the formation of CD62L⁺ TPEX per se was intact, indicating that *Myb* but not *Bach2* controls the development of CD62L⁺ TPEX cells and suggesting that *Myb* expression is intact in the absence of *Bach2*. Supporting this notion, we observed that Bach2-cKO and control TPEX cells expressed similar amounts of the *Myb* target c-Kit. Consistent with the presence of normal CD62L⁺ TPEX cells in the Bach2-cKO, we observed no premature attrition of the antigen-specific compartment in Bach2-cKO mice (**Reviewer Fig. 5**). Finally, unlike Myb-cKO mice, we only found a modest decrease of CX3CR1⁺ TEX cells in Bach2-cKO mice. Lastly, to explore how c-Myb deficiency affects the expression of *Bach2*, we analysed *Bach2* expression in control and Myb-cKO TPEX and TEX P14 cells and found that *Bach2* expression was similar between control and Myb-cKO cells. Altogether, our results indicate that *Bach2* and *Myb* expression are independent of each other and that both factors play distinct roles in the differentiation and maintenance of TPEX cells (**Reviewer Fig. 5**). As we believe that these results are somewhat beyond the scope of this study, we have included these data as a reviewer figure here only. However, if the reviewer or editor considers the data important enough, we would of course be delighted to include them in the manuscript.

7. A more semantic observation: given that CD62L^{neg} TPEX barely provide viral control at the given dose (Figure 4B), fail to expand in response to PD-1 antibody (Fig. 4J) and are biased towards forming CX3CR1^{neg} (*Tex2*) progeny, should they be still called TPEX cells?

Response: This is an interesting point of discussion that should be addressed in a future review. We certainly agree that the main ‘stem-like’ capacity for self-renewal and expansion is contained within CD62L⁺ TPEX. However, as seen in our transfer experiments, CD62L⁻ TPEX also have some capacity for self-renewal and expansion. Thus, CD62L⁺ TPEX are essential as a resource population to ‘feed’ the overall CD8⁺ T cell response.

Minor points

1. The authors minimize the effect of c-Myb deletion in the formation of TCM cells: “These results were in stark contrast to the minor role of c-Myb in the differentiation of central memory T cells in response to acute LCMV-Armstrong infection”. It should be pointed out that Extended data Figure 6A-C examines memory T cell formation only at very early time points (day 21), and in animal lacking c-Myb in both CD8 and CD4 T cell compartments. Nevertheless, consistent with the literature, the differences in TCM formation were statistically significant. The authors should tone down this statement.

Response: We thank the reviewer for raising this point. We certainly do not want to minimize the importance of c-Myb in the generation or maintenance of central memory T cells and have therefore removed the related sentence.

2. *Fig. 2N, in the legend mistakenly referred to as Fig. 2L: the range of values the axis should be linear.*

Response to minor comment 2: We thank reviewer for pointing out this mistake. We have corrected the data display (now **Extended Data Fig. 13a**).

Referee #2 (Remarks to the Author):

In this study, the authors investigate CD8 T cell precursors of exhausted cells (Tpex) in the mouse LCMV model. They identify CD62L as a marker for a subset of Pex cells with enhanced capacity to self-renew in the context of chronic infection. After identifying myb as a gene that is preferentially expressed by CD62L+ Tpex, a T cell conditional knockouts were used to assess the roles of myb in Tpex. This gave a dramatic and striking phenotype and resulted in fatal immunopathology in the context of chronic infection and not acute infection. This result was nicely investigated to determine that Myb is required for the function and self-renewal of exhausted cells, and that Myb expressing cells are also mediate the effects of PD1 blockade. Thus the authors show that CD62L+ PD-1+Myb-expressing Tpex cells can undergo a proliferative burst and differentiate in response to checkpoint blockade. While this study focuses only on the LCMV model of T cell exhaustion, this work provides a better understanding of the mechanisms of anti-PD1 based checkpoint blockade and could facilitate the development of strategies for identifying novel targets of intervention aimed at improving responses to checkpoint blockade immunotherapy. My specific comments are minor.

We would like to thank the reviewer for their insightful summary of our data and the positive assessment of our study. Specific issues are addressed below:

1. *Lines 63-66 should mention relevance of Ly108 only to mice.*

Response: We thank R2 for raising this point and we have modified the text to reflect the relevance of Ly108 only in mice. Please see line 63.

2. *How were batch effects accounted when combining public datasets? How equal were the cluster compositions (and subsets of Tpex cells) seen across each of the samples/datasets analyzed? Was it necessary to include these additional datasets to obtain the results shown?*

Response: We thank R2 for raising this comment. The batch correction was performed while integrating samples with the anchoring function implemented in Seurat as described in *Materials and methods* of our manuscript. In brief, anchors were calculated using normalised canonical correlation analysis and mutual nearest neighbour identification in a low dimensional state. Compared to classical batch correction, this approach ensures that only overlapping cell states among datasets are used as anchors but not non-overlapping cell states. As shown in **Reviewer Figure 6**, all clusters, including the two TPEX cell subsets, identified in this manuscript, can be observed in all datasets used when displayed independently, indicating that while our conclusions were not made based on the effect of single datasets, our combined clustering approach enabled a confident representation of a typical exhausted T cell landscape in the context of chronic LCMV infection.

3. *How was myb selected from the list of differentially expressed genes shown for Fig. 2ab?*

While some rationale is provided, this aspect of the narrative could be improved. Maybe a volcano plot could be used to further describe how this gene was selected?

Response: We thank R2 for raising this point. We have included here a summary of how we selected *Myb* from the list of signature transcripts. We used the DAVID Functional Annotation Bioinformatics Microarray Analysis tool to identify transcripts encoding proteins involved in DNA binding or positive and negative regulation of transcription. We obtained 15 transcription factors, which were further explored by enrichment analyses using existing RNAseq dataset from Hudson *et al.* Immunity 2019 and our own study (Utzschneider *et al.* Nat. Immunol. 2019), both of which contain comprehensive longitudinal transcriptomic data of TPEX and TEX CD8+ T cells in the context of chronic LCMV infection. Notably, we found that of the 15 transcriptional obtained, only *Myb* showed consistent enrichment in TPEX cells. *c-Myb* was selected based on this specificity in expression pattern and its known role as regulator of hematopoietic stem cells and involvement T cell memory. We have now provided additional explanation to our methodology in lines 539-542.

4. Did the authors assess viral loads for mice showing in Fig. 2h,i? Was immunopathology associated with any increase in antigen load? I presume not if CD8 depletion was protective but I wonder if the authors could comment on this.

Response: We thank R2 for raising this important point. We have now provided additional analysis of viral burden of *Myb*-cKO and littermate control animals after LCMV-Docile infection. Our data show that viral clearance was not different between control and *Myb*-cKO animals (**Extended Data Fig. 11o**).

Referee #3 (Remarks to the Author):

The CD8⁺ T cell response to chronic viral infection is sustained by a specialized subset of CD8⁺ T cells, referred to as progenitor exhausted cells (Tpex). Tpex have been defined by the expression of various markers including Cxcr5, Ly108, Tcf1 or more recently Id3. Only Tcf1 has been shown to be essential for the generation/maintenance of Tpex. Tsui et al have investigated whether Tpex (mostly defined here by Ly108 or Id3 expression) are heterogenous. They suggest that that long-term self-renewal capacity, multipotency and repopulation potential during chronic infection is associated with a subset of Tpex that expresses CD62L and that represents around 30% of Tpex. The authors further suggest that the transcription factor c-Myb is not needed for the generation of a Tpex compartment but is essential for the development of CD62L⁺ Tpex cells and for the long-term maintenance of the TCF1⁺ Tpex cell compartment. Finally, they provide evidence that the proliferative expansion in response to PD-1 checkpoint inhibition originates from CD62L⁺ Tpex cells in a c-Myb-dependent fashion.

The experiments are overall well performed and controlled. However, the overall conclusions drawn from this work are not fundamentally novel, as they are essentially the same as those drawn from the original discovery of Tpex based on Tcf1 expression (Im 2016, Utzschneider 2016, Wu 2016). The authors provide refinement of the prior analyses and suggest that the previously defined key functions of Tpex are assigned to a subset of the originally defined population. To what extent this refinement is novel, remains to be verified (see point 1 below).

We thank the reviewer for their critical assessment of our study, and we appreciate the opportunity to clarify some of the points that have not been made sufficiently clear. We also thank the reviewer for suggesting some important experiments that have allowed us to provide further support to our conclusions.

Addressing the novelty point raised by the reviewer, we fully acknowledge the importance of the original 2016 studies that identified TCF1⁺ TPEX cells among exhausted T cells and showed that these cells are critical for the maintenance of CD8⁺ T cell responses to chronic infection and tumours. Indeed, these studies have revolutionised our understanding of T cell exhaustion and have since served as the benchmark for future studies of chronic CD8⁺ T cell responses in mice and human. However, we believe that our study provides a fundamental advancement to the existing model. Specifically, while the original studies describe the TCF1⁺ TPEX cell compartment and demonstrate its self-renewal capacity and responses to checkpoint inhibition, our study for the first time identifies the true stem-like population and establishes the molecular basis for their function. Indeed, our data demonstrate that the vast majority (about 90%) of TCF1⁺ cells do not efficiently mediate self-renewal and responsiveness to checkpoint inhibition. Instead, these properties originate from a small population of phenotypically and transcriptionally distinct T cells that co-express TCF1, CD62L and c-Myb. Moreover, we show for the first time that these CD62L⁺ TPEX cells, while numerically inferior (1~2% of the entire antigen-specific compartment), uniquely possess potential for proliferation, repopulation and multipotency, all critical for maintaining long-term CD8⁺ T cell responses to chronic infection.

Crucially, these functional characteristics allow CD62L+ TPEX cells to respond most robustly to PD-1 checkpoint blockade. We identify c-Myb to be critical for the development of CD62L+ TPEX cells and thereby for the maintenance of the entire TPEX cell compartment in chronic infection. Lack of c-Myb not only severely impairs the maintenance of chronic CD8+ T cell responses but also the responses to checkpoint inhibition. Taken together, our findings demonstrate that TCF1 expression alone is not sufficient to maintain TPEX cells and chronic CD8+ T cell responses and therefore a poor predictor of self-renewal and multipotent capacities. Instead, both TCF1 and c-Myb are required for TPEX cell maintenance and function. We believe that our findings represent a fundamental qualitative advance, which can be viewed akin to the identification of true hematopoietic stem cells (HSCs) in the bone marrow. The self-renewal capacity of hematopoietic cells is contained in the progenitor compartment of the bone marrow; however, among the progenitors, true hematopoietic stem cells are responsible for self-renewal and multipotency. In a similar manner, the identification of true stem-like TPEX cells that maintain the antigen-specific T cell compartment in chronic infection allows us to advance beyond a mere theoretical concept of stemness and towards an actual understanding of the molecular underpinnings of this cellular entity. We hope the reviewer agrees with us that these findings are novel and fundamentally important.

Specific questions are addressed below:

Main points:

1. CD62L expression by a subset of T_{pex} has been reported before (see e.g. Im 2016, Utzschneider 2016). Further, functional heterogeneity of T_{pex} has also been reported before based on CD69 expression (Beltra 2020). Indeed, multiple aspects of CD69+ T_{pex} appear similar to those of CD62L+ T_{pex}, including gene expression profiles, performance in transfer experiments, etc. Thus, the relationship between CD62L+ and CD69+ T_{pex} needs to be addressed. For example, the Beltra paper suggested that CD69+ T_{pex} are resident in secondary lymphoid organs and that these cells yield recirculating CD69- T_{pex}.

Response: We thank R3 for raising this interesting point. We acknowledge that previous reports have shown that TPEX cells are phenotypically heterogenous. However, apart from Beltra *et al.* (Immunity 2020) who examined subpopulations identified based on CD69, there have been no attempts to further characterize different TPEX populations molecularly or functionally. Importantly, Beltra *et al.* have not shown or claimed that TPEX cell populations segregated based on CD69 were fundamentally different or had different repopulation potential. Indeed, their adoptive transfer experiment (Beltra *et al.* Fig. 2B-D, S2A-E) shows no difference in development and repopulation potential, nor have they explored the origin of the proliferative burst in response to PD-1 blockade.

To formally investigate the potential relationship between the populations identified by differential expression of CD69 and CD62L, we first performed new bioinformatic analyses of

our scRNA dataset. Unlike *Sell* (encoding CD62L), *Cd69* expression was homogenously distributed among the cells within the UMAP landscape and did not exhibit preferential expression in any of the TPEX or TEX cell clusters identified (new **Extended Data Fig. 7a**). We also performed enrichment analyses using the transcriptional signatures based on surface CD69 expression on TPEX cells published by Beltra *et al.* These analyses showed that the Ly108+CD69+ ('TEX prog1' as per Beltra *et al.*) signature was similarly expressed by CD62L+ and CD62L- TPEX and intermediate cells, while the Ly108+CD69- ('TEX prog2') signature was highly expressed by proliferating cells but not by TPEX cells (new **Extended Data Fig. 7b**). Finally, we observed surface CD69 expression in both CD62L+ and CD62L- TPEX cells, albeit at higher frequencies among CD62L- TPEX cell (new **Extended Data Fig. 7c**). Overall, our results point to the conclusion that CD69 expression, at transcript and protein levels, does not segregate transcriptionally distinct TPEX populations, but likely reflects differences in the activation status of exhausted T cells.

To directly test the developmental potential of TPEX populations identified by differential expression of CD69 under our experimental conditions, we sorted CD69+ and CD69- TPEX cells and transferred them into chronic infection-matched recipients. In line with the data by Beltra *et al.*, we saw no difference in the expansion and developmental potential and no preferential differentiation into CD62L+/- TPEX or TEX cells (new **Extended Data Fig. 7d-j**). Taken together, these new data show that there is no relationship between CD69 and CD62L expression and demonstrate that TPEX cells segregated based on CD69 show no differential developmental potential. Overall, these results strengthen our original conclusion that CD62L+ TPEX cells are a unique population of stem-like TPEX cells positioned at the apex of the exhausted T cell network. These additional data have been included in the new version of the manuscript (**Extended Data Fig. 7**).

2. CD62L can be shed from the surface of cells and may thus not be a reliable marker. Key aspects of the cluster of Sell+ cells identified by scRNAseq data should be validated at the protein level. The question is whether the key gene expression differences between Sell+ versus Sell- Tpx can be confirmed at the protein level in CD62L+ cells. Further, around 20% of CD62L+ cells lack Tpx markers (Ex Fig 2b), these CD62L+ Tex cells should thus also be analyzed.

Response: We thank R3 for raising these important points. Firstly, to formally confirm that key transcriptional differences between *Sell*+ and *Sell*- TPEX cells (as identified by scRNAseq) can be fully captured by CD62L+ and CD62L- TPEX cells, we performed new bulk RNAseq experiments on TPEX cells sorted based on surface CD62L expression. Importantly, CD62L protein expression highly correlated with the expression of *Sell*, *Ccr7*, *S1pr1* and *Klf2* transcripts (key markers of the *Sell*+ TPEX cluster) and many more in TPEX cells (**Extended Data Fig. 15c**). Moreover, the *Sell*+ TPEX cell transcriptional signature derived from our scRNAseq dataset was significantly enriched in our sorted CD62L+ TPEX compared with CD62L- TPEX cells, suggesting that surface CD62L expression in TPEX cells accurately captures the transcriptionally and functionally distinct *Sell*+ TPEX cells during chronic infection (**Reviewer Fig. 7a-c**). In support to this notion, results of our initial bulk RNAseq

experiments suggested that Myb-cKO TPEX cells, which lack surface CD62L expression, do not solely lose *Sell* expression, but indeed lose the entire transcriptional signature of the *Sell*+ TPEX cluster (**Extended Data Fig. 15b**). As suggested, we have also analysed the expression of proteins encoded by genes differentially expressed between CD62L+ and CD62L- TPEX cells. In line with our original scRNAseq and new bulk RNAseq dataset, CD62L+ TPEX cells expressed significantly less ICOS and c-Kit compared with CD62L- TPEX cells (**Extended Data Fig. 2e, f; Fig. 3d and Reviewer Fig. 7b**).

To directly address potential shedding of CD62L, we have also made use of the shedding inhibitor TAPI-1 (Mahnke *et al.* J. Invest. Dermatol. 2017) and processed our samples with and without this inhibitor. As seen in **Reviewer Fig. 7d**, there was no evidence for CD62L shedding under the experimental conditions used.

Lastly, as the reviewer points out, we do observe a small number of CD62L+ TEX cells. As these cells lack expression of TCF1, which is critical for TPEX function, we assumed that this expression of CD62L is residual and may represent cells that have recently progressed to the TEX compartment. To directly address this issue, we sorted CD62L+ TEX cells and adoptively transferred them into chronically infection-matched mice. In contrast to CD62L+ TPEX cells, CD62L+ TEX cells had no self-renewal capacity and significantly reduced re-expansion potential (**Extended Data Fig. 5**). Taken together, these new data, in addition to our original findings, strongly suggested that CD62L+ TPEX cells are transcriptionally and phenotypically distinct to CD62L- TPEX and TEX cells and possess superior developmental and re-expansion potential and unique stem-like properties.

3. The role of Myb in the T cell response to chronic infection is interesting, but not completely dissected and understood. Myb is required for T cell development. Thus, it has to be determined whether i) the naïve Myb-deficient mice used here have a normal CD8+ T cell compartment and ii) whether Myb-deficient and wild type bone marrow contributes equally to the naïve CD8+ T cell compartment in the mixed bone marrow chimera. iii) Further, as Myb expression during chronic infection is not restricted to CD62L+ TpeX, the effects of Myb-deficiency are difficult to assign. Provided the authors include additional controls (see just above and the remark regarding Fig2l below) the loss of CD62L+ TpeX is easy to assign. In contrast, the reasons for the transiently increased abundance of Tex, the increased effector functions and/or the increased immunopathology can currently not be pinpointed. An elegant experiment would be the deletion of Myb in Tex. At the least, functions of Myb-deficient cells should be addressed separately for TpeX and Tex (Fig 2) (see also below).

Response: We thank R3 for raising these important points and suggestions, which we address here:

i) We performed new experiments to analyse in detail the development of mature CD8+ T cells in *Mybfl/flCd4Cre* (Myb-cKO) and littermate *Mybfl/fl* control mice. In line with previous reports (Lieu *et al.* PNAS. 2014), our results show that Myb-cKO mice have largely normal

compartments of mature CD8⁺ T cells in the thymus, spleen and lymph nodes. Furthermore, mature c-Myb-deficient and control CD8⁺ T cells in naïve mice did not exhibit differences in activation status nor did they show differential expression of markers indicative of the naïve T cell fate (CD127 (IL-7R), CCR7 and CD25 (IL2R)) (**Extended Data Fig. 10**), overall indicating that the CD8⁺ T cell compartments in Myb-cKO mice are very similar to controls. This conclusion is also supported by the notion that Myb-cKO mice mount a normal response to acute infection, as shown in our original submission (now **Fig. 2g** and **Extended Data Fig. 11a-d**).

ii) In our initial experiments, we have extensively tested the quality and impact of chimerism on the T cell response. In these experiments, we found that mixed bone marrow chimeric mice harbouring about 50% of Myb-cKO CD8⁺ T cells (equal contribution) were prone to severe immunopathology similar to the Myb-cKO in response to LCMV-Docile infection (**Reviewer Fig. 1**). Thus, these chimeras were not suitable to study long-term outcomes of chronic infection. To overcome this effect, we reduced the proportion of Myb-cKO bone marrow to around 10~20% of Myb-cKO CD8⁺ T cells. These mice survived the chronic infection without any discernible pathology and were therefore used throughout our study (**Reviewer Fig. 1**). We have made this now clearer in the manuscript (lines 202-204). In line with the largely normal development of CD8⁺ T cells, the c-Myb-deficient and control CD8⁺ T cell compartments in these fully reconstituted chimeric mice contained very similar proportions of naïve CD8⁺ T cells (**Reviewer Fig. 8**).

iii) Unfortunately, there are no tools available to specifically delete c-Myb in T_{EX} cells. We have therefore, as requested by the reviewer, extended our characterisation of antigen-responsive Myb-cKO and control CD8 T cells. To this end, we analysed cytokine production capacity, granzyme B expression and cell cycle activity separately in T_{PEX} and T_{EX} cells. We found that increased cytokine production was evident in both Myb-cKO T_{PEX} and T_{EX} cells (**Extended Data Fig. 11k**), indicating that c-Myb generally limits cytokine production in exhausted T cells. In contrast, granzyme B expression was limited to T_{EX} cells in both genotypes, but was expressed at higher levels in Myb-cKO T_{EX} cells (**Extended Data Fig. 11m**). In addition, we found that cell cycle activity dysregulation (measured by Ki67) in the absence of c-Myb were different between T_{PEX} and T_{EX} cells. While Ki67 expression was initially similar or even higher in T_{EX} cells in the absence of c-Myb, it was tightly downregulated at later timepoints compared to control cells (**Extended Data Fig. 13c, d**).

4. The abstract states that CD62L⁺ T_{pex} cells “selectively preserve long-term self-renewal capacity, multipotency and repopulation potential during chronic infection.” To formally claim self-renewal capacity, the authors would need to perform tertiary transfers. To suggest multipotency CD62L⁺ T_{pex} would need to yield at least two types of progeny. However, based on adoptive transfers the authors suggest a single and linear developmental trajectory downstream of CD62L⁺ T_{pex}.

Response: We thank R3 for raising this interesting point and for allowing us to further strengthen our conclusions. As suggested by the reviewer we have addressed the point of ‘self-renewal’ experimentally by tertiary transfers. To this end, we compared the repopulation potential of CD62L⁺ and CD62L⁻ TPEX cells in consecutive transfers. In line with our original conclusion, we found that CD62L⁺ TPEX, but not CD62L⁻ TPEX cells, were able to re-expand and repopulate the exhausted T cell compartment after tertiary retransfer. More importantly, we observed no apparent deterioration of their capacity between secondary and tertiary transfers, strongly suggesting that robust self-renewal capacity and expansion potential are selectively preserved in CD62L⁺ TPEX cells. These data have now been included in **Extended Data Fig. 4**.

Regarding the term ‘multipotency’ it is important to note that in our original manuscript, we not only described the linear one-way development trajectory of CD62L⁺ TPEX cells in the context of chronic antigen exposure. We also showed that CD62L⁺ TPEX, but not CD62L⁻ TPEX cells, were capable of differentiating into KLRG1⁺ effector cells in an acute recall response (**Figs 1o and 4b**). Notably, CD62L⁺ TPEX cells compared to CD62L⁻ TPEX cells were also superior in repopulating the CX3CR1⁺ TEX compartment (**Extended Data Fig. 17e**), suggesting that CD62L⁺ TPEX cells can give rise to CX3CR1⁺ TEX cells without progressing through CD62L⁻ TPEX cells. Thus, CD62L⁺ TPEX cells can give rise not only to CD62L⁻ TPEX but also to KLRG1⁺ and CX3CR1⁺ effector cells *depending on the given environmental cues*. We believe these results justify the application of the term ‘multipotency’.

Additional points:

Fig. 1e: The abundance of CD62L⁺ and CD62L⁻ TpeX over time should be included.

Response: We agree that this is important information, which we have now included in **Extended Data Fig. 2c**.

Fig. 1g: The repopulation capacity of cells is influenced (among other things) by the initial take. Is that the same for the different populations tested herein? This issue is also crucial for the data shown in Fig. 4.

Response: We thank the reviewer for this important question. Importantly, in our experiments related to checkpoint inhibitor therapy (**Fig. 4**), that R3 refers to, we explicitly controlled for variation in take rate of distinct T cell subsets by measuring the fold-expansion of a transferred T cell subset in response to checkpoint inhibitors in relation to the same T cell subset’s size in absence of checkpoint inhibitors (**Fig. 4i-k**). In this setting, it is reasonable to assume an equal take for both conditions prior to initiation of the treatment. Of note: This procedure, has also been utilized successfully by Im *et al.* to control for potential differences in take rate (Im *et al.* Nature 2016). In addition, our c-Myb-KO experiments included in **Fig. 4l-n** do not rely on adoptively transferring different T cell subsets. Instead, and in agreement with our previous

data illustrating the selective depletion of the CD62L+ TPEX compartment in the c-Myb-KO, responsiveness to checkpoint inhibition is lost in c-Myb-KO cells, further supporting the conclusion that a functional CD62L+ TPEX cell compartment is required to fuel the proliferative response to checkpoint inhibition.

Line 145 states that the “role of Myb in the exhausted T cell network is unknown”. However, the effects of Myb overexpression have been addressed (Chen 2019). This paper provides evidence that enforced Myb expression promotes the Ly108+ CD39- (Tpex) subset and that this occurs at the expense of the Tim-3+ CD39+ (Tex) subset. While this paper is cited, the statement that “Myb was shown to promote the survival of Tpex cells” seems rather vague.

Response: We thank R3 for raising this point. We fully acknowledge these previous findings and have modified our description of this finding in the updated manuscript to make this point clearer. Please see lines 325-326.

Fig. 2g: What is the abundance of gp33+ cells?

Response: We have now included the abundance of gp33+ cells in **Extended Data Fig. 11d**.

Fig. 2j: Does the abundance of gp33+ cells differ?

Response: We have now included the abundance of gp33+ cells and the different subsets in **Extended Data Fig. 11r**. In this context, it is important to point to the overall cellularity of the spleen, which is substantially decreased (about 10-fold) in the Myb-cKO, a feature of severe immune pathology (**Extended Data Fig. 11e**). This is also reflected by the overall lower number of CD8+ T cells (**Extended Data Fig. 11r**). Taking this into consideration, there is a relative increase in gp33+ T cell numbers and a dramatic depletion of CD62L+ TPEX numbers in the Myb-cKO compared to controls.

Fig. 2k Ex Fig 6j: The authors should address where these changes occur both in Tpex and Tex. Incidentally it has also not been addressed whether wild-type CD62L+ and CD62L- Tpex differ with regard to cytokine production.

Response: We thank R3 for allowing us to clarify these results. As requested by the reviewer, we now provide a more detailed characterisation of control and Myb-cKO TPEX and TEX cell function (also related to major comment 3) in the context of chronic LCMV infection. We have included data on cytokine production and GZMB expression in TPEX and TEX cells in **Extended Data Fig. 11k, m**. We now also include new data related to the cytokine-producing capacities of the two different TPEX cell subsets in **Extended Data Fig. 2j**.

Fig. 2l: This should be shown as Tcf1 vs CD62L.

Response: We agree and have now altered the display as suggested (**Fig. 2l**)

Ex Fig 6n: The abundance of CD62L+ T_{pex} appears decreased while that of CD62L- T_{pex} is increased. Can the authors rule out the possibility that the cells are there but do simply not express CD62L? Can they define the population in an independent way?

Response: Our initial gene set enrichment analysis comparing Myb-deficient and control TPEX cells clearly showed significant loss not only of *Sell* itself but the whole *Sell*+ TPEX cell transcriptional signature in Myb-deficient TPEX cells (original **Fig. 3a**; now **Extended Data Fig. 15b**). This strongly indicated that the lack of CD62L among Myb-deficient TPEX cells was due to the loss of the CD62L+ TPEX cell population rather than the mere loss of CD62L from the cell surface. We have expanded on our analyses and provided a more detailed explanation under major comment 2.

Fig 2n: Do the mixed chimeras suffer from fatal immunopathology? It is stated that the entire Myb response contracts, but day 28 T_{ex} cells are still as abundant as the corresponding wild type cells, and later time points have not been analyzed.

Response: We thank R3 for this comment and giving us the opportunity to clarify our results and provide new experimental data. As also outlined under major comment 3, mixed bone marrow chimeric mice utilized in our original manuscript were reconstituted in manner that only 10-20% of the cells were derived from the KO while the majority of the immune cells was wildtype derived. These mice stayed healthy for the duration of the experiments (up to 8 weeks). In contrast, mice harbouring 50% Myb-deficient CD8+ T cells were not protected from immunopathology (**Reviewer Fig. 1**). We now provide a more detailed explanation of our experimental setup in the manuscript (lines 202-204). In response to the second part of the question, we have now extended our analyses significantly and tracked the immune response until day 70 post infection. These data confirm our original conclusion and show in the absence of c-Myb a loss of CD62L+ TPEX, followed by the loss TCF1+ TPEX cells and a subsequent overall loss of antigen-responsive CD8+ T cells, including gp33+ and PD-1+ cells (**Fig. 2o** and **Extended Data Fig. 13b**).

Ex Fig 7e, f: This should be shown as Tcf1 vs CD62L.

Response: we have now altered the display accordingly (**Extended Data Fig. 12g**)

Fig 3c: What is the abundance of the different T_{pex} subpopulations?

Response: we have now provided the abundance of different TPEX subpopulations (**Extended Data Fig. 15e**).

Reviewer Figure 1. (a-g) Mixed bone marrow chimeric mice harbouring Myb-cKO and *Cd4Cre* (Ctrl) T cells at 1:1 (b-d) and 1:4 (e-g) ratios were infected with LCMV-Docile and monitored for clinical symptoms p.i. (a) Schematic of the experimental setup. Flow cytometry plot (b) and quantification (c) showing ratios of CD8⁺ T cell make-up in reconstituted 1:1 mixed bone marrow chimeric mice pre-infection. (d) Survival curve of the corresponding cohort of chimeric mice p.i. with LCMV-Docile. Flow cytometry plot (e) and quantification (f) showing ratios of CD8⁺ T cell make-up in reconstituted 1:4 mixed bone marrow chimeric mice pre-infection. (g) Survival curve of the corresponding cohort of chimeric mice p.i. with LCMV-Docile. Box plots indicate range, interquartile and median; each dot represents an individual mouse; horizontal line in (c, f) indicate median. *P* values are from two-tailed unpaired t-tests (c, f); n.s., not significant; * *p* <0.05, ** *p* <0.01, *** *p* <0.001, **** *p* <0.0001.

Reviewer Figure 2. (a, b) Enriched *wildtype* splenic CD4⁺ Tregs were injected i.v. to *Myb^{fl/fl}Cd4^{Cre}* (Myb-cKO) and littermate *Myb^{fl/fl}* control (Ctrl) mice at a donor-to-recipient ratio of 1:1. The mice were then infected with LCMV-Docile. **(a)** Schematic of the experimental setup. **(b)** Survival curve of control and Myb-cKO mice p.i. with LCMV-Docile.

Reviewer Figure 3. (a-d) Mixed bone marrow chimeric mice harbouring Myb-cKO and *Cd4Cre* (Ctrl) T cells were infected with LCMV-Docile and analysed at the indicated time points p.i. **(a)** Schematic of the experimental setup. Quantifications showing the frequencies of control and Myb-deficient PD-1⁺ cells **(b)**, gp33⁺ cells **(c)** and proliferating (Ki67⁺) antigen-responsive cells **(d)** at the indicated time points. **(e-k)** Mixed bone marrow chimeric mice harbouring Myb-cKO and *Mybfl/fl* littermate control (Ctrl) T cells were infected with LCMV-Docile and analysed at the indicated time points p.i. **(e)** Schematic of the experimental setup. Flow cytometry plots and quantifications showing the frequencies of control and Myb-deficient PD-1⁺ cells **(f, g)**, gp33⁺ cells **(h, i)** and proliferating (Ki67⁺) antigen-responsive cells **(j, k)** at the indicated time points. Symbols and error bars in **(b-d, g, I, k)** indicate mean and s.e.m., respectively. *P* values are from Mann-Whitney tests; n.s., not significant; * *p* < 0.05, ** *p* < 0.01, *** *p* < 0.001, **** *p* < 0.0001.

Reviewer Figure 4. (a, b) Mixed bone marrow chimeric mice harbouring Myb-cKO and *Mybl1/fl* littermate or *Cd4Cre* (Ctrl) T cells were infected with LCMV-Docile, and antigen-responsive T cells were analysed BCL2 expressions. Box plots indicate range, interquartile and median, each dot represents an individual mouse; *P* values are from two-tailed unpaired t-tests; n.s., not significant.

Reviewer Figure 5. (a-g) *Bach2*fl/fl*Cd4*Cre (Bach2-cKO) and *Bach2*fl/fl littermate (Ctrl) mice were infected with LCMV-Docile and analysed at day 12 p.i. (a) Schematic of the experimental setup. (b) Quantification showing total number of splenocytes at day 12 p.i. (c) Flow cytometry plots and quantification showing the frequencies of gp33+ antigen-specific T cells. (d-g) Flow cytometry plots and corresponding quantifications showing the frequencies of (d) TPEX, TEX, (e) CD62L+ TPEX, (f) c-Kit+ TPEX and (g) CX3CR1+ TEX cells. (h-l) *Bach2*fl/fl*Cd4*Cre (Bach2-cKO) and *Bach2*fl/fl littermate (Ctrl) mice were infected with LCMV-Docile and analysed at indicated time points p.i. (h) Schematic of the experimental setup. Quantifications showing frequencies of (i) PD-1+, (j) gp33+, (k) PD-1+ TPEX and (l) gp33+ TPEX cells among total CD8+ T cells at the indicated time points. (m-o) Congenically marked *Myb*fl/fl*Cd4*Cre (Myb-cKO) and *Cd4*Cre (Ctrl) P14 T cells were adoptively transferred into naïve recipient mice, which were then infected with LCMV-Docile. Splenic P14 subsets were sorted at day 7 post infection and processed for bulk RNA sequencing. (m) Schematic of the experimental setup. (n) Representative flow cytometry plots showing the lack of CD62L+ TPEX cells in Myb-cKO P14 TPEX cells at day 7 p.i. compared with control P14 TPEX cells (o) Quantification showing *Bach2* expressions in the indicated samples. Box plots indicate range, interquartile and median, each dot represents an individual mouse; symbols and error bars in (i-j) indicate mean and s.e.m., respectively. *P* values are from two-tailed unpaired *t*-tests (b-g) and Mann-Whitney tests (i-l, o); n.s., not significant; * *p* <0.05, ** *p* <0.01, *** *p* <0.001, **** *p* <0.0001.

Reviewer Figure 6. Naïve C57BL/6 mice were infected with LCMV-Cl13 and TPEX-cell-enriched (PD-1+TIM-3^{lo}) CD8⁺ T cells were sorted and subjected to single cell (sc) RNA sequencing at day 30 post infection (p.i.) (GSM5135522, GSM5135523 - Dähling *et al.* Immunity 2022). The resulting data was combined with another publicly available scRNA sequencing dataset of murine exhausted CD8⁺ T cells (GSE122712 - Miller *et al.* Nat. Immunol. 2019) and analysed. **(a)** UMAP projections of the total 15,743 single exhausted T cells coloured according to cluster classification. **(b)** UMAP projections of single exhausted T cells coloured according to cluster classification in dataset-independent manner .

Reviewer Figure 7. (a-c) Congenically marked WT P14 T cells were adoptively transferred into naïve recipient mice, which were then infected with LCMV-Docile. Splenic CD62L⁺ TPEX and CD62L⁻ TPEX cells were sorted at day 7 post infection and processed for bulk RNA sequencing. (a) Schematic of the experimental setup. (b) Volcano plot showing key signature transcripts of Sell⁺ and Sell⁻ TPEX clusters being differentially in the sorted TPEX cells. (c) Barcode plot showing significant enrichment of the Sell⁺ TPEX transcriptional signature in sorted CD62L⁺ compared with sorted CD62L⁻ TPEX cells. (d) Congenically marked *wildtype* P14 cells were adoptively transferred to naïve C57BL/6 mice, which were then infected with LCMV-Docile. Splenocyte suspensions at day 22 p.i. were prepared and incubated with 100 μ M of TAPI-1 for 1 h at 37°C to inhibit shedding of CD62L (Mahnke et al. J. Invest. Dermatol. 2017). CD62L expressions on P14 T cell subsets were then analysed using flow cytometry. Box plots indicate range, interquartile and median, each dot represents an individual mouse. *P* values are from two-tailed unpaired t-tests (d); n.s., not significant.

Reviewer Figure 8. (a-c) Lethally irradiated CD45.1⁺ mice were reconstituted with mixed bone marrow harvested from *Myb^{fl/fl}Cd4^{Cre}* and either littermate *Myb^{fl/fl}* or *Cd4^{Cre}* (Ctrl) mice. Reconstitution was analysed 6~8 weeks after bone marrow transplantation. **(a)** Schematic of the experimental setup. **(b)** Flow cytometry plots and **(c)** quantification showing expressions of CD62L and CD44 in CD8⁺ T cells in the blood of reconstituted mixed bone marrow chimeras pre-infection. Horizontal lines in **(c)** represent median; each dot represents an individual mouse. *P* values are from Mann-Whitney test **(c)**; n.s., not significant; * *p* < 0.05, ** *p* < 0.01, *** *p* < 0.001, **** *p* < 0.0001.

Reviewer Reports on the First Revision:

Referees' comments:

Referee #1 (Remarks to the Author):

The authors have done an excellent job providing many experimental data to address my concerns. I believe that the revised manuscript has been significantly strengthened by these new findings, which support the authors' original conclusions.

Referee #2 (Remarks to the Author):

My comments have been adequately addressed.

Referee #3 (Remarks to the Author):

In the revised version of their manuscript the authors have ruled out that the Tpex subset defined by CD62L corresponds to previously investigated subset defined by CD69 expression. The authors have also carefully addressed most of my other specific points. These experiments are informative, well-performed and well-controlled.

The authors draw 2 central conclusions from their work. They have

- "identified a new population of precursor exhausted T cells with true stem-like potential"
- "identified c-Myb as a central transcriptional regulator of the exhausted T cell network"

These findings/conclusions are not fundamentally novel. They are essentially the same as those drawn from the original discovery of Tpex. The original work has already shown that Tpex are needed for the proliferative response to checkpoint blockade and demonstrated that the transcription factor Tcf1 is essential to maintain Tpex and thus the CD8+ T cell response to chronic viral infection (Im 2016, Utzschneider 2016, Wu 2016). The authors refine these analyses and show that most of the progenitor function of Tpex is mediated by Tcf1+ cells expressing CD62L (around 25% cells). Further, the authors identify Myb as transcription factor necessary for the formation/maintenance of Tpex, extending a growing list of transcription factors with similar roles including Tcf1, Tox or Bach2 (Yao et al Nat Imm 2021).